# An evaluation of multi-fidelity methods for quantifying uncertainty in projections of ice-sheet mass change

**John D. Jakeman**[1], **Mauro Perego**[2], **D. Thomas Seidl**[2], **Tucker A. Hartland**[3], **Trevor R. Hillebrand**[4], **Matthew J. Hoffman**[4], **and Stephen F. Price**[4]

[1]Optimization and Uncertainty Quantification, Sandia National Laboratories, Albuquerque, NM 87123, USA

[2]Scientific Machine Learning, Sandia National Laboratories, Albuquerque, NM 87123, USA

[3]Center for Applied Scientific Computing, Lawrence Livermore National Laboratory, Livermore, CA 94550, USA

[4]Fluid Dynamics and Solid Mechanics Group, Los Alamos National Laboratory, Los Alamos, NM 87544, USA

**Correspondence:** John D. Jakeman (jdjakem@sandia.gov)

**Abstract.** This study investigated the computational benefits of using multi-fidelity statistical estimation (MFSE) algorithms to quantify uncertainty in the mass change of Humboldt Glacier, Greenland, between 2007 and 2100 using a single climate change scenario. The goal of this study was to determine whether MFSE can use multiple models of varying cost and accuracy to reduce the computational cost of estimating the mean and variance of the projected mass change of a glacier. The problem size and complexity were chosen to reflect the challenges posed by future continental-scale studies while still facilitating a computationally feasible investigation of MFSE methods. When quantifying uncertainty introduced by a high-dimensional parameterization of the basal friction field, MFSE was able to reduce the mean-squared error in the estimates of the statistics by well over an order of magnitude when compared to a single-fidelity approach that only used the highest-fidelity model. This significant reduction in computational cost was achieved despite the low-fidelity models used being incapable of capturing the local features of the ice-flow fields predicted by the high-fidelity model. The MFSE algorithms were able to effectively leverage the high correlation between each model's predictions of mass change, which all responded similarly to perturbations in the model inputs. Consequently, our results suggest that MFSE could be highly useful for reducing the cost of computing continental-scale probabilistic projections of sea-level rise due to ice-sheet mass change.

## 1 Introduction

The most recent Intergovernmental Panel on Climate Change (IPCC) report predicts that the melting of ice sheets will contribute significantly to future rises in sea level (Masson-Delmotte et al., 2021), but the amount of sea-level rise is subject to a large degree of uncertainty. For example, estimates of the sea-level rise in 2100, caused by melting of the Greenland Ice Sheet, range from 0.01 to 0.18 m. Moreover, projections of the Antarctic Ice Sheet's contribution to sea-level rise are subject to even larger uncertainty (Bakker et al.,

2017; Masson-Delmotte et al., 2021; Edwards et al., 2019). Consequently, there is a strong need to accompany recent improvements in the numerical modeling of ice-sheet dynamics with rigorous methods that quantify uncertainty in model predictions.

Accurately quantifying uncertainty in ice-sheet predictions requires estimating the impacts of all sources of model variability. Prediction uncertainty is caused by three main factors, among others: (1) the inadequacy of the governing equations used by the model to approximate reality, (2) the errors introduced by the numerical discretization used to

solve the governing equations, and (3) the uncertainty in model parameters used to parameterize future climate forcing and the current condition of the ice sheet. Several studies have demonstrated that model discretization significantly affects model predictions (Cornford et al., 2013; Durand et al., 2009), but the impact of discretization errors has not been explicitly considered with other sources of uncertainty. In addition, while the comparison of model outputs has been used to estimate uncertainty arising from model inadequacy (Goelzer et al., 2018), such studies are not guaranteed to estimate the true model inadequacy (Knutti et al., 2010). Consequently, several recent efforts have focused solely on quantifying parametric uncertainty (Nias et al., 2023; Edwards et al., 2021; Ritz et al., 2015; Schlegel et al., 2018; Recinos et al., 2023), as we do in this study.

Parametric uncertainty is often estimated using Monte Carlo (MC) statistical estimation methods, which compute statistics or construct probability densities using a large number of model simulations evaluated at different random realizations of the uncertain model parameters. However, the substantial computational cost of evaluating ice-sheet models limits the number of model simulations that can be run and thus the precision of uncertainty estimates. For example, when estimating the mean of a model with MC, the mean-squared error (MSE) in the estimated value only decreases linearly as the number of model simulations increases. Therefore, recent uncertainty quantification (UQ) efforts have resulted in the construction of emulators (also known as surrogates) of the numerical model from a limited number of simulation data and in sampling of the surrogate to quantify uncertainty (Berdahl et al., 2021; Bulthuis et al., 2019; Edwards et al., 2019; Jantre et al., 2024). While surrogates can improve the computational tractability of UQ when uncertainty is parameterized by a small number of parameters, their application becomes impractical when there are more than 10–20 variables. This limitation arises because the number of simulation data required to build these surrogates grows exponentially with the number of parameters (Jakeman, 2023). Consequently, there is a need for methods capable of quantifying uncertainty in ice-sheet models parameterized by a large number of uncertain parameters, such as those used to characterize a spatially varying basal friction field.

Most recent studies have focused on estimating uncertainty in the predictions of ice-sheet models with small numbers of parameters, (e.g., Nias et al., 2023; Ritz et al., 2015; Schlegel et al., 2018; Jantre et al., 2024), despite large numbers of parameters being necessary to calibrate the ice-sheet model to observations (Barnes et al., 2021; Johnson et al., 2023; Perego et al., 2014). However, recently Recinos et al. (2023) used the adjoint sensitivity method to construct a linear approximation of the map from a high-dimensional parameterization of the uncertain basal friction coefficient and ice stiffness to quantities of interest (QoIs) – specifically the loss of ice volume above flotation predicted by a shallow-shelf approximation model at various future times. The linearized map and the Gaussian characterization of the distribution of the parameter uncertainty were then exploited to estimate statistics of the QoIs. While this method is very computationally efficient, linearizing the parameter-to-QoI map will introduce errors (bias) into estimates of uncertainty, which will depend on how accurately the linearized parameter-to-QoI map approximates the true map (Koziol et al., 2021). Moreover, the approach requires using adjoints or automatic differentiation to estimate gradients, which many ice-sheet models do not support. Consequently, in this study we focused on multi-fidelity statistical estimation (MFSE) methods that do not require gradients.

MFSE methods (Giles, 2015; Peherstorfer et al., 2016; Gorodetsky et al., 2020; Schaden and Ullmann, 2020) utilize models of varying fidelity, that is models with different inadequacy, numerical discretization, and computational cost, to efficiently and accurately quantify parametric uncertainty. Specifically, MFSE methods produce unbiased statistics of a trusted highest-fidelity model by combining a small number of simulations of that model with larger volumes of data from multiple lower-cost models. Note that while low-fidelity models with different discretization and inadequacy error are used, MFSE does not quantify the impact of these two types of errors in the high-fidelity statistics. Furthermore, provided the low-fidelity models are highly correlated with the high-fidelity model and are substantially cheaper to simulate, the mean-squared error (MSE) of the MFSE statistic will often be an order-of-magnitude smaller than the estimate obtained using solely high-fidelity evaluations for a fixed computational budget. However, such gains have yet to be realized when quantifying uncertainty in ice-sheet models.

This study investigated the efficacy of using MFSE methods to reduce the computational cost needed to estimate statistics summarizing the uncertainty in predictions of sea-level change obtained using ice-sheet models parameterized by large numbers of parameters. To facilitate a computationally feasible investigation, we focused on reducing the computational cost of estimating the mean and variance of mass change in Humboldt Glacier in northern Greenland. This mass change was driven by uncertainty in the spatially varying basal friction between the ice sheet and land mass, under a single climate change scenario between 2007 and 2100. Specifically, letting $f$ denote the mass change in 2100 computed by a mono-layer higher-order (MOLHO) (Dias dos Santos et al., 2022) model $\mathcal{M}$, $\boldsymbol{\theta}$ denote the parameters of the model characterizing the basal friction field, and $\boldsymbol{y}$ denote the observational data, we estimated the mean and variance of the distribution $p(f \mid \mathcal{M}, \boldsymbol{y}) = p(f \mid \boldsymbol{\theta})p(\boldsymbol{\theta} \mid \mathcal{M}, \boldsymbol{y})$ in two steps. First, using a piecewise linear discretization of a log-normal basal friction field, we used Bayesian inference to calibrate the resulting 11 536-dimensional uncertain variable to match available observations of glacier surface velocity. Specifically, we constructed a low-rank Gaussian approximation (Isaac et al., 2015; Recinos et al., 2023;

Barnes et al., 2021; Johnson et al., 2023; Perego et al., 2014) of the Bayesian posterior distribution of the model parameters $p(\boldsymbol{\theta} \mid \mathcal{M}, \boldsymbol{y})$ using a Blatter–Pattyn model (Hoffman et al., 2018). Second, we estimated the mean and variance of glacier mass change using 13 different model fidelities (including the highest-fidelity model), based on different numerical discretizations of the MOLHO physics approximation and shallow-shelf approximation (SSA; Morland and Johnson, 1980; Weis et al., 1999).

Our study makes two novel contributions to previously published glaciology literature. First, it represents the first application of MFSE methods to quantify the impact of high-dimensional parameter uncertainty in transient projections of ice-sheet models defined on a realistic physical domain. Our results demonstrate that MFSE can reduce the serial computational time required for a precise UQ study of the ice-sheet contribution to sea-level rise from years to a month. Note that Gruber et al. (2023) previously applied MFSE to an ice-sheet model; however, their study was highly simplified, as it only quantified uncertainty arising from two uncertain scalar parameters of an ice-sheet model defined on a simple geometric domain. Second, our paper provides a comprehensive discussion of the practical issues that arise when using MFSE, which are often overlooked in the MFSE literature.

This paper is organized as follows. First, Sect. 2 details the different ice-sheet models considered by this study and the parameterization of uncertainty employed. Second, Sect. 3 presents the calibration of the ice-sheet model and how the posterior samples were generated. Third, Sect. 4 presents the MFSE methods that were used to quantify uncertainty. Fourth, Sect. 5 presents the numerical results of the study and Sect. 6 presents our findings. Finally, conclusions are drawn in Sect. 7.

## 2 Methods

This section presents the model formulations (Sect. 2.1) and the numerical discretization of these models (Sect. 2.2) we used to model ice-sheet evolution, as well as the sources of model uncertainty we considered (Sect. 2.3) when quantifying uncertainty in the mass change from Humboldt Glacier between 2007 and 2100.

### 2.1 Model formulations

Ice sheets behave as a shear thinning fluid and can be modeled with the nonlinear Stokes equations (Cuffey and Paterson, 2010). This section details the Stokes equations and two computationally less expensive simplifications, MOLHO (Dias dos Santos et al., 2022) and SSA (Morland and Johnson, 1980; Weis et al., 1999), which were used to quantify uncertainty in predictions of the contribution of Humboldt Glacier to sea-level rise.

Let $x$ and $y$ denote the horizontal coordinates and $z$ the vertical coordinate, chosen such that the sea level, assumed to remain constant during the period of interest, corresponds to $z = 0$. We approximated the ice domain at time $t$ as a vertically extruded domain $\Omega$ defined as

$$\Omega(t) := \{(x, y, z) \text{ s.t. } (x, y) \in \Sigma \text{ and } l(x, y, t) < z < s(x, y, t)\},$$

where s.t. denotes "such that" CE1, $\Sigma \subset \mathbb{R}^2$ denotes the horizontal extent of the ice, $\Gamma_l(t) := \{(x, y, z) \text{ s.t. } z = l(x, y, t), (x, y) \in \Sigma\}$ denotes the lower surface of the ice at time $t$, and $\Gamma_s(t) := \{(x, y, z) \text{ s.t. } z = s(x, y, t), (x, y) \in \Sigma\}$ denotes the upper surface of the ice.[1]

The Stokes, MOLHO, and SSA models defined the thickness of the ice $H(x, y, t) = s(x, y, t) - l(x, y, t)$ as the difference between the ice-sheet surface $s(x, y, t)$ and the bottom of the ice sheet $l(x, y, t)$. The bottom of the ice sheet was allowed to be either grounded to the bed topography $b(x, y)$, such that $l(x, y, t) = b(x, y)$, or floating, such that $l(x, y, t) = -\frac{\rho}{\rho_w} H(x, y, t)$, where $\rho$ and $\rho_w$ are the densities of ice and ocean water, respectively. Different boundary conditions were then applied to the grounded portion $\Gamma_g$ of the ice bottom and to the floating portion $\Gamma_f$ of the ice bottom, where $\Gamma_g \cap \Gamma_f = \varnothing$ and the ice bottom is given by $\Gamma_g \cup \Gamma_f$. The lateral boundary of $\Omega$ was also partitioned into the ice-sheet margin (either terrestrial or marine margin) $\Gamma_m$ and an internal (artificial) boundary $\Gamma_d$ marking the interior extent of Humboldt Glacier that was considered. The relevant domains of the ice sheet are depicted in Fig. 1.

The Stokes equations model the horizontal ice velocities $(u(x, y, z, t), v(x, y, z, t))$, vertical ice velocity $w(x, y, z, t)$, and thickness $H(x, y, t)$ of an ice sheet as a function of the three spatial dimensions $(x, y, z)$. In contrast, the MOLHO model makes simplifications based on the observation that ice sheets are typically shallow; i.e., their horizontal extent is much greater than their thickness. These simplifications lead to a model that does not explicitly estimate the vertical velocity $w$ and only simulates the horizontal velocities $u(x, y, z, t)$, $v(x, y, z, t)$ as functions of the three spatial coordinates. In contrast again, the SSA model makes the additional assumption that the horizontal components of velocity do not vary with thickness (a reasonable approximation in regions where motion is dominated by basal slip) so that the horizontal velocities $u(x, y, t), v(x, y, t)$ are solved for only as functions of $(x, y)$.

The Stokes, MOLHO, and SSA models all evolve ice thickness $H(x, y, t)$ according to

$$\partial_t H + \nabla \cdot (\overline{\mathbf{u}} H) = f_H, \qquad H \geq 0, \tag{1}$$

where $\overline{\mathbf{u}} := \frac{1}{H} \int_l^s \mathbf{u} \, dz$ is the thickness-integrated velocity and $f_H$ is a forcing term that accounts for accumulation (e.g., snow accumulation) and ablation (e.g., melting) at the

---

[1]For simplicity, here we assume that $\Sigma$ does not change over time. This implies that the ice sheet cannot extend beyond $\Sigma$ but it can become thicker or thinner (to the point of disappearing in some regions).

(a)

(b)

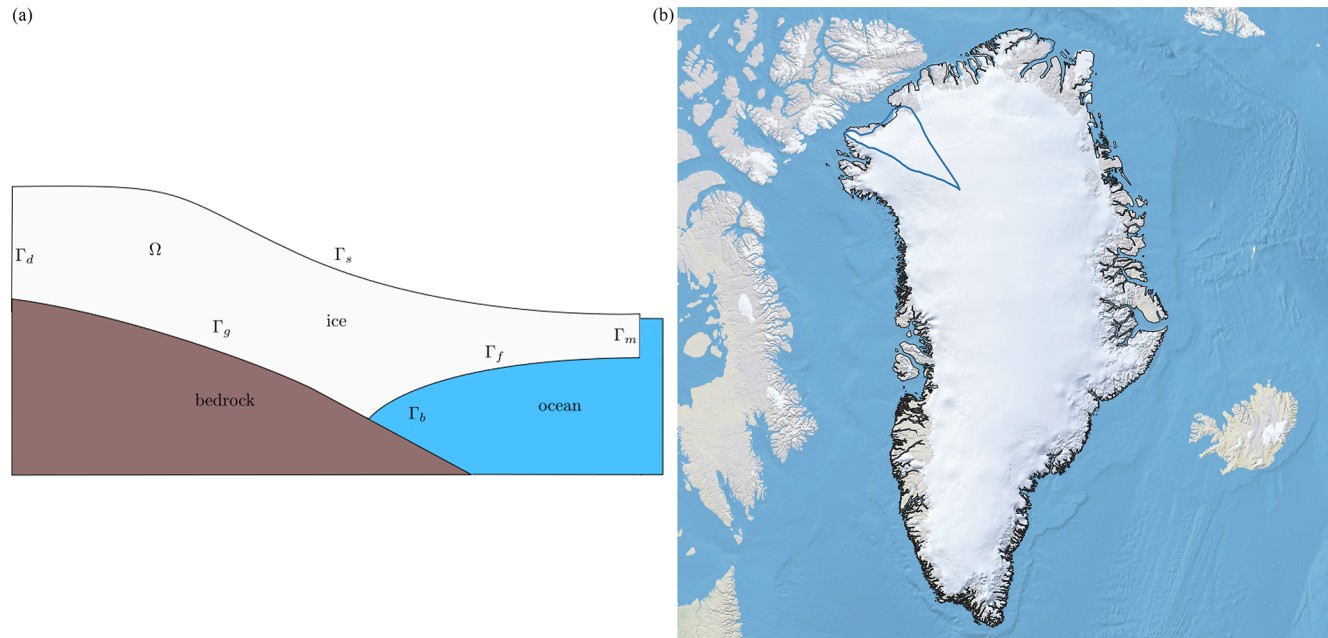

**Figure 1. (a)** Conceptual model of an ice sheet in the $x$–$z$ plane. **(b)** The boundaries (blue lines) of Humboldt Glacier in Greenland.

upper ($s$) and lower ($l$) surfaces of the ice sheet. However, each model determines the velocities of the ice sheet differently. The following three subsections detail how each model computes the velocity of the ice sheet.

### 2.1.1 Stokes model

This section introduces the Stokes model, which while not used in this study due to its impractical computational cost, forms the basis of the other three models used in this study. Specifically, the governing equations of the Stokes model are

$$-\nabla \cdot \boldsymbol{\sigma} = \rho \mathbf{g}, \tag{2}$$

$$\nabla \cdot \mathbf{u} = 0, \tag{3}$$

where these equations are solved for the velocities $\mathbf{u} = (u, v, w)$ and the pressure $p$. Additionally, $\mathbf{g} = (0, 0, -g)$ denotes the gravitational acceleration pointing downward ($g = 9.81\,\mathrm{m\,s^{-2}}$), $\rho$ denotes the density of ice, $\boldsymbol{\sigma} = 2\mu\mathbf{D} - pI$ denotes the stress tensor of the ice, and $\mathbf{D}_{ij}(\mathbf{u}) = \frac{1}{2}\left(\frac{\partial u_i}{\partial x_j} + \frac{\partial u_j}{\partial x_i}\right)$ denotes the strain-rate tensor of the ice; here we used the shorthand $\mathbf{u} = (u, v, w) = (u_1, u_2, u_3)$. The stress tensor is dependent on the nonlinear viscosity of the ice which satisfies

$$\mu = \frac{1}{2} A(T)^{-q} D_{\mathrm{e}}(\mathbf{u})^{q-1}, \tag{4}$$

where $A$ is the ice-flow factor that depends on the ice temperature $T$ and $q \leq 1$; in our study we set $q = \frac{1}{3}$, which is a typical choice (Hillebrand et al., 2022). In addition, the effective strain rate $D_{\mathrm{e}}(\mathbf{u})$ satisfies $D_{\mathrm{e}}(\mathbf{u}) = \frac{1}{\sqrt{2}}|\mathbf{D}(\mathbf{u})|$, where $|\cdot|$ denotes the Frobenius norm.

When used to model ice sheets, the Stokes equation must be accompanied by the following boundary conditions:

$$\begin{cases} \boldsymbol{\sigma}\mathbf{n} = 0 & \text{on } \Gamma_s & \text{stress-free,} \\ & & \text{atmospheric pressure neglected} \\ \boldsymbol{\sigma}\mathbf{n} = \rho_{\mathrm{w}}\,g\,\min(z, 0)\mathbf{n} & \text{on } \Gamma_{\mathrm{m}} & \text{boundary condition at} \\ & & \text{ice margin} \\ \mathbf{u} = \mathbf{u}_{\mathrm{d}} & \text{on } \Gamma_d & \text{Dirichlet condition at internal} \\ & & \text{boundary (ice-flow divide)} \\ \mathbf{u} \cdot \mathbf{n} = 0,\ (\boldsymbol{\sigma}\mathbf{n})_\parallel = \beta\mathbf{u}_\parallel & \text{on } \Gamma_g & \text{impenetrability} \\ & & + \text{sliding condition} \\ \boldsymbol{\sigma}\mathbf{n} = \rho_{\mathrm{w}}\,g\,z\,\mathbf{n} & \text{on } \Gamma_f & \text{hydrostatic pressure of ocean} \\ & & \text{under ice shelves.} \end{cases}$$

Here $\beta(x, y)$ is a linearized sliding (or friction) coefficient and $\mathbf{n}$ the outward-pointing unit normal to the boundary and the subscript $\parallel$ denotes the component tangential to the bed. The boundary condition at the margin includes an ocean back-pressure term when the margin is partially submerged ($z < 0$). For a terrestrial margin, $z > 0$, the boundary condition becomes a stress-free condition.

### 2.1.2 Mono-layer higher order (MOLHO)

The MOLHO model (Dias dos Santos et al., 2022) is based on the Blatter–Pattyn approximation (Pattyn, 2003; Dukowicz et al., 2010), which can be derived by neglecting the terms $w_x$ and $w_y$ (the derivatives of $w$ with respect to $x$ and $y$, respectively) in the strain-rate tensor $\mathbf{D}$ and using the incompressibility condition ($\nabla \cdot \mathbf{u} = 0$) such that $w_z$ can be expressed solely in terms of $u_x$ and $v_y$ and TS1

$$\mathbf{D} = \begin{bmatrix} u_x & \frac{1}{2}(u_y + v_x) & \frac{1}{2}u_z \\ \frac{1}{2}(u_y + v_x) & v_y & \frac{1}{2}u_z \\ \frac{1}{2}u_z & \frac{1}{2}v_z & -(u_x + v_y) \end{bmatrix}. \tag{5}$$

This leads (Jouvet, 2016) to the following elliptic equations for the horizontal velocities $(u, v)$:

$$-\nabla \cdot (2\mu\hat{\mathbf{D}}) = -\rho g \nabla_{xy} s, \qquad (6)$$

where $\nabla_{xy} := [\partial_x, \partial_y]^\top$, and

$$\hat{\mathbf{D}} = \begin{bmatrix} 2u_x + v_y & \frac{1}{2}(u_y + v_x) & \frac{1}{2}u_z \\ \frac{1}{2}(u_y + v_x) & u_x + 2v_y & \frac{1}{2}v_z \end{bmatrix} \qquad (7)$$

such that the viscosity $\mu$ in Eq. (4) has the effective strain rate

$$D_e = \sqrt{u_x^2 + v_y^2 + u_x v_y + \frac{1}{4}(u_y + v_x)^2 + \frac{1}{4}u_z^2 + \frac{1}{4}v_z^2}.$$

The MOLHO model is derived from the weak form of the Blatter–Pattyn model (Eq. 6), with the ansatz that the velocity can be expressed as

$$\mathbf{u}(x, y, z) = \mathbf{u}_b(x, y)\,\phi_b + \mathbf{u}_v(x, y)\,\phi_v\left(\frac{s - z}{H}\right),$$

$$\text{with} \quad \phi_b = 1 \quad \text{and} \quad \phi_v(\zeta) = 1 - \zeta^{\frac{1}{q}+1},$$

where the functions $\phi_b$ and $\phi_v$ are also used to define the test functions of the weak formulation of the MOLHO model. This ansatz allows the Blatter–Pattyn model to be simplified into a system of two two-dimensional partial differential equations (PDEs) for $\mathbf{u}_b$ and $\mathbf{u}_v$ – Dias dos Santos et al. (2022) give a detailed derivation – such that the thickness-averaged velocity satisfies $\overline{\mathbf{u}} = \mathbf{u}_b + \frac{(1+q)}{(1+2q)}\,\mathbf{u}_v$, where $q$ is the same coefficient appearing in the viscosity definition Eq. (4).

We used the following boundary conditions when using MOLHO to simulate ice flow:

$$\begin{cases} 2\mu\hat{\mathbf{D}}\,\mathbf{n} = 0 & \text{on } \Gamma_s & \text{stress-free, atmospheric} \\ & & \text{pressure neglected} \\ 2\mu\hat{\mathbf{D}}\,\mathbf{n} = \psi\mathbf{n} & \text{on } \Gamma_m & \text{boundary condition at} \\ & & \text{ice margin} \\ \mathbf{u} = \mathbf{u}_d & \text{on } \Gamma_d & \text{Dirichlet condition at internal} \\ & & \text{boundary (ice-flow divide)} \\ 2\mu\hat{\mathbf{D}}\,\mathbf{n} = \beta\mathbf{u}_\parallel & \text{on } \Gamma_g & \text{sliding condition} \\ 2\mu\hat{\mathbf{D}}\,\mathbf{n} = 0 & \text{on } \Gamma_f & \text{free slip under ice shelves.} \end{cases}$$

Additionally, we approximated the term $\psi = \rho g(s - z) + \rho_w g \min(z, 0)$ by its thickness-averaged value $\overline{\psi} = \frac{1}{2}gH(\rho - r^2\rho_w)$, where $r = \max\left(1 - \frac{s}{H}, 0\right)$ is the submerged ratio.

### 2.1.3 Shallow-shelf approximation (SSA)

The SSA model (Morland and Johnson, 1980) is a simplification of the Blatter–Pattyn model that assumes the ice velocity is uniform in $z$, so $\mathbf{u} = \overline{\mathbf{u}}$ and thus $u_z = 0$, $v_z = 0$. This simplification yields

$$\mathbf{D} = \begin{bmatrix} u_x & \frac{1}{2}(u_y + v_x) & 0 \\ \frac{1}{2}(u_y + v_x) & v_y & 0 \\ 0 & 0 & -(u_x + v_y) \end{bmatrix},$$

$$\hat{\mathbf{D}} = \begin{bmatrix} 2u_x + v_y & \frac{1}{2}(u_y + v_x) & 0 \\ \frac{1}{2}(u_y + v_x) & u_x + 2v_y & 0 \end{bmatrix}, \qquad (8)$$

and $D_e = \sqrt{u_x^2 + v_y^2 + u_x v_y + \frac{1}{4}(u_y + v_x)^2}$. Consequently, the SSA is a single two-dimensional PDE in $\Sigma$:

$$-\nabla \cdot \left(2\mu H\hat{\mathbf{D}}(\overline{\mathbf{u}})\right) + \beta\overline{\mathbf{u}} = -\rho g H \nabla_{xy} s, \quad \text{in } \Sigma, \qquad 35$$

where $\overline{\mu} = \frac{1}{2}\overline{A}(T)^{-q} D_e(\overline{\mathbf{u}})^{q-1}$ and $\overline{A}$ is the thickness-averaged flow factor. This study explored the use of SSA with the boundary conditions

$$\begin{cases} 2\mu H\hat{\mathbf{D}}(\overline{\mathbf{u}})\,\mathbf{n} = H\overline{\psi}\mathbf{n} & \text{on } \Gamma_m & \text{boundary condition at} \\ & & \text{ice margin} \\ \overline{\mathbf{u}} = \overline{\mathbf{u}}_d & \text{on } \Gamma_d & \text{Dirichlet condition at} \\ & & \text{internal boundary.} \end{cases}$$

With abuse of notation, here $\Gamma_m$ and $\Gamma_d$ denote subsets of the boundary of $\Sigma$.

### 2.2 Numerical discretization

The ability to predict ice-sheet evolution accurately is dictated not only by the governing equations used, but also by the properties of the numerical methods used to solve the governing equations. In this study, we discretized the thickness and the velocity equations of the MOLHO and SSA models using the popular Galerkin-based finite-element method with piecewise linear elements, which we implemented in FEniCS (Alnæs et al., 2015). The coupled thickness and velocity equations were solved in a semi-implicit fashion using a backward Euler time discretization for the thickness and lagging the evaluation of the velocity. The thickness equation was stabilized using the streamline upwind method. Additionally, the advection term was integrated by parts and the thickness was treated implicitly. The discretized problem was solved using the PETSc (Balay et al., 1998) scalable nonlinear equation solver (SNES). Using this time evolution process, we did not observe any numerical instabilities when using the time-step sizes adopted in this study.

Because the thickness $H$ obtained from Eq. (1) is not guaranteed to be positive due to the forcing term $f_H$ and the discretization used is not positivity preserving, we adopted two different approaches to guarantee the positivity of the thickness computed by our finite-element models. The first approach involved updating the thickness value at each node so that it was greater than or equal to a minimum thickness value $H_m = 1$ m. The second approach used an optimization-based approach (Bochev et al., 2020) to preserve the thickness constraint ($H \geq H_m$) and guarantee that the total mass change is always consistent with the forcing term in regions where the ice is present and with the boundary fluxes. The first approach is computationally cheaper than the second, but unlike the second method, it does not conserve mass.

In addition to mass conservation, the number of finite elements and the time-step size both affect the error in the finite-element approximation of the governing equations of

the MOLHO and SSA models. In this study we investigated the impact of the number of finite elements, which we also refer to as the spatial mesh resolution, and time-step size on the precision of statistical estimates of mass change. Specifically, the MOLHO and SSA models were both used to simulate ice-sheet evolution with four different finite-element meshes and four different time-step sizes. More details on the spatial mesh and time-step sizes used are given in Sect. 5.1. Figure 2 compares the four different finite-element meshes used to model Humboldt Glacier. Due to the differences in the characteristic element size of each mesh, the computational domain of each mesh is different. However, we will show that this did not prevent the use of these meshes in our study.

## 2.3   Parameterization of uncertainty

Many factors introduce parametric uncertainty into the predictions of ice-sheet models, including atmospheric forcing, ice rheology, basal friction, ice temperature, calving, and submarine melting. While all these sources of parametric uncertainty may significantly impact predictions of mass change from ice sheets, this study focused on quantifying uncertainty in modeled ice mass change subject to high-dimensional parameter uncertainty due to unknown basal friction, which is considered one of the largest sources of prediction uncertainty after future environmental forcing (Nias et al., 2018; Joughin et al., 2019; Brondex et al., 2019; Åkesson et al., 2021; Hillebrand et al., 2022; Nias et al., 2023). This singular focus was made to improve our ability to assess whether MFSE is useful for quantifying uncertain in ice-sheet modeling with high-dimensional parameter uncertainty, which most existing UQ methods cannot tractably address. By doing so, we ensured that the conclusions drawn by our study can be plausibly extended to studies considering additional sources of uncertainty.

The uncertainty in basal friction $\beta$, which impacts the boundary conditions of the MOLHO and SSA models, can be parameterized in a number of ways. For example, a lumped approach would assign a single scalar random variable to the whole domain or a semi-distributed approach may use different constants in predefined subdomains, e.g., catchments, of the glacier. In this study, we adopted a fully distributed approach that treated the friction as a log-Gaussian random field that is $\theta = \log(\beta) \in \mathbb{R}^{N_\theta}$, with a Gaussian prior distribution $p(\theta) \sim \mathcal{N}(\mu, \mathcal{C})$ with $\mu = 0$.

Following Isaac et al. (2015), we defined the prior covariance operator $\mathcal{C}$ to be an infinite-dimensional Laplacian squared operator. Specifically, we used a finite-dimensional discretization of the operator $\Sigma_{\text{prior}} \approx \mathcal{C}$ with

$$\Sigma_{\text{prior}}^{-1} = \mathbf{K}\mathbf{M}^{-1}\mathbf{K}, \tag{9}$$

where $\mathbf{K} \in \mathbb{R}^{N_\theta \times N_\theta}$ and $\mathbf{M} \in \mathbb{R}^{N_\theta \times N_\theta}$ are finite-element matrices for the elliptic and mass operators, defined as

$$K_{ij} = \gamma \int\limits_{\Gamma_l} \nabla\phi_i(\boldsymbol{x}) \cdot \nabla\phi_j(\boldsymbol{x})\mathrm{d}\boldsymbol{x} + \delta \int\limits_{\Gamma_l} \phi_i(\boldsymbol{x}) \cdot \phi_j(\boldsymbol{x})\mathrm{d}\boldsymbol{x}$$
$$+ \xi \int\limits_{\partial\Gamma_l} \phi_i(\boldsymbol{x}) \cdot \phi_j(\boldsymbol{x})\mathrm{d}\boldsymbol{x}, \tag{10}$$

$$M_{ij} = \int\limits_{\Gamma_l} \phi_i(\boldsymbol{x}) \cdot \phi_j(\boldsymbol{x})\mathrm{d}\boldsymbol{x}, \tag{11}$$

where $\phi_i$ denotes finite-element basis functions and $\boldsymbol{x} = (x, y)$. The first term in the definition of $\mathbf{K}$ is the Laplacian operator, the second term is a mass operator representing a source term, and the last term is a boundary mass operator for Robin boundary conditions. The ratio of the coefficients $\gamma$ and $\delta$ determines the correlation length $l = \sqrt{\frac{\gamma}{\delta}}$ of the covariance. In our simulations, we set $\gamma = 2000\,\text{km}$, $\delta = 2\,\text{km}^{-1}$ and $\xi = 20$; hence $l \approx 31.6\,\text{km}$. These values were found to balance the smoothness of realizations of the friction field with the ability to capture the fine-scale friction features needed to produce an acceptable match between the model prediction of surface velocity and the observed values. Two random samples from the prior distribution of the log friction are depicted in Fig. 3.

The parameterization of the prior in Eq. (9) has two main advantages. First, computationally efficient linear algebra can be used to draw samples from the prior distribution. In this study we drew samples from the prior using

$$\theta = \mu_{\text{prior}} + \mathbf{L}n, \quad n \sim \mathcal{N}(\mathbf{0}, \mathbf{I}_{N_\theta}),$$

where $\mu_{\text{prior}} = \mathbf{0}$, $\mathbf{I}_{N_\theta}$ is the identity matrix with $N_\theta$ rows, $\mathbf{L} = \mathbf{K}^{-1}\mathbf{M}^{\frac{1}{2}}$ such that $\Sigma_{\text{prior}} = \mathbf{L}\mathbf{L}^\top$, and we lump the mass matrix $\mathbf{M}$. The second advantage is that this Gaussian prior enables an efficient procedure for computing the posterior distribution of the friction field constrained by the observations, which we present in Sect. 3.

## 2.4   Additional model setup

Additional details regarding the model setup are as follows. First, the glacier's bed topography, ice surface elevation, and ice thickness were obtained from observations (refer to Hillebrand et al., 2022, for details) and interpolated onto the finite-element mesh. Second, the MIROC5 climate forcing from the CMIP5 for the Representative Concentration Pathway 2.6 (RCP2.6) scenario was used to generate the surface mass balance (difference between ice accumulation and ablation) $f_H$ and drive the ice-sheet evolution from 2007 to 2100. This surface mass balance was provided by the Ice Sheet Model Intercomparison Project for CMIP6 (ISMIP6), which downscaled output from Earth system models using the state-of-the-art regional climate model MAR (Nowicki et al., 2020).

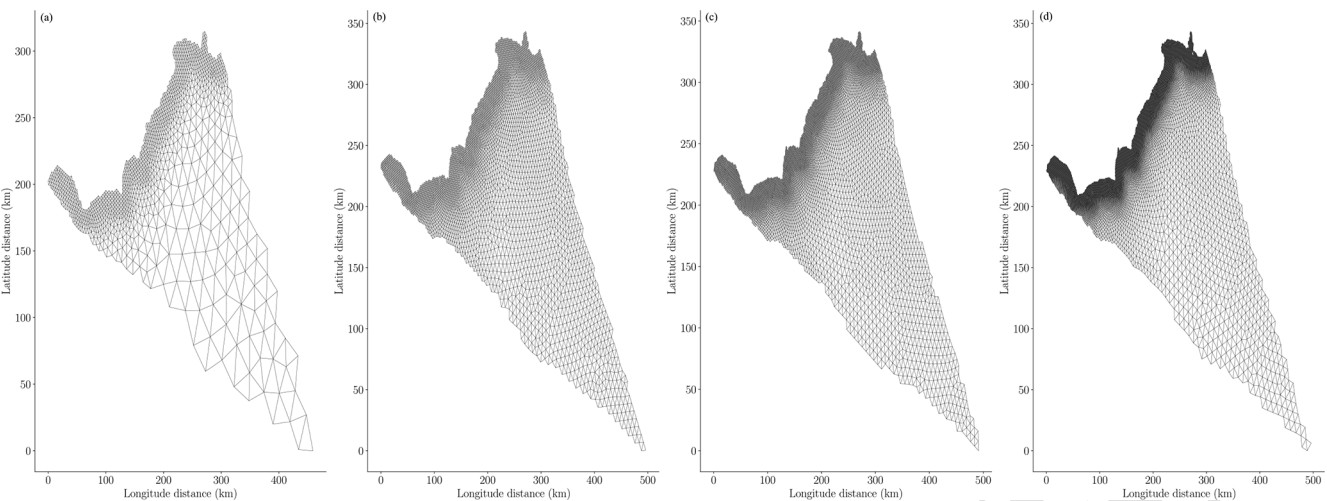

**Figure 2.** Comparison of the four finite-element meshes used to model Humboldt Glacier with characteristic finite-element sizes of 1, 1.5, 2, and 3 km, shown in **(a)** to **(d)**, respectively.

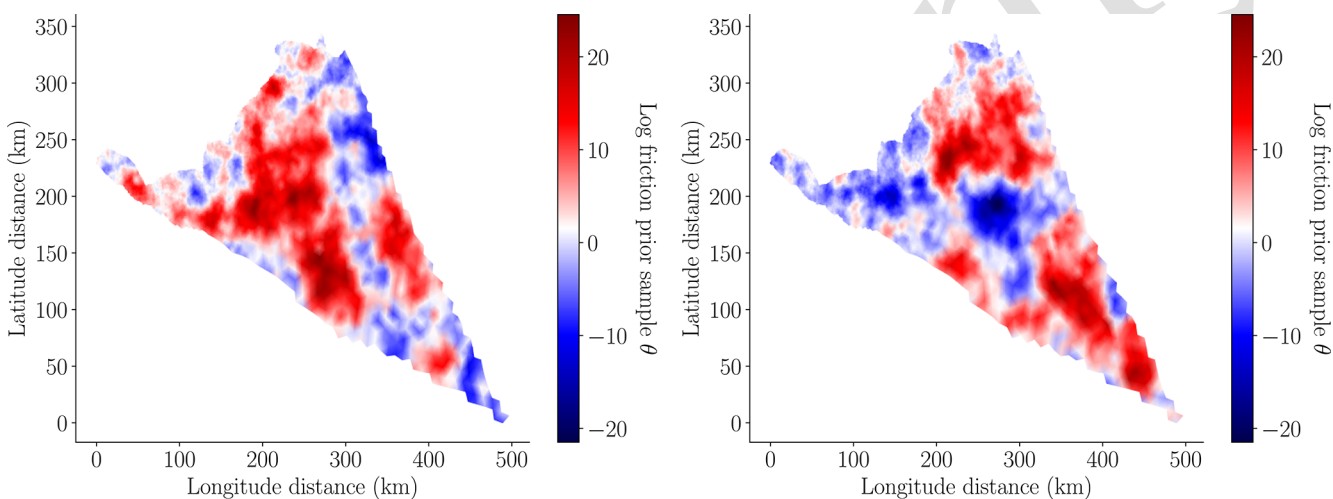

**Figure 3.** Two random samples from the prior distribution of the log friction $p(\boldsymbol{\theta}) \sim \mathcal{N}(\mathbf{0}, \boldsymbol{\Sigma}_{\text{prior}})$, where $\boldsymbol{\Sigma}_{\text{prior}}$ is defined in Eq. (9).

Finally, the ice front was kept fixed such that any ice that moved beyond the calving front was assumed to melt and any explicit ocean forcing was ignored.

## 3   Calibration

The goal of this study was to investigate uncertainty in predictions of the future mass change of Humboldt Glacier. However, generating realistic predictions with a model requires calibrating that model to available data. Consequently, in this paper we calibrated the basal friction field of our numerical models to measurements of surface velocity of the ice sheet. We processed Humboldt Glacier geometry data and surface velocity observations for year 2007 as detailed in Hillebrand et al. (2022). The geometry was assumed to be error-free, and the ice sheet was assumed to be in thermal equilibrium. Thus, we calibrated the friction field by fitting the outputs of a high-resolution steady-state thermo-coupled flow model to the observational data.

Ice-sheet models are typically calibrated using deterministic optimization methods that find the values of the model parameters that lead to the best match between observations and the model prediction of the observations, e.g., MacAyeal (1993), Morlighem et al. (2010), Petra et al. (2012), Perego et al. (2014), and Goldberg et al. (2015). However, such approaches produce a single optimized parameter value to represent the uncertainty in the model parameters that arises from using a limited number of noisy observational data.

Bayesian inference uses Bayes' theorem to quantify the probability of the parameters conditioned on the data $p(\boldsymbol{\theta} \mid \mathbf{y}) \in \mathbb{R}$, known as the posterior distribution, as proportional to the conditional probability of observing the data given the

parameters $p(\boldsymbol{y} \mid \boldsymbol{\theta}) \in \mathbb{R}$, known as the likelihood, multiplied by the prior probability assigned to the parameters $p(\boldsymbol{\theta}) \in \mathbb{R}$:

$$p(\boldsymbol{\theta} \mid \boldsymbol{y}) \propto p(\boldsymbol{y} \mid \boldsymbol{\theta}) p(\boldsymbol{\theta}).$$

In this work we assumed that the observational data (two-dimensional surface velocities $\boldsymbol{y} = \boldsymbol{u}_{\mathrm{obs}} \in \mathbb{R}^{2N_{\mathrm{obs}}}$) were corrupted by centered Gaussian noise $\boldsymbol{\eta} \sim \mathcal{N}(\mathbf{0}, \boldsymbol{\Sigma}_{\mathrm{noise}}) \in \mathbb{R}^{2N_{\mathrm{obs}}}$. Specifically, given a Blatter–Pattyn flow model $\boldsymbol{g}(\boldsymbol{\theta}) \in \mathbb{R}^{2N_{\mathrm{obs}}}$ that maps the logarithm of the basal friction to the computed surface velocity, we assumed $\boldsymbol{y} = \boldsymbol{g}(\boldsymbol{\theta}) + \boldsymbol{\eta}$ such that the likelihood function was given by

$$p(\boldsymbol{y} \mid \boldsymbol{\theta}) = (2\pi |\boldsymbol{\Sigma}_{\mathrm{noise}}|)^{-\frac{1}{2}} \exp\left(-\frac{1}{2}(\boldsymbol{y} - \boldsymbol{g}(\boldsymbol{\theta}))^{\top} \boldsymbol{\Sigma}_{\mathrm{noise}}^{-1}(\boldsymbol{y} - \boldsymbol{g}(\boldsymbol{\theta}))\right).$$

Here, $\boldsymbol{g}(\boldsymbol{\theta})$ denotes the output of the steady-state ice-sheet model at the locations of the observations for a given realization of the model parameters. We were able to calibrate the model using only a steady-state model without time stepping because we assumed that the velocity data were collected over a short period of time over which the ice-sheet state was approximately in steady state. We also assumed that the error in the observations were uncorrelated and set

$$\boldsymbol{\Sigma}_{\mathrm{noise}} = \frac{1}{\xi} \left[ \begin{array}{cc} \mathbf{U}_{\mathrm{err}} \mathbf{M}_s^{-1} \mathbf{U}_{\mathrm{err}} & \\ & \mathbf{U}_{\mathrm{err}} \mathbf{M}_s^{-1} \mathbf{U}_{\mathrm{err}} \end{array} \right] \in \mathbb{R}^{2N_{\mathrm{obs}} \times 2N_{\mathrm{obs}}}, \quad (12)$$

where $\mathbf{U}_{\mathrm{err}} = \mathrm{Diag}(\boldsymbol{u}_{\mathrm{err}})$ is the diagonal matrix containing the root mean square errors $\boldsymbol{u}_{\mathrm{err}} \in \mathbb{R}^{N_{\mathrm{obs}}}$ of the surface velocity magnitudes, $\mathbf{M}_s \in \mathbb{R}^{N_{\mathrm{obs}}}$ is the mass matrix computed on the upper surface $\Gamma_s$, and $\xi$ is a scaling term. We set $\xi = 8\,\mathrm{km}^{-2}$.

Before continuing, we wish to emphasize two important aspects of the calibration used in this study that mean our results must be viewed with some caution. First, we assumed the observational data to be uncorrelated, as assumed in most ice-sheet inference studies, including Recinos et al. (2023) and Isaac et al. (2015). Moreover, we also assumed our Gaussian error model to be exact. However, in reality, neither of these assumptions is likely to be exactly satisfied. For example, Koziol et al. (2021) showed that, for an idealized problem, ignoring spatial correlation in the observational noise can lead to uncertainty being underestimated. Second, our optimization of the MAP point was constrained by the coupled velocity flow equations and steady-state enthalpy equation, which is equivalent to implicitly assuming that the ice is in thermal equilibrium. Theoretically, this assumption could be avoided if the temperature tendencies were known, but they are not. Alternatively, transient optimization over long time periods, comparable to the temperature timescales, could be used (Adalgeirsdottir et al., 2014). However, this approach would be computationally expensive and would require including time-varying temperature data (e.g., inferred by ice cores), which are very sparse.

Quantifying uncertainty in mass-change projections conditioned on observational data requires drawing samples from the posterior of $\log(\boldsymbol{\beta})$, evaluating the transient model for each sample, and computing estimates of statistics summarizing the prediction uncertainty using those evaluations. Typically, samples are drawn using Markov chain Monte Carlo methods (Hoffman and Gelman, 2014); however such methods can be computationally intractable for high-dimensional uncertain variables (Bui-Thanh et al., 2013), such as the variable we used to parameterize basal friction. Consequently, we used the two-step method presented in Bui-Thanh et al. (2013) and Isaac et al. (2015) to construct a Laplace approximation of the posterior. Please note that, recently, variational inference has been used to infer high-dimensional basal friction (Brinkerhoff, 2022); however we did not use such methods in our study.

First, we performed a PDE-constrained deterministic optimization to compute the maximum a posteriori (MAP) point $\boldsymbol{\theta}_{\mathrm{MAP}}$:

$$\boldsymbol{\theta}_{\mathrm{MAP}} = \underset{\boldsymbol{\theta}}{\mathrm{argmin}} \frac{1}{2}(\boldsymbol{y} - \boldsymbol{g}(\boldsymbol{\theta}))^{\top} \boldsymbol{\Sigma}_{\mathrm{noise}}^{-1}(\boldsymbol{y} - \boldsymbol{g}(\boldsymbol{\theta}))$$
$$+ \frac{1}{2}(\boldsymbol{\theta} - \boldsymbol{\mu}_{\mathrm{prior}})^{\top} \boldsymbol{\Sigma}_{\mathrm{prior}}^{-1}(\boldsymbol{\theta} - \boldsymbol{\mu}_{\mathrm{prior}}), \quad (13)$$

which maximizes the posterior $p(\boldsymbol{\theta} \mid \boldsymbol{y})$. For linear models and Gaussian priors, the MAP point has close ties with the optimal solution obtained using Tikhonov regularization (Stuart, 2010). Specifically, the first term above minimizes the difference between the model predictions and the observations and the second term penalizes the deviation of the optimal point from the prior mean.

Second, we constructed a low-rank quadratic approximation of the log posterior, centered at the MAP point:

$$\log(p(\boldsymbol{\theta} \mid \boldsymbol{y})) \approx C - \frac{1}{2}(\boldsymbol{\theta} - \boldsymbol{\theta}_{\mathrm{MAP}})^{\top} \boldsymbol{\Sigma}_{\mathrm{post}}^{-1}(\boldsymbol{\theta} - \boldsymbol{\theta}_{\mathrm{MAP}}), \quad (14)$$

where

$$\boldsymbol{\Sigma}_{\mathrm{post}}^{-1} = \mathbf{H}_{\mathrm{MAP}} + \boldsymbol{\Sigma}_{\mathrm{prior}}^{-1}, \quad (15)$$

$\mathbf{H}_{\mathrm{MAP}} \in \mathbb{R}^{N_{\theta} \times N_{\theta}}$ is the Hessian of $\frac{1}{2}(\boldsymbol{y} - \boldsymbol{g}(\boldsymbol{\theta}))^{\top} \boldsymbol{\Sigma}_{\mathrm{noise}}^{-1}(\boldsymbol{y} - \boldsymbol{g}(\boldsymbol{\theta}))$ at $\boldsymbol{\theta} = \boldsymbol{\theta}_{\mathrm{MAP}}$, and $C$ is a constant independent of $\boldsymbol{\theta}$. This resulted in a Gaussian approximation of the posterior $p(\theta \mid \boldsymbol{y}) \approx \mathcal{N}(\boldsymbol{\theta}_{\mathrm{MAP}}, \boldsymbol{\Sigma}_{\mathrm{post}})$, also known as a Laplace approximation of the posterior. Naively computing the posterior covariance using the aforementioned formula for $\boldsymbol{\Sigma}_{\mathrm{post}}$ is computationally intractable. That approach requires solving $2N_{\theta}$ linearized (adjoint) flow models to compute and invert the large dense matrix $\mathbf{H}_{\mathrm{MAP}}$, which requires $\mathrm{O}(N_{\theta}^3)$ operations. For reference, in this study we use $N_{\theta} = 11\,356$ variables to parameterize the basal friction and the adjoints of the flow model had $227\,120$ unknowns. Consequently, we used a low-rank Laplace approximation, which is detailed in Appendix A, to efficiently draw random samples from the posterior distribution.

The posterior characterizes the balance between the prior uncertainty in the friction field and the model–observation mismatch, weighted by the observational noise. In the limit

of infinite observational data, the posterior distribution will collapse to a single value. However, in practice when using a finite number of data, the posterior will only change substantially from the prior in directions of the parameter space informed by the available data, which were captured by our low-rank approximation.

## 4 Uncertainty quantification

This study investigated the efficacy of using MFSE to compute the uncertainty in predictions of future mass change from Humboldt Glacier. We defined mass change to be the difference between the final mass[2] of the glacier at $t = 2100$ and its mass at $t = 2007$. While the mass change is a functional of the ice-sheet thickness $H$, for simplicity the following discussion simply refers to the mass change as a scalar function of only the model parameters; that is $f_\alpha(\theta) \in \mathbb{R}$, where $\alpha$ indexes the model fidelity that was used to simulate the ice sheet. Previous UQ studies computed statistics summarizing the uncertainty in ice-sheet predictions, such as mean and variance, using *single-fidelity* Monte Carlo (SFMC) quadrature, that is MC quadrature applied to a single physics model with a fixed numerical discretization $\alpha$, for example Ritz et al. (2015) or Schlegel et al. (2018). However, in this study we used MFSE to reduce the computational cost of quantifying uncertainty. Specific details on the MFSE methods investigated are given in the following subsections.

### 4.1 Single-fidelity Monte Carlo quadrature

SFMC quadrature is a highly versatile procedure that can be used to estimate a wide range of statistics for nearly any function regardless of the number of parameters involved. SFMC can be used to compute the mean $Q^\mu \in \mathbb{R}$ and variance $Q^{\sigma^2} \in \mathbb{R}$ of the Humboldt Glacier mass change predicted by a single model using a three-step procedure. The first step randomly samples $N$ realizations of the model parameters $\Theta = \{\theta^{(n)}\}_{n=1}^N$ from their posterior distribution. The second step simulates the model at each realization of the random variable (basal friction field) and computes the mass change at the final time $f_\alpha^{(n)} = f_\alpha(\theta^{(n)})$. The third step approximates the mean and variance using the following unbiased estimators:

$$\mathbb{E}_\pi\left[f_\alpha\right] \approx Q_\alpha^\mu(\Theta) = N^{-1}\sum_{n=1}^N f_\alpha^{(n)}, \tag{16}$$

$$\mathbb{V}_\pi\left[f_\alpha\right] \approx Q_\alpha^{\sigma^2}(\Theta) = (N-1)^{-1}\sum_{n=1}^N\left(f_\alpha^{(n)} - Q_\alpha^\mu(\Theta)\right)^2, \tag{17}$$

where we use the script $\pi$ on the exact expectation $\mathbb{E}_\pi\left[f_\alpha\right]$ and variance $\mathbb{V}_\pi\left[f_\alpha\right]$ to make clear these statistics are al-

---

[2]In this work, we compute the mass of the glacier considering only the ice above flotation, which is what contributes to sea-level change.

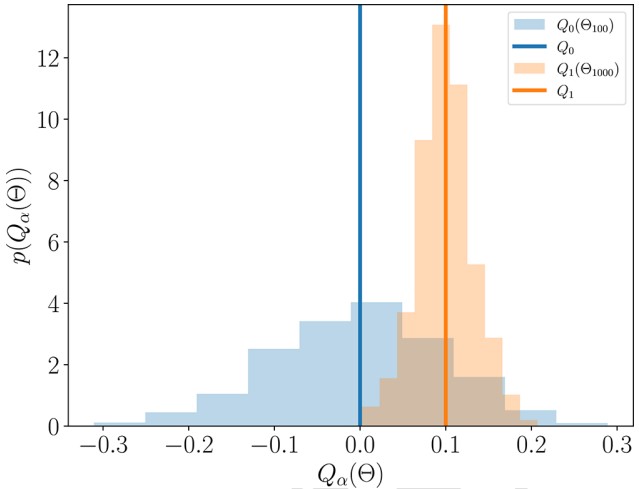

**Figure 4.** The bias–variance trade-off (Eq. 18) of MC estimators of the same computational cost is depicted in blue and orange. The blue line represents the true mean of a computationally expensive model $f_0$, and the orange line represents the mean of a model $f_1$ that is 10 times cheaper but less accurate. The models are only conceptual and not related to the ice-sheet models used in this study and were designed so that evaluating $f_0$ 100 times took the same computational effort as evaluating $f_1$ 1000 times. The histograms were constructed by computing the means of $f_0$ and $f_1$ 1000 times using different realizations of the parameter set $\Theta_N$, where $N$ denotes the number of parameter samples in $\Theta_N$.

ways computed by sampling from the distribution of $\boldsymbol{\theta}$. In our study, we sampled from the posterior distribution of the basal friction parameters; i.e., $\pi(\boldsymbol{\theta}) = p(\boldsymbol{\theta} \mid \mathcal{M}, \boldsymbol{y})$.

MC estimators converge to the true mean and variance of $f_\alpha$ as the number of samples tends to infinity, but using a finite number of samples $N$ introduces an error into the MC estimator that depends on the sample realizations used to compute the estimators. That is, two different realizations of $N$ parameter samples $\Theta$ and the associated QoI values will produce two different mean and variance estimates (see Fig. 4). Consequently, any MC estimator $Q_\alpha(\Theta)$ of an exact statistic $Q$, such as $Q_\alpha^\mu(\Theta)$ and $Q_\alpha^{\sigma^2}(\Theta)$, is a random variable.

The mean-squared error (MSE) is typically used to quantify the error in an MC estimate of a statistic and is given by

$$\mathbb{E}_\Theta\left[(Q_\alpha(\Theta) - Q)^2\right] = \mathbb{E}_\Theta\left[(Q_\alpha(\Theta) - \mathbb{E}_\Theta\left[Q_\alpha(\Theta)\right]\right.$$
$$\left. + \mathbb{E}_\Theta\left[Q_\alpha(\Theta)\right] - Q)^2\right] = \underbrace{\mathbb{V}_\Theta\left[Q_\alpha(\Theta)\right]}_{I} + \underbrace{(\mathbb{E}_\Theta\left[Q_\alpha(\Theta)\right] - Q)^2}_{II}, \tag{18}$$

where $\mathbb{E}_\Theta\left[\cdot\right]$ and $\mathbb{V}_\Theta\left[\cdot\right]$ denote taking the expectation and the variance, respectively, over different realizations of the set of parameter realizations $\Theta$. The MSE of an MC estimator, Eq. (18), consists of two terms referred to as the estimator variance (I) and the estimator bias (II). The bias term of the MSE is caused by using a numerical model, with inadequacy and discretization errors, to compute the mass change. More

specifically, letting $Q_\infty$ denote the exact value of the statistic of a numerical model with zero discretization error but non-zero model inadequacy error and $Q_0$ denote the highest-fidelity computationally tractable model approximation of $Q_\infty$, the bias can then be decomposed into three terms:

$$
\begin{aligned}
(\mathbb{E}\left[Q_\alpha(\Theta)\right] - Q) &= (\mathbb{E}_\Theta\left[Q_\alpha(\Theta)\right] - Q_0 + Q_0 - Q_\infty \\
&\quad + Q_\infty - Q) = (\mathbb{E}_\Theta\left[Q_\alpha(\Theta)\right] - Q_0) \\
&\quad + (Q_0 - Q_\infty) + (Q_\infty - Q).
\end{aligned} \tag{19}
$$

The first term is caused by using a model $f_\alpha$ with numerical discretization that is inferior to that employed by the highest-fidelity model $f_0$. The second term represents the error in the statistic introduced by the numerical discretization of the highest-fidelity model. The third term quantifies the model inadequacy error caused by the numerical model being an approximation of reality. The variance of an MC estimator comes from using a finite number of samples and decreases as the number of samples increases. For example, the variances of the estimators of mean and variance are

$$
\mathbb{V}_\Theta\left[Q_\alpha^\mu(\Theta)\right] = \frac{1}{N}\mathbb{V}_\pi\left[f_\alpha\right] \text{ and}
$$

$$
\mathbb{V}_\Theta\left[Q_\alpha^{\sigma^2}(\Theta)\right] = \frac{1}{N}\left(\frac{2}{(N-1)}\mathbb{V}_\pi\left[f_\alpha\right]^2 \right. \\
\left. + \mathbb{V}_\pi\left[(f_\alpha - \mathbb{E}_\pi\left[f_\alpha\right])^2\right]\right), \tag{20}
$$

respectively, where the variances involving $f_\alpha$ are exact statistics of the model, which are typically unknown. A detailed derivation of the expression for $\mathbb{V}_\pi\left[Q_\alpha^{\sigma^2}(\Theta)\right]$ can be found in Dixon et al. (2023).

Constructing an SFMC estimator of statistics, such as the mean (Eq. 16) or variance (Eq. 17), with a small MSE ensures that the value of the estimator will likely be close to the true value for any set of model parameter samples. However, when using numerical models approximating a physical system, constructing an unbiased estimator of $Q$ is not possible. All models are approximations of reality, and thus the model inadequacy contribution $Q_\infty - Q$ to the bias decomposition in Eq. (19) can never be driven to zero. Additionally, it is impractical to quantify the discretization error $Q_0 - Q_\infty$. Consequently, SFMC methods focus on producing unbiased estimators of $Q_0$ such that $\mathbb{E}_\Theta\left[Q_\alpha(\Theta)\right] = Q_0$.

Unfortunately, even when ignoring inadequacy and discretization errors, constructing an SFMC estimator with a small MSE (Eq. 18) using a computationally expensive high-fidelity model is computationally demanding. The cost is high because the variance term of the MSE of an estimator (Eq. 18) only decreases linearly with the number of samples. In contrast, $N$ can be significantly increased if a cheaper, lower-fidelity model is used, but the corresponding decrease in the estimator variance will be offset by an increase in its bias. Consequently, the bias and variance of any estimator (see Fig. 4) should be balanced, but most SFMC analyses do

not consider this trade-off explicitly when choosing the fidelity of the model used. In the following section we detail how to use MFSE to substantially improve the precision of estimated statistics for a fixed computational cost.

## 4.2 Two-model multi-fidelity uncertainty quantification

MFSE leverages the correlation between models of varying cost and fidelity to reduce the computational cost of constructing MC estimators with a desired MSE. While various multi-fidelity estimators have been developed, this study used approximate control variate (ACV) estimators (implemented in `PyApprox`; Jakeman, 2023), which include most existing estimators, including multi-level Monte Carlo (MLMC) (Giles, 2015) and multi-fidelity Monte Carlo (MFMC) (Peherstorfer et al., 2016), as special cases.[3] In this section, we describe how to construct an ACV estimate of the mean of a model using two models. We then introduce the ACV procedure we used to compute the mean and variance of our highest-fidelity ice-sheet model using an ensemble of 13 models.

Using only high-fidelity model simulations to estimate a statistic with single-fidelity MC produces an unbiased estimator of $Q_0$. However, when the computational cost of running a high-fidelity model limits the number of model simulations that can be used, the variance and thus the MSE of the MC estimator will be large. Fortunately, the MSE of the estimator can be reduced by correcting the high-fidelity estimator with statistics computed using lower-fidelity models. For example, given a high-fidelity model $f_0(\theta)$ and a single low-fidelity model $f_1(\theta)$, an MFMC ACV estimator approximates the mean of the high-fidelity model using

$$
\begin{aligned}
Q_{\text{ACV}}^\mu(\Theta_0, \Theta_1) &= N_0^{-1}\sum_{n=1}^{N_0} f_0(\theta_0^{(n)}) \\
&\quad + \eta\left(N_0^{-1}\sum_{n=1}^{N_0} f_1(\theta_0^{(n)}) - N_1^{-1}\sum_{j=1}^{N_1} f_1(\theta_1^{(j)})\right) \\
&= Q_0^\mu(\Theta_0) + \eta(Q_1^\mu(\Theta_0) - Q_1^\mu(\Theta_1)) \\
&\approx \mathbb{E}_\Theta\left[f_0\right].
\end{aligned} \tag{21}
$$

The two-model ACV estimator in Eq. (21) uses a weighted combination of a high-fidelity MC estimator and two low-fidelity estimators. The high-fidelity model evaluations are used to ensure the ACV estimator is unbiased; i.e., $\mathbb{E}_\Theta\left[Q_{\text{ACV}}^\mu(\Theta_0, \Theta_1)\right] = \mathbb{E}_\pi\left[f_0\right]$, while the low-fidelity evaluations are used to reduce the variance of the estimator. The estimator of the low-fidelity mean $Q_1^\mu(\Theta_0)$ is referred to as a control variate because it is a random variable, which

---

[3]Recently, multilevel best linear unbiased estimators (ML-BLUEs; Schaden and Ullmann, 2020) were developed as an alternative to ACV estimators to estimate the expectation of a high-fidelity model using an ensemble of models of varying cost and fidelity. However, we did not use MLBLUEs in this study because they can only be used to estimate the mass-change mean and not its variance.

is correlated with the random estimator $Q_0^\mu(\Theta_0)$, and can be used to control the variance of that high-fidelity estimator. The term $Q_1^\mu(\Theta_1) \approx Q_1^\mu$ is an approximation of the true low-fidelity statistic $Q_1$ that is used to ensure that the ACV estimator is unbiased, i.e., $\mathbb{E}_\Theta[Q_{\text{ACV}}^\mu(\Theta_0, \Theta_1)] = \mathbb{E}_\Theta[Q_0^\mu(\Theta)] + \eta(\mathbb{E}_\Theta[Q_1^\mu(\Theta_0)] - \mathbb{E}_\Theta[Q_1^\mu(\Theta_1)]) = Q_0^\mu + \eta(Q_1^\mu - Q_1^\mu) = Q_0^\mu$. The weight $\eta$ can be either fixed, e.g., MLMC sets $\eta = -1$, or optimized to minimize the MSE of the estimator. However, an ACV estimator will always be unbiased with respect to $Q_0$ regardless of the value of $\eta$ because the expected values of the second and third terms will always cancel each other out.

Computing the ACV estimate of the high-fidelity mean in Eq. (21) requires two different sets of model evaluations. These evaluations must be obtained by first drawing two sets of samples $\Theta_0 = \{\theta_0^{(n)}\}_{n=1}^{N_0}$, $\Theta_1 = \{\theta_1^{(n)}\}_{n=1}^{N_1}$ from the distribution of the random variables. In our study, we draw random samples from the posterior distribution of the log basal friction, i.e., $p(\theta \mid \mathcal{M}, \dagger)$. The high-fidelity model must be evaluated on all the samples in $\Theta_0$, and the low-fidelity model must be evaluated on both the sets $\Theta_0$ and $\Theta_1$. Typically $N_0 < N_1$. In most practical applications, such as this study, the model $f_0$ used with an ACV estimate is chosen to be the highest-fidelity model that can be simulated $O(10)$ times. However, when a model utilizes a numerical discretization that can be refined indefinitely, MLMC can be used to adaptively set $Q_0$ such that the discretization error $Q_0 - Q_\infty$, in Eq. (19), is equal to the variance $\mathbb{V}_\Theta[Q_{\text{ACV}}]$ of the MLMC estimator. Balancing these two errors ensures that computational effort is not wasted on resolving one source of error more than the other. However, in practice, the geometric complexity of many spatial domains makes generating large numbers of meshes impractical, and estimating the discretization error using techniques like posterior error estimation can be challenging.

The ACV estimator is an unbiased estimator of the mean high-fidelity model. Therefore the MSE (Eq. 18), ignoring the model inadequacy and model discretization errors, is equal to the variance of the estimator, which, when $\Theta_0 \subset \Theta_1$, is TS2

$$\mathbb{V}_\Theta[Q_{\text{ACV}}^\mu(\Theta_0, \Theta_1)] = N_0^{-1} \mathbb{V}_\pi[f_0]\left(1 - \frac{N_1 - N_0}{N_1}\mathbb{C}\text{orr}_\pi[f_0, f_1]^2\right). \quad (22)$$

Thus, for fixed $N_0$ and $N_1$, if the high- and low-fidelity models are highly correlated, the ACV estimator will be much more accurate than the SFMC estimator; see Eq. (20). Moreover, the values of $N_0, N_1$ can be optimized to minimize the error in an ACV estimator given a fixed computational budget. In the following section, we provide more details on how to construct an ACV estimator using more than one low-fidelity model, including information on how to optimize $\eta$ and the number of samples used to evaluate each model.

## 4.3 Many-model multi-fidelity uncertainty quantification

Given an ensemble of $M+1$ models $\{f_\alpha(\theta)\}_{\alpha=0}^M$, an ACV MC estimator can be used to compute a vector-valued estimator $\boldsymbol{Q}_0 = [Q_{0,1}, \cdots, Q_{0,K}]^\top \in \mathbb{R}^K$ of statistics of the highest-fidelity model $f_0$; the specific instances of the ice-sheet models used by this study are presented in Sect. 5.1. The vector $\boldsymbol{Q}_0$ may be comprised of a single type of statistic computed for multiple quantities of interest (QoIs), multiple statistics of a single QoI, or a combination of both. For example, in this study we computed the ACV estimator of the mean and variance of the mass change; that is $\boldsymbol{Q}_0 = [Q_0^\mu, Q_0^{\sigma^2}]^\top \in \mathbb{R}^2$.

Any ACV estimators $\boldsymbol{Q}_{\text{ACV}} = [Q_{\text{ACV}}^\mu, Q_{\text{ACV}}^{\sigma^2}]^\top \in \mathbb{R}^2$ of the vector-valued high-fidelity statistic $\boldsymbol{Q}_0$ can be expressed as

$$\boldsymbol{Q}_{\text{ACV}}(\Theta_0, \Theta_1^*, \Theta_1, \ldots, \Theta_M^*, \Theta_M) = \begin{bmatrix} Q_0^\mu \\ Q_0^{\sigma^2} \end{bmatrix}$$

$$+ \begin{bmatrix} \eta_{1,1} & \cdots & \eta_{1,2M} \\ \eta_{2,1} & \cdots & \eta_{2,2M} \end{bmatrix} \begin{bmatrix} Q_1^\mu(\Theta_1^*) - Q_1^\mu(\Theta_1) \\ Q_1^{\sigma^2}(\Theta_1^*) - Q_1^{\sigma^2}(\Theta_1) \\ \vdots \\ Q_M^\mu(\Theta_M^*) - Q_M^\mu(\Theta_M) \\ Q_M^{\sigma^2}(\Theta_M^*) - Q_M^{\sigma^2}(\Theta_M) \end{bmatrix},$$

where $Q_{\text{m}}^\mu(\Theta_{\text{m}}^*)$ and $Q_{\text{m}}^\mu(\Theta_{\text{m}})$ are single-model MC estimates of the mean, $\mathbb{E}_\Theta[f_{\text{m}}]$ (Eq. 16), computed using the $m$th model, $m = 0, \ldots, M$, and different sample sets $\Theta_{\text{m}}^*$ and $\Theta_{\text{m}}$. Similarly, $Q_{\text{m}}^{\sigma^2}(\Theta_{\text{m}}^*)$ and $Q_{\text{m}}^{\sigma^2}(\Theta_{\text{m}})$ are estimates of the model variance, $\mathbb{V}_\Theta[f_{\text{m}}]$ (Eq. 17), computed using the $m$th model. In more compact notation,

$$\boldsymbol{Q}_{\text{ACV}}(\Theta_{\text{ACV}}) = \boldsymbol{Q}_0(\Theta_0) + \eta\boldsymbol{\Delta}(\Theta_\Delta), \quad (23)$$

where $\Theta_\Delta = \{\Theta_1^*, \Theta_1, \ldots, \Theta_M^*, \Theta_M\}$, $\Theta_{\text{ACV}} = \{\Theta_0, \Theta_\Delta\}$,

$$\boldsymbol{\Delta}(\Theta_\Delta) = \begin{bmatrix} \boldsymbol{\Delta}_1(\Theta_1^*, \Theta_1) \\ \vdots \\ \boldsymbol{\Delta}_M(\Theta_M^*, \Theta_M) \end{bmatrix} \in \mathbb{R}^{2M},$$

$$\boldsymbol{\Delta}_{\text{m}}(\Theta_{\text{m}}^*, \Theta_{\text{m}}) = \begin{bmatrix} Q_{\text{m}}^\mu(\Theta_{\text{m}}^*) - Q_{\text{m}}^\mu(\Theta_{\text{m}}) \\ Q_{\text{m}}^{\sigma^2}(\Theta_{\text{m}}^*) - Q_{\text{m}}^{\sigma^2}(\Theta_{\text{m}}) \end{bmatrix} \in \mathbb{R}^2,$$

$$m = 1, \ldots, M, \quad (24)$$

and the entries of $\eta \in \mathbb{R}^{2 \times 2M}$ are called control variate weights. Formulating the control variate weights as a matrix enables the ACV estimator to exploit the correlation between the statistics $Q^\mu$ and $Q^{\sigma^2}$, producing estimates of these individual statistics with lower mean-squared error (MSE) than would be possible if the two statistics were estimated independently.

A multi-model ACV estimator is constructed by evaluating the highest-fidelity model for a single set of samples $\Theta_0$ and evaluating each low-fidelity model for two sets of samples $\Theta_\alpha^* = \{\theta^{(n)}\}_{n=1}^{N_{\alpha^*}}$ and $\Theta_\alpha = \{\theta^{(n)}\}_{n=1}^{N_\alpha}$. Different ACV estimators can be produced by changing the way each sample set is structured. For example, MFMC estimators sample

the uncertain parameters such that $\Theta_\alpha^* \subset \Theta_\alpha$ and $\Theta_\alpha^* = \Theta_{\alpha-1}$ and MLMC estimators sample such that $\Theta_\alpha^* \cap \Theta_\alpha = \varnothing$ and $\Theta_\alpha^* = \Theta_{\alpha-1}$.

By construction, any ACV estimator is an unbiased estimator of $\boldsymbol{Q}_0$ because $\mathbb{E}_\Theta[\boldsymbol{\Delta}_\alpha] = 0$, with $\alpha > 0$. Consequently, the MSE of the ACV estimator (Eq. 18) can be minimized by optimizing the determinant of the estimator covariance matrix. When estimating a single statistic ($K = 1$), this is equivalent to minimizing the variance of the estimator. Given sample sets $\Theta_{\mathrm{ACV}}$, the determinant of the covariance of an ACV estimator, $\mathbb{C}\mathrm{ov}_\Theta[\boldsymbol{Q}_{\mathrm{ACV}}, \boldsymbol{Q}_{\mathrm{ACV}}]$ in Eq. (26), can be minimized using the optimal weights

$$\boldsymbol{\eta}(\Theta_{\mathrm{ACV}}) = -\mathbb{C}\mathrm{ov}_\Theta[\boldsymbol{Q}_0, \boldsymbol{\Delta}]\mathbb{C}\mathrm{ov}_\Theta[\boldsymbol{\Delta}, \boldsymbol{\Delta}]^{-1},$$
$$\mathbb{C}\mathrm{ov}_\Theta[\boldsymbol{Q}_0, \boldsymbol{\Delta}] \in \mathbb{R}^{2 \times 2M}, \text{ and}$$
$$\mathbb{C}\mathrm{ov}_\Theta[\boldsymbol{\Delta}, \boldsymbol{\Delta}] \in \mathbb{R}^{2M \times 2M}, \tag{25}$$

which produces an ACV estimator with covariance

$$\mathbb{C}\mathrm{ov}_\Theta[\boldsymbol{Q}_{\mathrm{ACV}}, \boldsymbol{Q}_{\mathrm{ACV}}](\Theta_{\mathrm{ACV}}) = \mathbb{C}\mathrm{ov}_\Theta[\boldsymbol{Q}_0, \boldsymbol{Q}_0]$$
$$- \mathbb{C}\mathrm{ov}_\Theta[\boldsymbol{Q}_0, \boldsymbol{\Delta}]\mathbb{C}\mathrm{ov}_\Theta[\boldsymbol{\Delta}, \boldsymbol{\Delta}]^{-1}\mathbb{C}\mathrm{ov}_\Theta[\boldsymbol{Q}_0, \boldsymbol{\Delta}]^\top \in \mathbb{R}^{2 \times 2}, \tag{26}$$

where the dependence of $\boldsymbol{\Delta}$ and $\boldsymbol{Q}_0$ on the sample sets $\Theta_\Delta$ and $\Theta_0$ was dropped to improve readability. Note that, in Eqs. (25) and (26) and the remainder of this paper, we use $\mathbb{C}\mathrm{ov}[\boldsymbol{v}, \boldsymbol{v}]$ as longhand for $\mathbb{V}[\boldsymbol{v}]$ to emphasize that the covariance is a matrix when the random variable $\boldsymbol{v}$ is a vector.

## 4.4 Computational considerations for multi-fidelity uncertainty quantification

The approximation of model statistics using ACV estimators is broken down into two steps. The first step, referred to as the *pilot study* or *exploration phase*, involves collecting evaluations of each model on a common set of samples. These evaluations are used to compute the so-called pilot statistics that are needed to evaluate Eqs. (25) and (26). Subsequently, these pilot statistics are used to find the optimal sample allocation of the best estimator (see Algorithm 1). The second step, known as the *exploitation phase*, involves evaluating each model according to the optimal sample allocation and then computing the model statistics using Eq. (23). We will discuss the important computational aspects of these two phases in the following subsections.

### 4.4.1 Estimating pilot statistics

Computing the covariance of an ACV estimator, $\mathbb{C}\mathrm{ov}_\Theta[\boldsymbol{Q}_{\mathrm{ACV}}, \boldsymbol{Q}_{\mathrm{ACV}}]$ in Eq. (26), requires estimates of the covariance between the estimator discrepancies $\boldsymbol{\Delta}$ (Eq. 24) with each other and the high-fidelity estimator and the covariance of the high-fidelity estimator, i.e., $\mathbb{C}\mathrm{ov}_\Theta[\Delta, \Delta]$ and $\mathbb{C}\mathrm{ov}_\Theta[Q_0, \Delta]$. In practice, these quantities, which we call *pilot statistics*, are unknown and must be estimated with a pilot study. Specifically, following standard practice

(Peherstorfer and Willcox, 2016), we used MC quadrature with $P$, so-called *pilot samples* $\Theta_{\mathrm{pilot}} = \{\theta^{(p)}\}_{p=1}^P$, to compute the pilot statistics. This involves computing the high-fidelity model and all the low-fidelity models for the same set of samples $\Theta_{\mathrm{pilot}}$. For example, we approximated $\mathbb{C}\mathrm{ov}_\pi[f_\alpha, f_\beta]$, which is needed to compute the quantities in Eq. (26), by

$$\mathbb{C}\mathrm{ov}_\pi[f_\alpha, f_\beta] \approx P^{-1}\sum_{p=1}^P \left(f_\alpha(\theta^{(p)}) - Q_\alpha^\mu(\Theta_{\mathrm{pilot}})\right)$$
$$\times \left(f_\beta(\theta^{(p)}) - Q_\beta^\mu(\Theta_{\mathrm{pilot}})\right) \in \mathbb{R}^{2 \times 2}. \tag{27}$$

Please refer to Dixon et al. (2023) to see the additional quantities needed to compute the covariance blocks of $\mathbb{C}\mathrm{ov}_\Theta[\boldsymbol{\Delta}_\alpha, \boldsymbol{\Delta}_\beta]$ and $\mathbb{C}\mathrm{ov}_\Theta[\boldsymbol{Q}_0, \boldsymbol{\Delta}_\alpha]$, which are required to compute the estimator covariance $\mathbb{C}\mathrm{ov}_\Theta[\boldsymbol{Q}_{\mathrm{ACV}}, \boldsymbol{Q}_{\mathrm{ACV}}]$ of a vector-valued statistic that consists of both the mean and the variance of a model. Finally, we recorded the CPU time needed to simulate each model at all pilot samples and set the model costs $\boldsymbol{w}^\top = [w_0, w_1, \ldots, w_M] \in \mathbb{R}^{M+1}$ to be the median simulation time of each model.

Unfortunately, using a finite $P$ introduces sampling errors into $\mathbb{C}\mathrm{ov}_\Theta[\boldsymbol{\Delta}, \boldsymbol{\Delta}]$ and $\mathbb{C}\mathrm{ov}_\Theta[\boldsymbol{Q}_0, \boldsymbol{\Delta}]$, which in turn induces error in the ACV estimator covariance (Eq. 26). This error can be decreased using a large $P$, but this would require additional evaluations of expensive numerical models, which we were trying to avoid. Consequently, in this study we investigated the sensitivity of the error in ACV MC estimators to the number of pilot samples. Results of this study are presented in Sect. 5.

### 4.4.2 Optimal computational resource allocation

The covariance of an ACV estimator, $\mathbb{C}\mathrm{ov}_\Theta[\boldsymbol{Q}_{\mathrm{ACV}}, \boldsymbol{Q}_{\mathrm{ACV}}]$ in Eq. (26), is dependent on how samples are allocated to the sets $\Theta_\alpha$ and $\Theta_\alpha^*$, which we call the sample allocation $\mathcal{A}$. $\mathcal{A}$ uniquely defines the allocation strategy by listing the number of samples of each set $\Theta_\alpha$ and $\Theta_\alpha^*$ and their pairwise intersections. Namely, $\mathcal{A} = \{N_0, N_{\alpha \cap \beta}, N_{\alpha^* \cap \beta}, N_{\alpha \cap \beta^*}, N_{\alpha^* \cap \beta^*} \mid \alpha, \beta = 1, \ldots, M\}$, where $N_{\alpha \cap \beta} = |\Theta_\alpha \cap \Theta_\beta|$, $N_{\alpha^* \cap \beta} = |\Theta_\alpha^* \cap \Theta_\beta|$, $N_{\alpha \cap \beta^*} = |\Theta_\alpha \cap \Theta_\beta^*|$, and $N_{\alpha^* \cap \beta^*} = |\Theta_\alpha^* \cap \Theta_\beta^*|$ denote the number of samples in the intersections of pairs of sets, and $N_{\alpha \cup \beta} = |\Theta_\alpha \cup \Theta_\beta|$, $N_{\alpha^* \cup \beta} = |\Theta_\alpha^* \cup \Theta_\beta|$, $N_{\alpha \cup \beta^*} = |\Theta_\alpha \cup \Theta_\beta^*|$, and $N_{\alpha^* \cup \beta^*} = |\Theta_\alpha^* \cup \Theta_\beta^*|$ denote the number of samples in the union of pairs of sets. Thus, the best ACV estimator can be theoretically found by solving the constrained nonlinear optimization problem:

$$\min_{\mathcal{A} \in \mathbb{A}} \mathrm{Det}[\mathbb{C}\mathrm{ov}_\Theta[\boldsymbol{Q}_{\mathrm{ACV}}, \boldsymbol{Q}_{\mathrm{ACV}}](\mathcal{A})] \quad \text{s.t.} \quad W(\boldsymbol{w}, \mathcal{A}) \leq W_{\max}. \tag{28}$$

In the above equation, $\mathbb{A}$ is the set of all possible sample allocations and the constraint ensures that the computational cost of computing the ACV estimator,

$$W(\boldsymbol{w}, \mathcal{A}) = \sum_{\alpha=0}^M N_{\alpha^* \cup \alpha} w_\alpha, \tag{}$$

is smaller than a computational budget $W_{\max} \in \mathbb{R}$. The solution to this optimization problem is often called the optimal sample allocation.

Unfortunately, a tractable algorithm for solving Eq. (28) has not yet been developed, largely due to the extremely high number of possible sample allocations in the set $\mathbb{A}$. Consequently, various ACV estimators have been derived in the literature that simplify the optimization problem by specifying what we call the sample structure $\mathcal{T}$, which restricts how samples are shared between the sets $\Theta_\alpha$ and $\Theta_\alpha^*$. For example, optimizing the estimator variance (Eq. 22) of a two-model MFMC (Peherstorfer et al., 2016) mean estimator (Eq. 21) requires solving

$$\min_{N_0, N_1} N_0^{-1} \mathbb{V}\left[f_0\right]\left(1 - \frac{N_1 - N_0}{N_1}\mathbb{C}\mathrm{orr}[f_0, f_1]^2\right)$$

s.t. $\quad N_0 w_0 + N_1 w_1 \leq W_{\max}$,

$\mathcal{T} = \{N_{0 \cap 1*} = N_0, N_{0 \cup 1*} = N_0, N_{0 \cap 1} = N_0,$

$\quad N_{0 \cup 1} = N_1, N_{1* \cap 1} = N_0, N_{1* \cup 1} = N_1\}.$

Alternatively, minimizing the estimator variance of the two-model MLMC (Giles, 2015)[4] mean estimator requires solving

$$\min_{N_0, N_1} N_0^{-1} \mathbb{V}\left[f_1 - f_0\right] + (N_1 - N_0)^{-1} \mathbb{V}\left[f_1\right]$$

s.t. $\quad N_0 w_0 + N_1 w_1 \leq W_{\max}$,

$\mathcal{T} = \{N_{0 \cap 1*} = N_0, N_{0 \cup 1*} = N_0, N_{0 \cap 1} = 0,$

$\quad N_{0 \cup 1} = N_1, N_{1* \cap 1} = 0, N_{1* \cup 1} = N_1\}.$

MLMC and MFMC employ sample structures $\mathcal{T}$ that simplify the general expression for the estimator covariance $\mathbb{C}\mathrm{ov}_\Theta\left[\boldsymbol{Q}_{\mathrm{ACV}}, \boldsymbol{Q}_{\mathrm{ACV}}\right]$ in Eq. (26). These simplifications were used to analytically derive solutions of the sample allocation optimization problem in Eq. (28) when estimating the mean, $\mathbb{E}_\Theta\left[f_0\right]$ in Eq. (16), for a scalar-valued model. However, the optimal sample allocation of MLMC and MFMC must be computed numerically when estimating other statistics, such as variance $\mathbb{V}_\Theta\left[f_0\right]$ in Eq. (17). Similarly, numerical optimization must be used to optimize the estimator covariance, $\mathbb{C}\mathrm{ov}_\Theta\left[\boldsymbol{Q}_{\mathrm{ACV}}, \boldsymbol{Q}_{\mathrm{ACV}}\right]$ in Eq. (26), of most other ACV estimators, including the ACVMF and ACVIS (Gorodetsky et al., 2020), as well as their tunable generalizations (Bomarito et al., 2022).

Each existing ACV estimator was developed to exploit alternative sample structures $\mathcal{T}$ to improve the performance of ACV estimators in different settings. For example, a three-model ACVMF estimator performs well when the low-fidelity models are conditionally independent of the high-fidelity model. Imposing this conditional independence is useful when knowing one low-fidelity model does not provide any additional information about the second low-fidelity

model, given enough samples of the high-fidelity model. This situation can arise when the low-fidelity models use different physics simplifications of the high-fidelity model. In contrast, MLMC assumes that each model in the hierarchy is conditionally independent of all other models given the next higher-fidelity model. This allows MLMC to perform well with a set of models ordered in a hierarchy by bias relative to the exact solution of the governing equations.

The performance of different ACV estimators is problem dependent. Consequently, in this paper we investigated the use of a large number of different ACV estimators from the literature. For each estimator we used the general-purpose numerical optimization algorithm proposed in Bomarito et al. (2022) to find the optimal sample allocations that minimize the determinant of the estimator covariance.[5]

### 4.4.3 Model and estimator selection

Using data from all available models may produce an estimator with larger MSE than an estimator that is only constructed using a subset of the available models. This occurs when a subset of the low-fidelity models correlate much better with the high-fidelity model than the rest of the low-fidelity models. For instance, some low-fidelity models may fail to capture physical behaviors that are important to estimating the QoI. Consequently, it is difficult to determine the best estimator a priori. However, we can accurately predict the relative performance of any ACV estimator using only the model simulations run during the pilot study. Thus, in this study we enumerated a large set of estimator types encoded by the different sample structures $\mathcal{T}$ and model subsets.

Algorithm 1 summarizes the procedure we use to choose the best ACV estimator. Line 8 loops over all model subsets $\mathcal{S}$. In this study, we enumerated all permutations of the sets of models that contained the high-fidelity model and at most three low-fidelity models. Line 10 enumerates each parametrically defined estimator $E$. We enumerated the large sets of parametrically defined generalized multi-fidelity (GMF), generalized independent sample (GIS), and generalized recursive difference (GRD) ACV estimators introduced by Bomarito et al. (2022). These sets of estimators include ACVMF, MFMC, and MLMC (with optimized control variate weights) as special cases. For each estimator $E$ and model subset $\mathcal{S}$, line 12 was used to find the optimal sample allocation $\mathcal{A}_E$, using the pilot values $\{f_\alpha(\Theta_{\mathrm{pilot}})\}_{\alpha \in \mathcal{S}}$ when minimizing Eq. (28). Lines 13–16 were used to record the best estimator found at each iteration of the outer loops.

Whether a model is useful for reducing the MSE of a multi-fidelity estimator depends on the correlations between that model, the high-fidelity model, and the other low-fidelity models. For toy parameterized PDE problems, such

---

[4]MLMC estimators set all the control variate weights in Eq. (23) to $\eta = -1$. Refer to Gorodetsky et al. (2020) for more details on the connections between ACV and MLMC estimators.

[5]The presentation of the optimization algorithms in Bomarito et al. (2022) focuses on the estimation of a single statistic but can trivially be extended to the vector-valued QoIs considered here.

---

**Algorithm 1** Estimator selection.

---

1: **Input**
2: $\{f_\alpha(\Theta_{\text{pilot}})\}_{\alpha=0}^{M}$     Pilot evaluations of each model
3: **Output**
4: $\mathcal{A}_{\text{best}}$             Best estimator sample allocation
5: $J_{\text{best}}$             Best estimator objective
6: $J_{\text{best}} \leftarrow \infty$
7: ▷ Loop over all low-fidelity model subsets
8: **for** $\mathcal{S} \subseteq \{1, \ldots, M\}$ **do**
9:     ▷ Loop over all MF estimators, e.g., MLMC, MFMF, ACVMF.
10:     **for** $E \in \mathcal{E}$ **do**
11:        ▷ Compute the optimal estimator objective $J_E$ and sample allocation $\mathcal{A}_E$ for the current estimator and subset of models
12:        $J_E, \mathcal{A}_E \leftarrow$ Solve Eq. (28) using $\mathbb{A}_{E,\mathcal{S}}$ and $\{f_\alpha(\Theta_{\text{pilot}})\}_{\alpha \in \mathcal{S}}$
13:        **if** $J_E < J_{\text{best}}$ **then**
14:           ▷ Update the best estimator
15:           $\mathcal{A}_{\text{best}} \leftarrow \mathcal{A}_E$
16:           $J_{\text{best}} \leftarrow J_E$
17:        **end if**
18:     **end for**
19: **end for**

---

as the diffusion equation with an uncertain diffusion coefficient, theoretical convergence rates and theoretical estimates of computational costs can be used to rank models. However, for the models we used in this study, and likely many other ice-sheet studies, ordering models hierarchically, that is, by bias or correlation relative to the highest-fidelity model, before evaluating them is challenging. Indeed, the best model ensemble for multi-fidelity UQ may not be hierarchical (see Gorodetsky et al., 2020). However, estimators such as MLMC and MFMC only work well on model hierarchies. Consequently, having a practical approach for learning the best model ensemble is needed. Yet to date this issue has received little attention in the multi-fidelity literature. Section 5 provides a discussion of the impact of the pilot study on model selection and the MSE of multi-fidelity estimators.

## 5 Results

This section presents the results of our MFSE study. First, we describe the ensemble of numerical models we used to solve the governing equations presented in Sect. 5.1. Second, we summarize the results of our Bayesian model calibration. Third, in Sect. 5.3 we present the results of our pilot study. Specifically, we compare the computational costs of each model and their SFMC-based estimates of the mean and variance of the mass change computed using the pilot samples. We also report the MSE of ACV estimators predicted using the pilot and note the subset of models they employed. Fourth, we detail the impact of increasing the number of pilot samples on the predicted MSE of the ACV estimators in Sect. 5.4. Finally, we quantify the improvement in the precision of MFSE estimates of mass-change statistics relative to SFMC in Sect. 5.5. All results were generated with the `PyApprox` software package (Jakeman, 2023).

### 5.1 Multi-fidelity model ensemble

In this study we investigated the use of 13 different models of varying computational cost and fidelity to compute glacier mass change. Specifically, we used MFSE to estimate the mean and variance of a highly resolved finite-element model using an ensemble of 12 low-fidelity models. We compactly denote the fidelity of each model using the notation $\text{PHYSICSNAME}_{\text{dx,dt}}$, where PHYSICSNAME refers to the governing equations solved, dx denotes the size of the representative spatial element, and dt denotes the size of the time step. The four different meshes we used are shown in Fig. 2.

The highest-fidelity model we considered in this study was a MOLHO-based model denoted by $\text{MOLHO}^*_{1\,\text{km},9\,\text{d}}$, where the star indicates that the model was modified to conserve mass. The low-fidelity model ensemble consisted of four MOLHO-based low-fidelity models, $\text{MOLHO}_{1\,\text{km},36\,\text{d}}$, $\text{MOLHO}_{1.5\,\text{km},36\,\text{d}}$, $\text{MOLHO}_{2\,\text{km},36\,\text{d}}$, and $\text{MOLHO}_{3\,\text{km},36\,\text{d}}$, and eight SSA-based low-fidelity models, $\text{SSA}_{1\,\text{km},36\,\text{d}}$, $\text{SSA}_{1.5\,\text{km},36\,\text{d}}$, $\text{SSA}_{2\,\text{km},36\,\text{d}}$, $\text{SSA}_{3\,\text{km},36\,\text{d}}$, $\text{SSA}_{1\,\text{km},365\,\text{d}}$, $\text{SSA}_{1.5\,\text{km},365\,\text{d}}$, $\text{SSA}_{2\,\text{km},365\,\text{d}}$, and $\text{SSA}_{3\,\text{km},365\,\text{d}}$. The number of elements associated with the four meshes with characteristic element sizes 1, 1.5, 2, and 3 km were 22 334, 13 744, 9238, 2611, and 13 744, respectively. The number of nodes for the same four meshes were 1422, 4846, 7154, and 11 536. Note that no low-fidelity model enforced the conservation of mass.

The models we used were all different numerical discretizations of two distinct physics models. However, in the future we could also use alternative classes of low-fidelity models if they become available. For example, we could use linearizations of the parameter-to-QoI map, proposed by Recinos et al. (2023), if our MOLHO and/or SSA codes become capable of efficiently computing the gradient of the map. Such an approach would require only one nonlinear forward transient solve of the governing equations followed by a linear solve of the corresponding backward adjoint. Once constructed, the linearized map could then be evaluated very cheaply and used to reduce the estimator variance, $\mathbb{C}\text{ov}_\Theta\left[\boldsymbol{Q}_{\text{ACV}}, \boldsymbol{Q}_{\text{ACV}}\right]$ in Eq. (26), of the MFSE estimators, provided the error introduced by the linearization is not substantial. Other types of surrogates could also be used in principle; however, the large number of parameters used poses significant challenges to traditional methods such as the Gaussian processes used in Jantre et al. (2024). Recently developed machine-learning surrogates (Jouvet et al., 2021; Brinkerhoff et al., 2021; He et al., 2023) could be competitive alternatives to the low-fidelity models considered in this work.

Lastly, note that we used a different model to the 13 described above for the Bayesian calibration of the basal friction parameters. Specifically, we used the C++ code `MALI` (Hoffman et al., 2018), which can solve the Blatter–Pattyn equations (Pattyn, 2003; Dukowicz et al., 2010) and compute the action of the Hessian on a vector. `MALI` efficiently computed these Hessian-vector products, needed to compute our Laplace approximation of the posterior in Eq. (14), by solving the adjoint equations for the steady-state Blatter–Pattyn equations. However, SSA equations (Sect. 2.1.3) are not currently implemented in `MALI` and the MOLHO (Sect. 2.1.2) equations have only recently been implemented (after the simulations for this work were performed). Consequently, we used `FEniCS` (Alnæs et al., 2015) to implement both MOLHO and SSA to ensure that the relative computational timings of these models would be consistent. Solving the MOLHO model using the C++-based `MALI` code and solving the SSA using the Python-based `FEniCS` would have corrupted the MFSE results. Moreover, implementing SSA in `MALI` would be time-consuming because it is currently only used to solve 3D models and not 2D models, such as SSA. Indeed, motivation for this study was partially to determine the utility of implementing the SSA equations in `MALI`.

## 5.2 Bayesian model calibration

In this study we used the `MALI` ice-sheet code (Hoffman et al., 2018; Tezaur et al., 2021) to calibrate the basal friction field on the finest mesh, as described in Sect. 3. The MAP point of the posterior, determined using Eq. (13), is depicted in the left panel of Fig. 5. The pointwise variance of the Laplace approximation of the posterior of the log friction (i.e., the diagonal of Eq. 15) is depicted in the right panel of Fig. 5. When this figure is compared to the pointwise variance of the prior (i.e., the diagonal of Eq. 9) depicted in the center panel of Fig. 5, it is clear that conditioning the prior uncertainty in the surface velocity significantly reduced the uncertainty in the basal friction field. This conclusion is further corroborated by Fig. 6, which compares a random sample from the prior and a random sample from the posterior. The minimum and maximum values of the posterior sample of the log friction are much smaller than the same bounds of the prior sample. However, the posterior sample has much higher frequency content because the data only informed the lower-frequency modes of the friction field.

To demonstrate a projection to 2100 using a calibrated model, Fig. 7 depicts the difference between the final (year 2100) and initial (year 2007) ice thickness and the final surface velocity of Humboldt Glacier computed by the highest-fidelity model ($\text{MOLHO}^*_{1\,\text{km},9\,\text{d}}$) for a random posterior realization of the basal friction field. The final ice thickness shown differs substantially from the initial thickness, with thickness decreasing substantially at lower elevations of the glacier in the ablation zone where increasingly negative surface mass balance occurs through 2100. In general, the glacier speeds up as negative surface mass balance causes the surface to steepen near the terminus. The largest speedup occurs in the region of fast flow in the north where basal friction is small. Also note that the high-frequency differences present in the thickness difference were due to the high-frequency oscillations in the posterior sample; see Fig. 6.

## Prior sensitivity

In this study we used our domain experience to determine the best values of the prior hyper-parameters $\gamma$, $\delta$, and $\eta$ reported in Sect. 2.3 and the likelihood hyper-parameter $\xi$ in Eq. (12). However, varying these hyper-parameters would likely change the estimates of uncertainty in ice-sheet predictions produced by this study. Similarly to previous studies (Isaac et al., 2015), we did not investigate these sensitivities extensively. We heuristically chose the prior hyper-parameters so that the prior samples would have a variance and spatial variability that we deemed in line with our domain experience. Further, we found that reducing $\xi$ substantially from the value we ultimately used while keeping the prior hyper-parameters fixed prevented the MAP point from capturing the high-frequency content of the basal friction field needed to accurately match the observed surface velocities. Future studies should investigate the sensitivity of mass change to the values of the hyper-parameters more rigorously using an approach such as the one developed by Recinos et al. (2023).

## Interpolating basal friction

In this study we drew samples from the posterior distribution of the friction parameters defined on the finest spatial mesh. However, a posterior sample defined on the fine mesh cannot be used to predict mass change with a low-fidelity model defined on a coarser mesh. Consequently, before using a low-fidelity model with a coarse mesh to predict mass change, we first interpolated each sample of the posterior distribution of the basal friction field defined on the finest mesh onto the mesh used by the low-fidelity model. Specifically, we used the linear finite-element basis of the fine mesh to interpolate onto the coarser meshes. This procedure ensured that varying the basal friction field (the random parameters of the model) would affect each model similarly, regardless of the mesh discretization employed. However, the linear interpolation we used may have overly smoothed the friction on coarse meshes relative to alternative higher-order interpolation methods. Consequently, using alternative interpolation methods may increase the correlation between the mass loss predicted by the coarse meshes and that predicted using the finest mesh. However, we did not explore the use of alternative interpolation schemes because our results demonstrate that linear interpolation still produces models that can be used to produce a computationally efficient MFSE.

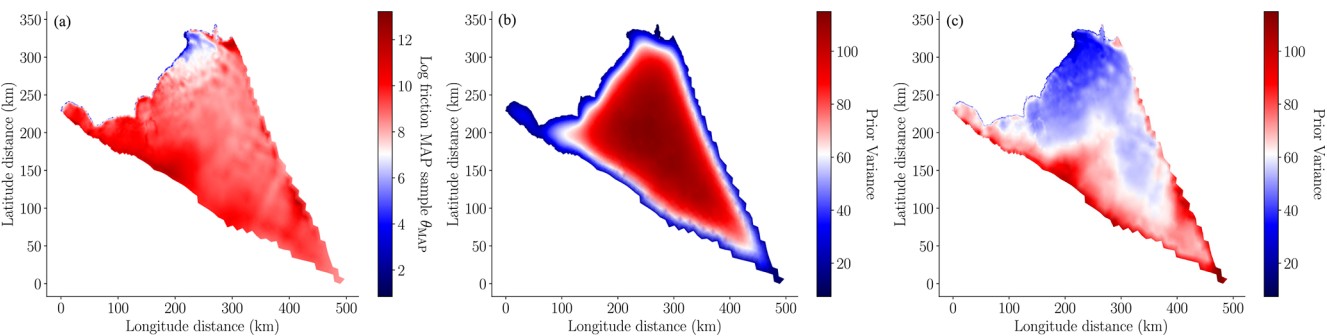

**Figure 5. (a)** Log of the basal friction MAP point, $\theta_{\mathrm{MAP}}$, computed using Eq. (13). **(b)** Pointwise prior variance, i.e., the diagonal entries of $\Sigma_{\mathrm{prior}}$, defined in Eq. (9). **(c)** Pointwise posterior variance, i.e., the diagonal entries of $\Sigma_{\mathrm{post}}$, defined in Eq. (15). Note that the color scales of each plot span different ranges so that the variability in the quantities plotted is visible.

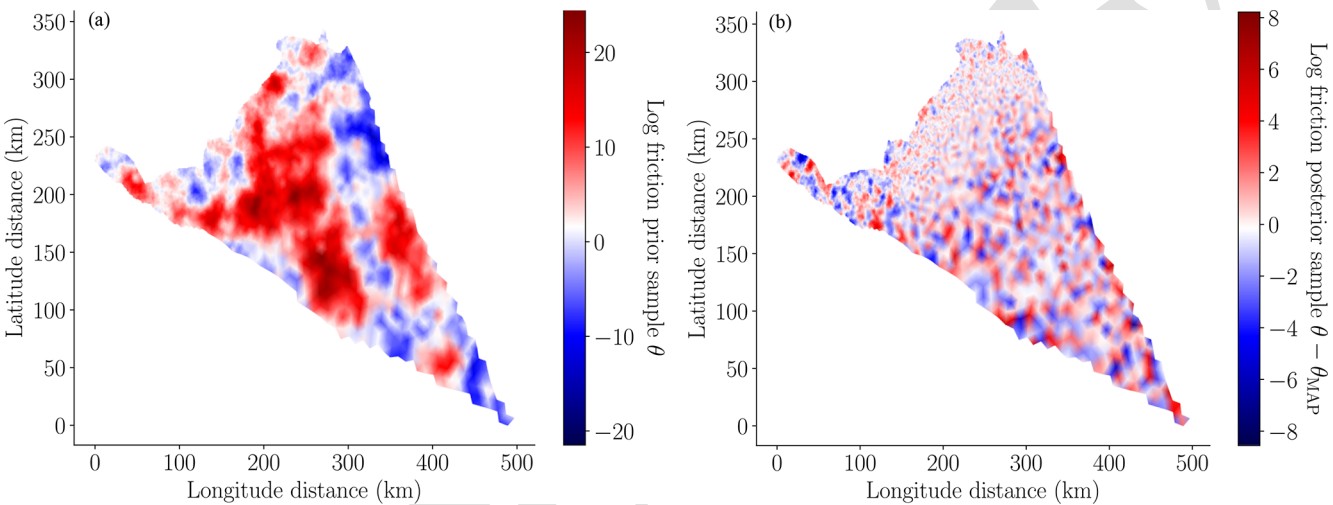

**Figure 6. (a)** A random sample from the prior distribution of the log friction $p(\theta) \sim \mathcal{N}(\mathbf{0}, \Sigma_{\mathrm{prior}})$, where $\Sigma_{\mathrm{prior}}$ is defined in Eq. (9). **(b)** A random sample from the Laplace approximation of the posterior $p(\theta \mid \mathcal{M}, \mathbf{y}) \sim \mathcal{N}(\theta_{\mathrm{MAP}}, \Sigma_{\mathrm{post}})$, defined in Eq. (14). Note that the color scales of each plot span different ranges so that the variability in the quantities plotted is visible.

## 5.3 Initial pilot study

This section details the results of the pilot study that we used to obtain the computational cost, $\boldsymbol{w}$, of each model and the pilot statistics, e.g., Eq. (27), needed to construct ACV estimators. First, we evaluated each of our 13 models for the same 20 random pilot samples of the model parameters $\Theta_{\mathrm{pilot}}$, i.e., 20 different basal friction fields drawn from the Laplace approximation of the posterior distribution $p(\theta \mid \mathcal{M}, \mathbf{y})$ (Eq. 14). Second we computed the median computational cost (wall time), $\boldsymbol{w}$, required to solve each model for one pilot sample. The median computational costs are plotted in the top panel of Fig. 8, and the total cost of evaluating all 13 models was approximately 144 h. Third, using the pilot samples, we computed the SFMC estimators of the mean, Eq. (16), and standard deviation, using the square root of Eq. (17), of the mass change predicted by each of the 13 models. The middle and lower panels of Fig. 8 show that the means and standard deviations of each model differ. However

in the next section, we show that despite the differences between the statistics computed using each model and the differences between the ice evolution predicted by each model (see Fig. 14), MFSE was able to effectively increase the precision of the mean and variance of the mass change relative to SFMC.

The exact gain in performance achieved by MFSE is dependent on the correlations between each model and the other pilot quantities needed to compute $\mathbb{C}\mathrm{ov}_\Theta[\Delta, \Delta]$ and $\mathbb{C}\mathrm{ov}_\Theta[Q_0, \Delta]$ in Eq. (26). Consequently, in Fig. 9 we plot the entries of the correlation matrix, $\mathbb{C}\mathrm{orr}_\pi[\boldsymbol{f}, \boldsymbol{f}]$ with $\boldsymbol{f} = [f_0, \ldots, f_M]^\top$. This figure shows that, despite the differences between each model's prediction of ice thickness and velocities at the final time (see Fig. 14), as well as the variations in the SFMC estimate of the mean and variance computed using each model, the correlation between each model's prediction

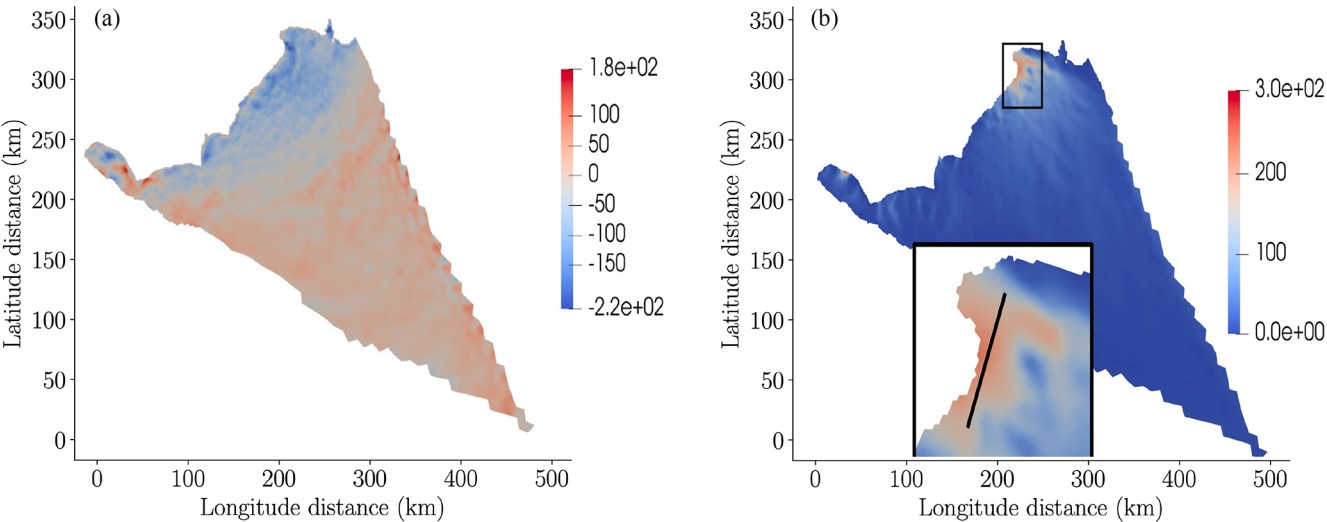

**Figure 7. (a)** The difference between the final and initial ice thickness in meters and **(b)** the surface velocity of Humboldt Glacier. The black inset at the bottom of panel **(b)** is a zoomed-in picture of the top-right tip of the glacier. The black line in the inset was used to plot cross-sections of the thickness and friction profiles in 2100 in a region with high velocities (see Fig. 14). Both **(a)** and **(b)** were generated using the highest-fidelity model $\text{MOLHO}^*_{1\,\text{km},9\,\text{d}}$ evaluated for one random realization of the posterior of the basal friction field.

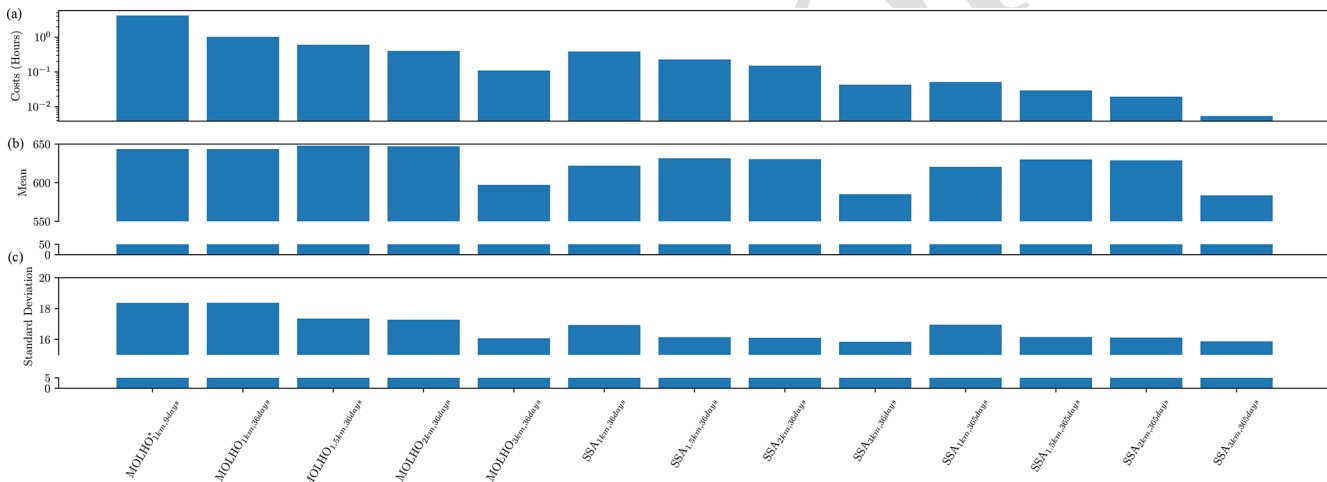

**Figure 8. (a)** The median computational cost $\boldsymbol{w}$ (wall time in hours) of simulating each model used in this study for one realization of the random parameters. **(b)** The mean mass loss – negative expected mass change (metric gigatons) – in 2100. **(c)** The standard deviation of the mass change (metric gigatons) in 2100. Each quantity was computed using 20 pilot samples.

of the mass change is high.[6] However, inspecting the correlation between models can only qualitatively suggest the relative utility of each model for reducing the error in an MFSE estimator. Thus, to be more precise, we used Eq. (28) and our pilot statistics to predict the determinant of the ACV estimator covariance, $\text{Det}\left[\mathbb{C}\text{ov}_\Theta\left[\boldsymbol{Q}_{\text{ACV}},\boldsymbol{Q}_{\text{ACV}}\right]\right]$. Specifically, we made these predictions assuming that a budget of 160 high-fidelity model evaluations would be allocated to the high-

and low-fidelity models. Moreover, this cost was assumed to be additional to the computational cost of simulating each model at the pilot samples. We then computed the so-called *variance reductions* of the ACV estimator,

$$\mathcal{R}_\Theta[Q^\mu_{\text{ACV}}] = \mathbb{V}_\Theta\left[Q^\mu_0\right]/\mathbb{V}_\Theta\left[Q^\mu_{\text{ACV}}\right] \quad \text{and}$$
$$\mathcal{R}_\Theta[Q^{\sigma^2}_{\text{ACV}}] = \mathbb{V}_\Theta\left[Q^{\sigma^2}_0\right]/\mathbb{V}_\Theta\left[Q^{\sigma^2}_{\text{ACV}}\right], \tag{29}$$

by extracting the diagonal elements of the estimator covariance, $\mathbb{C}\text{ov}_\Theta\left[\boldsymbol{Q}_{\text{ACV}},\boldsymbol{Q}_{\text{ACV}}\right]$ in Eq. (26). To ensure a fair comparison, we compared the ACV estimator variance to the SFMC estimator variance obtained using a computational

---

[6]The correlation between $\text{MOLHO}^*_{1\,\text{km},9\,\text{d}}$ and $\text{MOLHO}_{1\,\text{km},36\,\text{d}}$, reported in Fig. 9, was not exactly 1. Each correlation was rounded to four significant digits.

budget equivalent to 160 high-fidelity evaluations plus the computational cost of collecting the pilot model evaluations.

The existing literature assumes that the pilot statistics used with Eq. (28) are exact; however using a small number of pi-5 lot samples can introduce error into the estimator covariance $\mathbb{C}\mathrm{ov}_\Theta \left[ \boldsymbol{Q}_{\mathrm{ACV}}, \boldsymbol{Q}_{\mathrm{ACV}} \right]$. Moreover, we found that the error introduced by using a small number of pilot samples can be substantial, yet it is typically ignored in the existing literature. Consequently, in Fig. 10a we plot the variance reduc-10 tion of the ACV estimators of the mean, $Q_{\mathrm{ACV}}^\mu$, and variance, $Q_{\mathrm{ACV}}^{\sigma^2}$, of mass loss for 21 different bootstraps of the 20 pilot samples (1 bootstrap was just the original pilot data, and each bootstrap set contained 20 samples). The plot is created by randomly sampling the model evaluations with re-15 placement, computing the pilot statistics with those samples, and solving Eq. (28). Please note that, while we enumerate over numerous estimators, each with a different variance reduction, the variability in the plots is induced entirely by the bootstrapping procedure we employed. The box plots report 20 the largest variance reduction, across all estimators, for each bootstrapped sample.

The median variance reduction was over 40 for the ACV estimators of both the mean and the variance of the mass change. In other words, our initial pilot study predicted that 25 using ACV estimators would reduce the cost of estimating uncertainty in projections of the mass change by over a factor of 40 when compared to SFMC estimators, which only use the highest-fidelity model. However, the box plots in Fig. 10a highlight that using only 20 samples introduces a large de-30 gree of uncertainty into the estimated variance reduction. The 10 % quantile of the variance reduction for both the mean and the variance estimators was close to 30.

The estimators obtained by bootstrapping the initial 20 pilot samples not only had different estimator variances (see 35 Fig. 10a), but also predicted that different model subsets (combinations of models) are needed to minimize the estimator variance. Figure 10b plots the model subsets chosen by the bootstrapped estimators and the number of times (frequency) each subset was chosen; the set (0, 9, 10, 12) was 40 chosen when the original 20 pilot samples were used (bootstrapping was not used). Moreover, bootstrapping the estimators also revealed that not all models are equally useful when reducing the variance of the ACV estimator. In some cases, using three models was more effective than us-45 ing four models. Specifically, only 8 out of the 13 models considered were chosen at least once by a bootstrapped estimator. The models $\mathrm{MOLHO}_{1.5\,\mathrm{km},36\,\mathrm{d}}$, $\mathrm{MOLHO}_{2\,\mathrm{km},36\,\mathrm{d}}$, $\mathrm{SSA}_{1\,\mathrm{km},36\,\mathrm{d}}$, $\mathrm{SSA}_{1.5\,\mathrm{km},36\,\mathrm{d}}$, and $\mathrm{SSA}_{2\,\mathrm{km},36\,\mathrm{d}}$ were never selected by any of the bootstrapped estimators. Moreover, in 50 some cases only two low-fidelity models were chosen and in other cases three low-fidelity models were chosen. Lastly, not only did the chosen model subsets vary between bootstrapped estimators, but also the type of estimator chosen varied. In 7 cases, a hierarchical relationship was identified,

and in the other 14 cases, a non-hierarchical relationship was 55 identified; a non-hierarchical estimator was chosen using the original 20 pilot samples (the 21st estimator). Recall that a model ensemble is hierarchical if it can be ordered by bias or correlation relative to the highest-fidelity model and each low-fidelity is only used to reduce the variance of the estima-60 tor of the next higher-fidelity model in a recursive fashion.

## 5.4  Secondary pilot study

Upon quantifying the impact of only using 20 pilot samples on the estimator covariance, $\mathbb{C}\mathrm{ov}_\Theta \left[ \boldsymbol{Q}_{\mathrm{ACV}}, \boldsymbol{Q}_{\mathrm{ACV}} \right]$, and the model subsets, $\mathcal{S}$, chosen by Algorithm 1, we increased the 65 number of pilot samples we used to compute the performance of the ACV estimators. To avoid wasting computational resources in our secondary pilot study, we only evaluated the eight models selected by at least one bootstrapped estimator on an additional 10 pilot samples. The combined cost 70 of the initial and secondary pilot study was approximately 197 h, which equated to the equivalent of approximately 47 simulations of the highest-fidelity model. Note that only the models included in the second pilot were simulated 30 times. The models only included in the first pilot were simulated 20 75 times.

Figure 11c plots the variance reductions in the mean and variance of mass loss, given by $\mathcal{R}[Q_{\mathrm{ACV}}^\mu]$ and $\mathcal{R}[Q_{\mathrm{ACV}}^{\sigma^2}]$, respectively, as the maximum number of models used by the ACV estimators is increased. Note that an estimator allowed 80 to choose four models may still choose fewer than four models, which will happen when some of those models are not highly informative. Of the final 30 pilot samples, 21 different bootstraps were used to quantify the error in the variance reductions caused by only using a small number of pilot sam-85 ples. Comparing Fig. 11c with Fig. 10a, which plots variance reductions using only 20 pilot samples, we observed that increasing the number of pilot samples decreased the variability in the estimator variances. However, increasing the number of pilot samples also increased the computational cost of 90 the pilot study, which in turn reduced the reported median variance reduction. That is, the median variance reductions obtained using 30 pilot samples (Fig. 11c) were lower than the median variance reductions reported using 20 pilot samples (Fig. 10a). This fact can be explained by recalling that 95 the SFMC estimator variance, e.g., $\mathbb{V}_\Theta \left[ Q_0^\mu \right]$, was obtained using a computational budget equivalent to 160 high-fidelity evaluations plus the computational cost of collecting the pilot model evaluations. In contrast, the ACV estimator variance, e.g., $\mathbb{V}_\Theta \left[ Q_{\mathrm{ACV}}^\mu \right]$, does not depend on the number of pi-100 lot samples. Therefore, while increasing the number of pilot samples decreases the SFMC estimator variance, it does not decrease the ACV estimator variance. Consequently, increasing the pilot cost reduces the variance reduction achieved by the ACV estimator. 105

The median variance reduction decreased because ACV estimators utilize the pilot samples solely to compute pilot

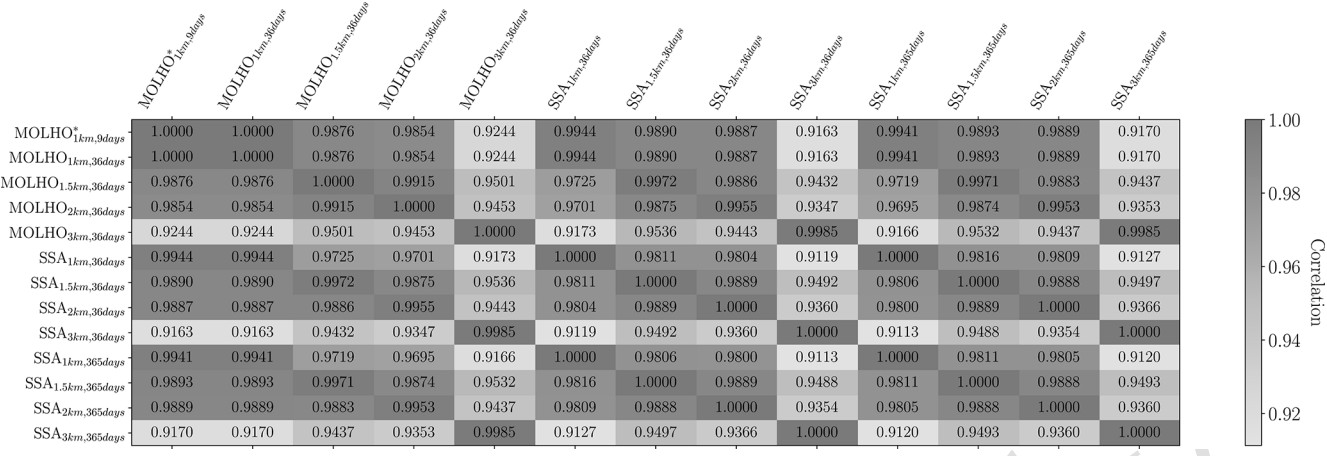

**Figure 9.** The correlations, $\mathbb{C}\mathrm{orr}_\pi[\boldsymbol{f}, \boldsymbol{f}]$ with $\boldsymbol{f} = [f_0, \ldots, f_M]^\top$, between the 13 ice-sheet models considered by this study using 20 pilot samples.

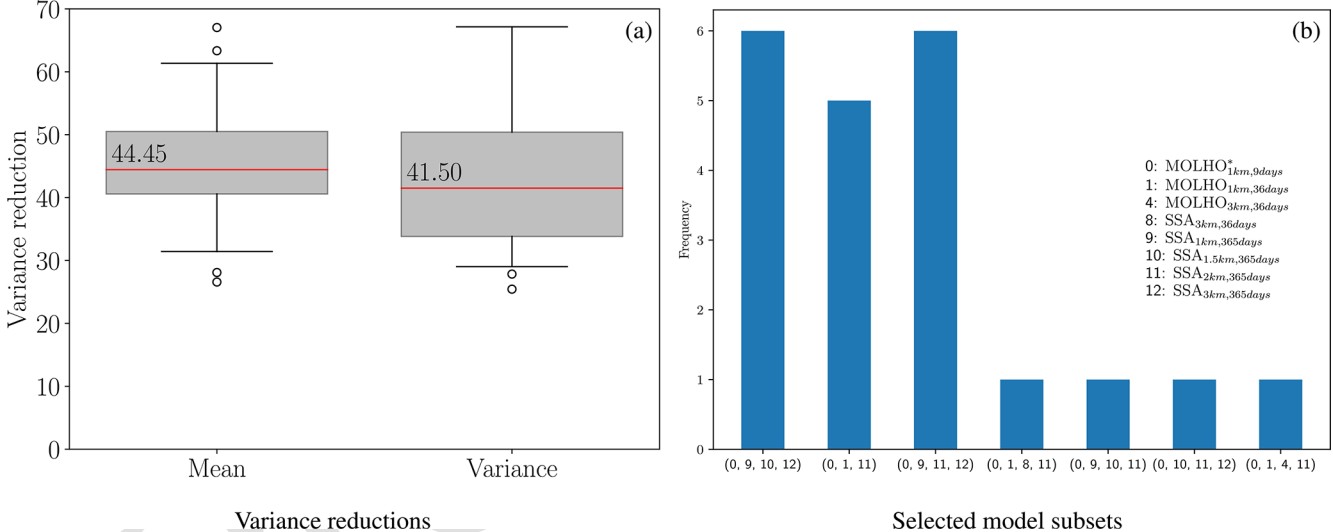

**Figure 10.** **(a)** The predicted variance reductions $\mathcal{R}_\Theta[Q^\mu_{\mathrm{ACV}}]$ (mean) and $\mathcal{R}_\Theta[Q^{\sigma^2}_{\mathrm{ACV}}]$ (variance) (see Eq. 29), obtained using bootstrapping of the initial 20 pilot samples. The red lines represent the median estimator variance reductions. The lower and upper whiskers represent the 10 % and 90 % quantiles. Note that two outliers, with values 73 and 125, do not appear on the plot for $\mathcal{R}[Q^{\sigma^2}_{\mathrm{ACV}}]$. **(b)** The model subsets chosen by the bootstrapped estimators using the initial 20 pilot samples.

statistics, such as variance, and do not reuse these samples for calculating the final statistics. In contrast, an equivalent SFMC estimator can leverage both the pilot and the exploitation budgets to estimate the final statistics. In other words, the variance of an SFMC estimator decreases linearly with the number of pilot samples, whereas the variance of an ACV estimator does not exhibit the same behavior. Specifically, the variance of an ACV estimator is only marginally affected by an increase in the number of pilot samples, as the sample allocation becomes more optimal.

While increasing the number of pilot samples decreased variability, we believed that the benefit of further increasing the number of pilot samples would be outweighed by the resulting drop in the variance reduction. Despite the remaining variability in the variance reduction, we were able to confidently conclude that the MSE of the final ACV estimator we would construct would be much smaller than the MSE of an SFMC estimator of the same cost because even the smallest variance reduction was greater than 14. Consequently, we used the 30 unaltered pilot samples to determine the ACV estimator and its optimal sample allocation, which we used to construct our final estimates of the mean and variance of the mass change. The best estimator chosen was an MFMC estimator that used the three models $\mathrm{MOLHO}^*_{1\,\mathrm{km},9\,\mathrm{d}}$, $\mathrm{MOLHO}_{1\,\mathrm{km},36\,\mathrm{d}}$, and $\mathrm{SSA}_{1.5\,\mathrm{km},365\,\mathrm{d}}$.

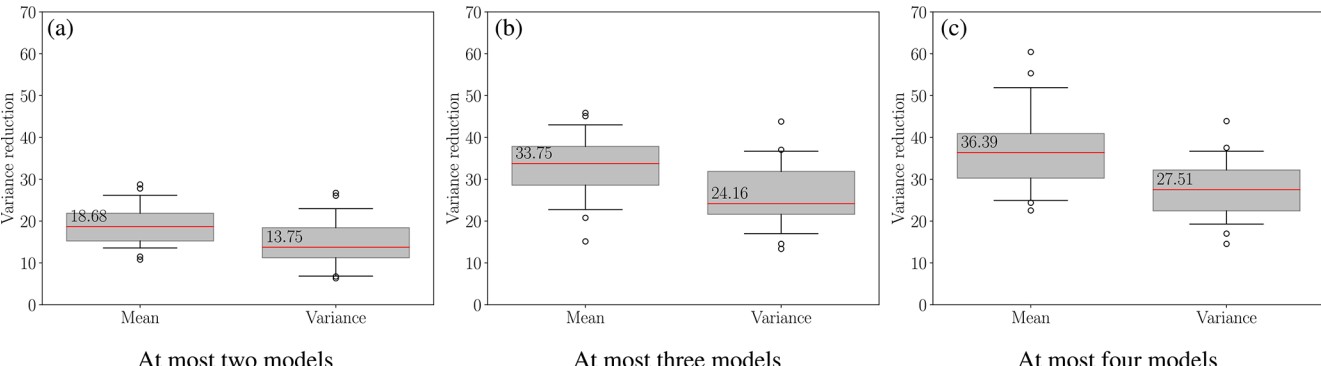

**Figure 11.** The predicted variance reductions, $\mathcal{R}[Q_{\mathrm{ACV}}^{\mu}]$ (mean) and $\mathcal{R}[Q_{\mathrm{ACV}}^{\sigma^2}]$ (variance) (see Eq. 29), of the best ACV estimators obtained by bootstrapping the final 30 pilot samples while enforcing a limit on the number of models an estimator can use, including the highest-fidelity model. The red lines indicate the median estimator variance reductions. The lower and upper whiskers represent the 10 % and 90% quantiles.

## 5.5   Multi-fidelity sea-level rise projections

The cost of constructing our final estimator, $\boldsymbol{Q}_{\mathrm{ACV}}$ in Eq. (23), was equal to the sum of the pilot cost (197.13 h) and the exploitation cost ($160 \times 4.18$) h, which was approxi-
5 mately 36 d on a single CPU. The pilot cost was the sum of evaluating all 13 models on the initial 20 pilot samples and 8 models on an additional 10 pilot samples (see Sect. 5.4). The exploitation cost was fixed at the beginning of the study to the computational cost equivalent to evaluating the high-fidelity
model 160 times and the median time taken to run a single simulation of the glacier for a single realization of basal friction, being 4.18 h. The number of samples allocated to evaluating each model by the ACV estimator during the exploitation phase is shown in Fig. 12. Only two samples of the
high-fidelity model were used. Yet, while running these simulations only accounted for approximately 1.25% of the total computational cost budget, these samples ensured the estimators were unbiased with respect to the highest-fidelity model. In contrast, many more evaluations of the lower-fidelity mod-
els were used. The lower computational costs of these models and their high-correlation with each other and the highest-fidelity model were effectively exploited to significantly reduce the MSE of the ACV estimator relative to the SFMC estimator.
We constructed our final estimator of the mean, $Q_{\mathrm{ACV}}^{\mu}$, and variance, $Q_{\mathrm{ACV}}^{\sigma^2}$, of the mass change by evaluating each model at the number of samples determined by Fig. 12. All models were evaluated on the same two samples, the two low-fidelity models were both evaluated on another 351 sam-
ples, and the $\mathrm{SSA}_{1.5\,\mathrm{km},365\,\mathrm{d}}$ model was evaluated on another 10 130 samples. The small number of samples allocated to the highest-fidelity model was due to the extremely high correlation between that model and the model $\mathrm{MOLHO}_{1\,\mathrm{km},36\,\mathrm{d}}$. This high correlation suggests that the temporal discretiza-
tion error in the highest-fidelity model is smaller than the spatial discretization error for the ranges of discretizations used in this study.

Note that the exact number of samples allocated to each model that we reported is determined by the properties of the MFMC estimator chosen. However, if another estimator,
for example MLMC, was chosen to use the same models, the way samples are shared between models would likely change.[7]

The mean and standard deviation computed using the best ACV estimator were $-639.06 \pm 0.23$ and $17.68 \pm 6.67$ Gt,
respectively. It is clear that with our budget, we were able to confidently estimate the expected mass change in the year 2100. However, our estimates of the standard deviation in the mass change were less precise. We could improve the precision of both estimated statistics by further increasing
the exploitation budget; however, we choose not to do so, as our results emphasize that estimating high-order statistics, such as variance, is more computationally demanding than estimating a mean. Moreover, the precision requirements of a UQ study should be determined by the stakeholders, who
will use the uncertainty estimates to make decisions.

The left panel of Fig. 13 plots the time evolution of mass loss predicted by the three models selected by our final ACV estimator. The right panel plots the distribution of mass loss in the final year, 2100, computed using the $\mathrm{SSA}_{1.5\,\mathrm{km},365\,\mathrm{d}}$
model. The bias in the $\mathrm{SSA}_{1.5\,\mathrm{km},365\,\mathrm{d}}$ is clear in both plots; for example, in the right panel the mean of the blue distribution is not close to the mean computed by the ACV estimator. However, we must emphasize that, by construction, the ACV estimate of the mean mass loss, as well as
its variance, is unbiased with respect to the highest-fidelity model $\mathrm{MOLHO}_{1\,\mathrm{km},9\,\mathrm{d}}$. We also point out that while our Laplace approximation of the posterior is a Gaussian distribution that is obtained by linearizing the steady-state observational model, the push forward of this distribution through
the $\mathrm{SSA}_{1.5\,\mathrm{km},365\,\mathrm{d}}$ model is not a Gaussian distribution; the

---

[7]Of the 10 130 $\mathrm{SSA}_{1.5\,\mathrm{km},365\,\mathrm{d}}$ model simulations, 37 failed, so an additional 37 simulations at new random realizations of the friction field were run.

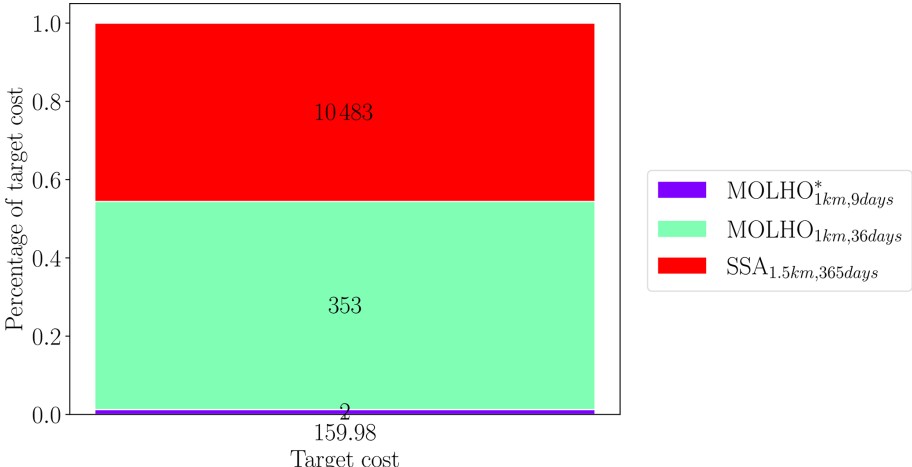

**Figure 12.** The optimal number of samples (number inside rectangles), computed using Eq. (28), required by the best ACV estimator to simulate each model.

right tail of the push-forward density is longer than the left tail. This indicates that the QOIs are nonlinearly dependent on the model parameters. We were unable to compute reasonable push-forward densities with the simulations obtained from the other two models used to construct the ACV estimator due to an insufficient number of simulations. However, we believe it is reasonable to assume that the parameter-to-QoI map of these models is also nonlinear.

## 6 Discussion

The cost of constructing our final estimator was equal to the pilot cost and the exploitation cost, totaling $197.13 + (160 \times 4.18)$ h, or approximately 36 d. Additionally, the median variance reduction obtained by the bootstrapped estimators was $\mathbb{V}_\Theta\left[Q_0^\mu\right]/\mathbb{V}_\Theta\left[Q_{\mathrm{ACV}}^\mu\right] = 38.24$ for estimating the mean and $\mathbb{V}_\Theta\left[Q_0^\mu\right]/\mathbb{V}_\Theta\left[Q_{\mathrm{ACV}}^{\sigma^2}\right] = 28.91$ for estimating the variance of the mass change. Achieving the same precision with SFMC estimators using only the highest-fidelity model would require approximately $28.91 \times 160 \times 4.18\,\mathrm{h} = 805\,\mathrm{d}$. This calculation used the smallest variance reduction motivated by the observation that high-fidelity simulation data can be used to compute both the mean and the variance. Thus, MFSE reduced the cost of estimating uncertainty from over 2.5 years of CPU time to just over a month, assuming the models are evaluated in serial order. Note that while applying MFSE to Humboldt Glacier took over a month of serial computations, the clock time needed for MFSE can be substantially reduced because MFSE is embarrassingly parallel. Each simulation run in the pilot stage can be executed in parallel without communication between them. Similarly, in the exploitation phase, each simulation can also be computed in parallel. Consequently, while using MFSE for continental-scale UQ studies may require years of serial CPU time, distributed computing could substantially reduce this cost, po-

tentially by 1 to 2 orders of magnitude. The exact reduction would depend on the number of CPUs used and the scalability of the computational models.

While the highest-fidelity model, MOLHO, was capable of capturing ice-sheet dynamics that the SSA model was not – that is vertical changes in the horizontal velocities (Fig. 14 shows the different ice thicknesses predicted at the final time by the MOLHO and SSA model) – the best ACV estimator was still able to use the simplified physics of SSA to reduce the MSE of the best ACV estimator. Moreover, the best ACV estimator also used evaluations of the SSA model on a coarse mesh, which failed to resolve all the local features of the friction and ice-sheet flow field (see Fig. 14) and did not conserve mass, unlike the highest-fidelity model. This result demonstrates that, provided there is high correlation between the model predictions of a QoI, MFSE can be effective when there is high correlation between the model predictions of a QoI, even when the model states vary differently across time and space for a single realization of the random model parameters. Moreover, future MFSE studies may benefit from using not only low-fidelity models derived from different physics assumptions and numerical discretizations but also those based on data-driven models, such as machine-learning operators (He et al., 2023; Lowery et al., 2024) or adjoint-based linearizations (Recinos et al., 2023). However, if such models are used, the computational cost of constructing them must also be considered (Peherstorfer, 2019), just as we accounted for the pilot cost in this study.

Our study used a high-dimensional representation of the basal friction field capable of capturing high-frequency modes. However, previous studies have commonly used lower-dimensional parameterizations (Nias et al., 2023; Ritz et al., 2015; Schlegel et al., 2018; Jantre et al., 2024). Consequently, we investigated the impact of using a low-frequency, low-dimensional representation of the friction field on the

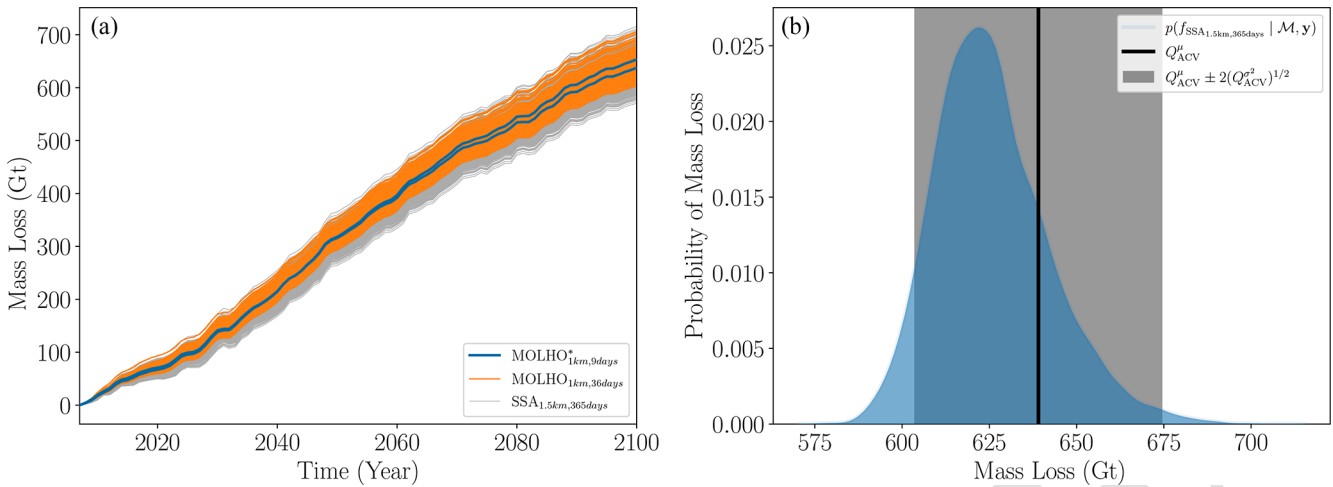

**Figure 13. (a)** The evolution of mass loss predicted by the three models we used in our final ACV estimator, corresponding to each of the simulations used to construct the estimator. **(b)** The probability of mass loss computed using the $SSA_{1.5\,km,365\,d}$ model. The vertical black line represents the ACV estimate of the mean, while the gray-shaded region represents plus and minus 2 standard deviations, again computed by the ACV estimator.

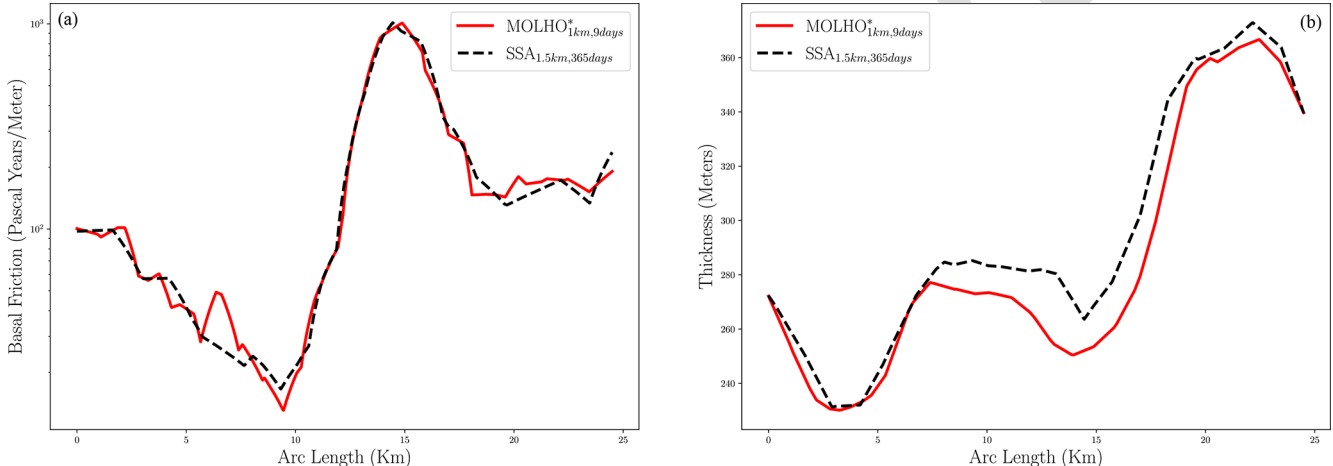

**Figure 14. (a)** The basal friction, $\beta$, along the cross-section (black line) depicted in the right panel of Fig. 7. **(b)** The difference between the thickness fields simulated by the $MOLHO^*_{1\,km,9\,d}$ model and the $SSA_{1.5\,km,365\,d}$ along the same cross-section.

efficiency of ACV estimators applied to ice-sheet models. Specifically, we estimated the mean and variance of the mass change using a 10-dimensional Karhunen–Loève expansion (KLE) to represent the posterior uncertainty in the basal friction field (complete details are presented in Appendix B). We found that using the low-dimensional KLE smoothed realizations of the basal friction, which in turn drastically improved the variance reduction in MFSE to over a factor of 200. However, only using 10 modes to represent the basal friction caused the variance of the mass change to be substantially underestimated. Recinos et al. (2023) also demonstrated that lower-dimensional parameterizations can result in misleading estimates. Consequently, while low-dimensional representations of friction enable faster UQ, the results may be

misleading. Thus, future research is needed to balance the increased bias introduced by the low-dimensional parameterization with the improved variance reduction properties of an ACV estimator.

This study emphasizes that the relative effectiveness of ACV estimators – such as MLMC, MFMC, and ACVMF – is problem dependent. Although each MFSE algorithm in the literature has its own theoretical advantages and disadvantages, it is often difficult to determine which will be the most effective at the onset of a study. Indeed, several types of estimators enumerated by this study yielded estimates of the mean and variance of the mass change with similar precision. For example, Fig. 11a, b, and c show that while using three models is clearly better than using two, there is little,

if any, marginal benefit in moving from three to four models, as indicated by the size of the box plots. Moreover, it is difficult to determine a priori the numerical discretizations and model physics needed by a model ensemble to produce an ACV estimator with the smallest MSE. Consequently, we used a small pilot sample to compute the correlation between model outputs and then used the analytical properties of ACV estimators to predict the MSE of each estimator produced by popular MFSE algorithms.

While pilot studies are required for ACV methods, our results suggest that using a small number of pilot samples can introduce non-trivial variability into the optimal sample allocation used by ACV estimators. Consequently, we introduced a novel two-step bootstrapping procedure to quantify the impact of a small number of pilot samples. While our two-step procedure was able to down-select from a large set of possible models, further research is needed to develop algorithms that can efficiently conduct pilot studies involving a large numbers of models. Furthermore, it is essential that new algorithms balance the computational cost of computing the correlation between models with the impact of the error in the estimated correlations when determining the optimal MSE of an ACV estimator.

Our study predicted the mean and standard deviation of mass change (in metric gigatons) from Humboldt Glacier to be $-639.06$ and $17.68$, respectively. However, the exact values of these statistics were impacted by our modeling choices. First, we only quantified uncertainty due to unknown basal friction, which ignores other contributions to mass-loss variability arising from uncertain climate and ice-sheet processes such as iceberg calving, subglacial hydrology, and submarine melting. Including these processes would have likely affected both the mean and the variance of the mass change. Indeed, our predicted mass loss is significantly less than in two recent studies of Humboldt Glacier (Hillebrand et al., 2022; Carr et al., 2024) due to our use of a low-emissions climate scenario and our neglect of ocean forcing. Moreover, introducing more complicated physics in the highest-fidelity model, such as calving, could degrade the performance of MFSE. For example, ice melt at the boundary can induce strong dynamical responses in a marine-terminating glacier, which could potentially reduce the correlation with models that do not capture this phenomenon. However, despite our imperfect description of uncertainty, we believe our study reflects the challenges of a more comprehensive study while still facilitating a computationally feasible investigation of MFSE methods.

This study focused on investigating the efficacy of using MFSE to accelerate the quantification of parametric uncertainty using deterministic ice-sheet models. We did not quantify the uncertainty arising from model inadequacy. Recently Verjans et al. (2022) attempted to quantify model uncertainty by developing stochastic ice-sheet models designed to simulate the impact of glaciological processes, such as calving and subglacial hydrology, that exhibit variability that cannot be captured by the spatiotemporal resolution typically employed by ice-sheet models. The MFSE algorithms presented in this paper can be applied to such stochastic models by sampling the model parameters and treating the stochasticity of models as noise. However, the noise typically reduces the correlation between models and thus the efficiency of MFSE (Reuter et al., 2024). Moreover, this study only focused on estimating the mean and variance of mass change. Consequently, the efficacy of MFSE may change when estimating statistics – such as probability of failure, entropic risk, and average value at risk (Rockafellar and Uryasev, 2013; Jakeman et al., 2022) – to quantify the impact of rare instabilities and feedback mechanisms in the system. We anticipate that larger numbers of pilot samples than the number used in this study will be needed to estimate such tail statistics, potentially reducing the efficiency of MFSE.

Many recent studies have conducted formal uncertainty quantification of projections of ice-sheet change considering numerous sources of uncertainty, such as climate forcing, iceberg calving, basal friction parameters, and ice viscosity, although these generally deal with scalar parameters, such as a single calving threshold stress (Aschwanden and Brinkerhoff, 2022; Jantre et al., 2024) or scalar adjustment factors to basal friction and ice viscosity fields (Nias et al., 2023; Felikson et al., 2023; Jantre et al., 2024). However, recently automatic differentiation has been used to linearize the parameter-to-QoI map of an SSA model, facilitating the computationally efficient quantification of uncertainty caused by high-dimensional parameterizations of basal friction and ice stiffness (Recinos et al., 2023), Additionally, other UQ studies have primarily relied on a large number of simulations from a single low-fidelity model (e.g., Nias et al., 2019; Bevan et al., 2023), sometimes with informal validation using a small number of higher-fidelity simulations (e.g., Nias et al., 2023), or on the construction of surrogate models to sufficiently sample the parameter space (e.g., Bulthuis et al., 2019; Berdahl et al., 2021; DeConto et al., 2021; Hill et al., 2021; Aschwanden and Brinkerhoff, 2022; Jantre et al., 2024). Furthermore, another set of studies quantified the uncertainty associated with the use of many different numerical models – termed an "ensemble of opportunity" – which includes a wide range of modeling choices that sample parameter values and model fidelity in an unsystematic manner (Edwards et al., 2021; Seroussi et al., 2023; Van Katwyk et al., 2023; Yoo et al., 2024). While this study is limited in scope because it focuses on solely estimating parametric uncertainty induced by basal friction variability, our results demonstrate that even when low-fidelity ice-sheet models do not capture the flow features predicted by higher-fidelity models, they can still be effectively utilized by MFSE methods to reduce the cost of quantifying high-dimensional parametric uncertainty in ice-sheet model predictions. Consequently, low-fidelity models, when used with MFSE methods, may be able to substantially reduce the computational cost of future efforts to quantify uncertainty in the projection

of the mass change from the entire Greenland Ice Sheet and entire Antarctic Ice Sheet.

## 7 Conclusions

Mass loss from ice sheets is anticipated to contribute $O(10)$ cm to sea-level rise in the next century under all but the lowest-emission scenarios (Edwards et al., 2021). However, projections of sea-level rise due to ice-sheet mass change are inherently uncertain, and quantifying the impact of this uncertainty is essential for making these projections useful to policymakers and planners. Unfortunately, accurately estimating uncertainty is challenging because it requires numerous simulations of a computationally expensive numerical model. Consequently, we evaluated the efficacy of MFSE for reducing the computational cost of quantifying uncertainty in projections of mass loss from Humboldt Glacier, Greenland.

This study used MFSE to estimate the mean and the variance of uncertain mass-change projections caused by uncertainty in glacier basal friction using 13 different models of varying computational cost and fidelity. While ice sheets are subject to other sources of uncertainty, focus was given to basal friction because its inherently high dimensionality typically makes quantifying its impact on the uncertainty in model predictions challenging. Yet, despite this challenge, we found that for a fixed computational budget, MFSE was able to reduce the MSE in our estimates of the mean and variance of the mass change by over an order of magnitude compared to an SFMC-based approach that just used simulations from the highest-fidelity model.

In our study, we were able to use MFSE to substantially reduce the MSE error in the statistics by exploiting the correlation between the predictions of the mass change produced by each model. However, it was not necessary to use simulations from all of the models to reduce the MSE. Indeed, the MFSE algorithm determined that only three models (including the highest-fidelity model) were needed to minimize the MSE in the statistics given our computational budget. The low-fidelity models selected (1) used simplifications of the high-fidelity model physics, (2) were solved on coarser-resolution spatial and temporal meshes, and (3) were solved without the requirement of mass conservation. These simplifications result in significant computational cost savings relative to use of the high-fidelity model alone. This result demonstrated that MFSE can be effective even when the lower-fidelity models are incapable of capturing the local features of the ice-flow fields predicted by the high-fidelity model. Moreover, while the utility of the lower-fidelity models ultimately chosen for MFSE was not clear at the onset of the study, we were still able to estimate uncertainty at a fraction of the cost of single-fidelity MC approaches. This was achieved despite the need to conduct a pilot study that evaluated all models a small number of times.

Finally, this study demonstrated that MFSE can be used to reduce the computational cost of quantifying parametric uncertainty in projections of a single glacier, which suggests that MFSE could plausibly be used for continental-scale studies of ice-sheet evolution in Greenland and Antarctica. However, the predicted mean mass loss from Humboldt Glacier that we reported should be viewed with caution, as the length scales of the prior distribution we employed for the uncertain basal friction field were not finely tuned, e.g., as done by Recinos et al. (2023), due to the computational expense of such a procedure. Since the length scales of basal friction are very difficult to determine a priori, the potentially over-informative length scales used in this paper could substantially impact the posterior mass loss estimates. Future research should address this issue and increase the complexity of this study in two further directions. First, future studies should include additional sources of ice-sheet uncertainty beyond the basal friction field studied here, for example uncertain surface mass balance and ocean forcing. Second, future studies should include the use of model fidelities that capture additional physical processes such as calving, fracture, and ocean-forced melting. Consequently, while our findings should be interpreted with caution given the aforementioned limitations, they encourage future studies to utilize MFSE for reducing the cost of computing probabilistic projections of sea-level rise due to ice-sheet mass change.

## Appendix A: Low-rank Laplace approximation

Following Bui-Thanh et al. (2013) and Isaac et al. (2015), we computed the covariance of the Laplace approximation of the posterior distribution of the friction parameters (Eq. 14) using

$$\mathbf{\Sigma}_{\mathrm{post}} = \left(\mathbf{H}_{\mathrm{MAP}} + \mathbf{\Sigma}_{\mathrm{prior}}^{-1}\right)^{-1} = \mathbf{L}\left(\mathbf{L}^\top \mathbf{H}_{\mathrm{MAP}} \mathbf{L} + \mathbf{I}\right)^{-1} \mathbf{L}^\top,$$

where $\mathbf{H}_{\mathrm{MAP}}$ is the Hessian of $\frac{1}{2}(\boldsymbol{y} - \boldsymbol{g}(\boldsymbol{\theta}))^\top \mathbf{\Sigma}_{\mathrm{noise}}^{-1}(\boldsymbol{y} - \boldsymbol{g}(\boldsymbol{\theta}))$ at $\boldsymbol{\theta} = \boldsymbol{\theta}_{\mathrm{MAP}}$; $\mathbf{L} = \mathbf{K}^{-1}\mathbf{M}^{\frac{1}{2}}$; and the entries of $\boldsymbol{K}$ and $\boldsymbol{M}$ are defined in Eqs. (10) and (11), respectively.

Drawing samples from this Gaussian posterior is computationally challenging because the posterior covariance $\mathbf{\Sigma}_{\mathrm{post}}$ depends on the Hessian $\mathbf{H}_{\mathrm{MAP}}$, which is a high-dimensional dense matrix. Consequently, following Bui-Thanh et al. (2013) and Isaac et al. (2015), we constructed a low-rank approximation of the prior-preconditioned Hessian $\mathbf{L}^\top \mathbf{H}_{\mathrm{MAP}} \mathbf{L}$ using matrix-free randomized methods that require only multiplications of the Hessian with random vectors. Specifically, computing a spectral decomposition of

$$\mathbf{L}^\top \mathbf{H}_{\mathrm{MAP}} \mathbf{L} = \mathbf{U} \boldsymbol{\Lambda} \mathbf{U}^\top, \tag{A1}$$

with $\mathbf{U}$ orthogonal and $\boldsymbol{\Lambda}$ diagonal matrices and noting

$$\mathbf{\Sigma}_{\mathrm{post}} = \mathbf{L}\left(\mathbf{U} \boldsymbol{\Lambda} \mathbf{U}^\top + \mathbf{I}\right)^{-1} \mathbf{L}^\top$$

$$= \mathbf{L}\left(\mathbf{U}(\boldsymbol{\Lambda} + \mathbf{I})\mathbf{U}^\top\right)^{-1} \mathbf{L}^\top = \mathbf{L}\mathbf{U}(\boldsymbol{\Lambda} + \mathbf{I})^{-1}\mathbf{U}^\top \mathbf{L}^\top,$$

we factorized $\boldsymbol{\Sigma}_{\text{post}}$ as

$$\boldsymbol{\Sigma}_{\text{post}} = \mathbf{T}\mathbf{T}^{\top},$$

$$\mathbf{T} = \mathbf{L}\mathbf{U}(\boldsymbol{\Lambda} + \mathbf{I})^{-\frac{1}{2}}\mathbf{U}^{\top} = \mathbf{L}\mathbf{U}\left((\boldsymbol{\Lambda} + \mathbf{I})^{-\frac{1}{2}} - \mathbf{I}\right)\mathbf{U}^{\top} + \mathbf{L}.$$

In order to perform a low-rank approximation of the matrix $\mathbf{T}$, we truncated the spectral decomposition of $\mathbf{W} = \mathbf{U}\left((\boldsymbol{\Lambda} + \mathbf{I})^{-\frac{1}{2}} - \mathbf{I}\right)\mathbf{U}^{\top}$ by discarding the eigenvalues $\lambda_i$ such that $\left|1 - \frac{1}{\sqrt{\lambda_i+1}}\right| \ll 1$. This ensured that the low-rank approximation of $\mathbf{T}$ approximated $\mathbf{T}$ well in the spectral-norm sense. The eigenvalues and two eigenvectors of the spectral decomposition we computed are depicted in Fig. A1.

We computed the truncated spectral decomposition using randomized algorithms (see Hartland et al., 2023; Halko et al., 2011) implemented in `PyAlbany`; see Liegeois et al. (2023). The algorithms used were matrix-free and only required the multiplication of $\mathbf{L}^{\top}\mathbf{H}_{\text{MAP}}\mathbf{L}$ with vectors. Moreover, as described in Hartland et al. (2023) and Isaac et al. (2015), the multiplication of the Hessian with a vector required solving two adjoint systems of the flow model. Similarly, the multiplication of the matrix $\mathbf{L}$ with a vector required the solution of the two-dimensional linear elliptic system with matrix $\boldsymbol{K}$, defined in Eq. (10). Consequently, we were able to efficiently draw samples from the posterior distribution of the friction parameters using

$$\boldsymbol{\theta}_{\text{post}} = \boldsymbol{\theta}_{\text{MAP}} + \mathbf{T}\boldsymbol{n}, \quad \boldsymbol{n} \sim \mathcal{N}(\mathbf{0}, \mathbf{I}).$$

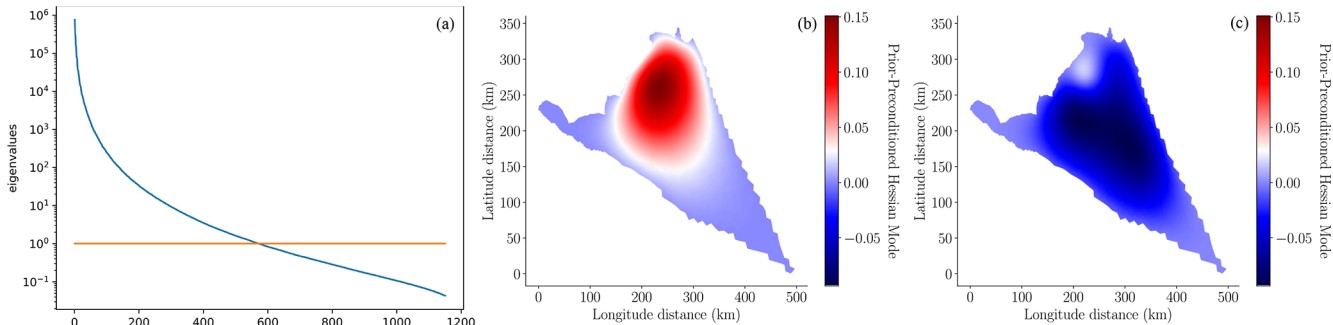

**Figure A1.** **(a)** Eigenvalues $\lambda_i$ of the prior-preconditioned Hessian, computed by solving Eq. (A1) and **(b, c)** its eigenvectors associated with the **(b)** largest and **(c)** third-largest eigenvalue. Note that similarly to Isaac et al. (2015), we plot the eigenvectors $\mathbf{V}_i = \mathbf{L}\mathbf{U}_i$ that are orthonormal with respect to the prior-induced dot product; that is, $\mathbf{V}_i^\top \mathbf{\Sigma}_{\text{prior}}^{-1} \mathbf{V}_j = \delta_{ij}$.

## Appendix B: Low-dimensional representation of basal friction using a Karhunen–Loève expansion

In our main study we found that when using a high-dimensional representation of the uncertainty in the basal friction field, bootstrapped ACV estimators rarely chose to use models that had coarse spatial meshes relative to the mesh used by the high-fidelity model. This was likely due to the fact that our high-dimensional representation of the friction uncertainty was constructed on the high-fidelity mesh and interpolated onto coarser meshes. To verify this hypothesis, we investigated using a lower-dimensional representation of the friction field based on a Karhunen–Loève expansion (KLE) of the friction field that smoothed out the high-frequency variations in the posterior samples of the friction field we used in our main study.

### B1   Construction of the KLE

In our investigations we used a KLE,

$$\boldsymbol{\theta} = \boldsymbol{\theta}_{\text{MAP}} + \sum_{i=1}^{D} \sqrt{\lambda_i}\,\psi_i\,\eta_i, \qquad \eta_i \sim \mathcal{N}(0, 1), \tag{B1}$$

to provide a low-dimension representation of the Laplace approximation of the posterior of the log basal friction field. We computed the eigenvalues $\lambda_i$ and the orthonormal eigenvectors $\psi_i$ by solving the eigenvalue problem,

$$\mathbf{\Sigma}_{\text{post}}\psi_i = \lambda_i \psi_i, \tag{B2}$$

using the randomized matrix-free methods of Hartland et al. (2023) and Halko et al. (2011).

While a KLE basis could have been constructed on any of the four meshes we considered, in this study we solved the discretized eigenvalue problem using the finest mesh. The 1st, 2nd, and 10th modes of the KLE used in this study are depicted in Fig. B1. Note that unlike what is typically seen when constructing a KLE of a field with a pointwise variance that is constant across the domain, the low-frequency KLE modes constructed here are localized where the posterior uncertainty is highest. The finite-element basis on the finest mesh was then used to interpolate the KLE basis from the fine mesh onto the coarser meshes. This procedure ensured that varying the coefficients of the KLE basis (the random parameters of the model) would affect each model similarly regardless of the mesh discretization employed. Similarly to the KLE basis, the mean of the log KLE field (taken to be the mean of the Laplace approximation) was computed on the finest mesh.

Figure B2 compares a realization of the log of the basal friction perturbation (mean zero) drawn from the Laplace approximation of the posterior and a random realization of the log of the basal friction perturbation computed using the KLE. It is clear that the KLE smooths out much of the high-frequency content present in the realization drawn from the Laplace approximation of the posterior.

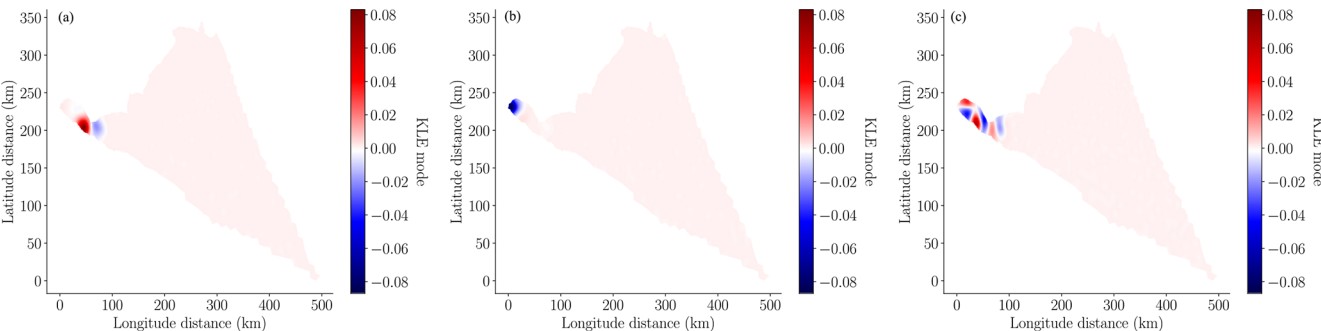

**Figure B1.** From **(a)** to **(c)**, the 1st, 2nd, and 10th modes of the KLE (Eq. B1) used in this study, computed using Eq. (B2)

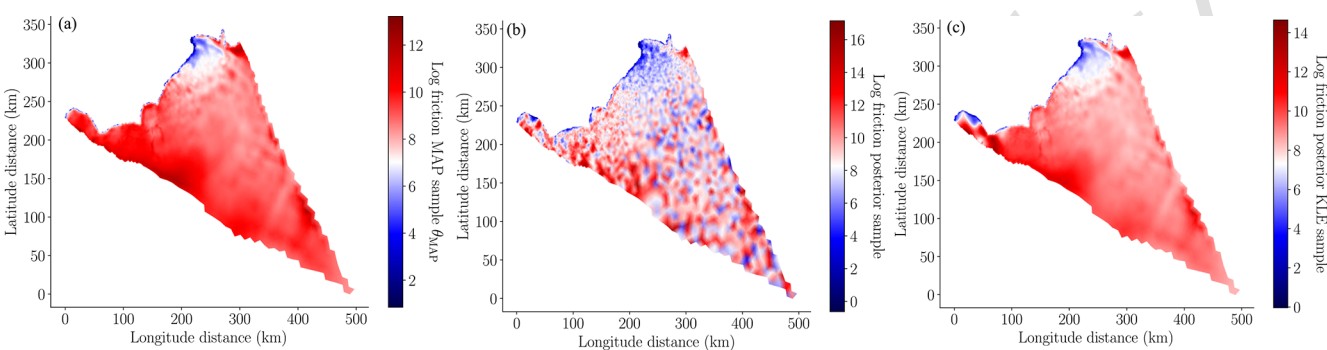

**Figure B2. (a)** The mean of the log of the basal friction, $\theta_{\mathrm{MAP}}$ in Eq. (13). **(b)** A random realization of the log of the basal friction drawn from the Laplace approximation of the posterior $p(\theta \mid \mathcal{M}, y) \sim \mathcal{N}(\theta_{\mathrm{MAP}}, \Sigma_{\mathrm{post}})$. **(c)** A random realization of the log of the 10-dimensional basal friction computed using the KLE approximation (Eq. B1) of the posterior.

## B2  Pilot study

In this section, we detail the pilot study we undertook to investigate the impact of using a low-dimensional KLE to represent friction when using MFSE to estimate statistics of mass change. We did not move beyond the pilot study to compute the values of the statistics to limit the computational cost of this supplementary study.

First, we evaluated each of our 13 models at 20 random pilot samples of the KLE. Second we computed the pilot statistics needed to find the best ACV estimator. Third we bootstrapped the pilot samples to estimate the median and confidence intervals on the variance reduction obtained by the best ACV estimator.

The mean and variance bootstrapped variance reduction are depicted in Fig. B4. The variance reductions reported are almost an order-of-magnitude larger than those reported for MFSE based on the Laplace approximation of the posterior. This improved performance is because correlations between the models (Fig. B3) are significantly higher than the correlations obtained when sampling from the Laplace approximation of the posterior (Fig. 9). However, the KLE representation underestimates the uncertainty in the predicted mass change at 2100. Specifically, the standard deviation of the mass change computed using 20 pilot samples of the highest-fidelity model using the Laplace approximation of the posterior is significantly higher than the standard deviation computed using the KLE (see Fig. B5).

| | $\text{MOLHO}^*_{1km,9days}$ | $\text{MOLHO}_{1km,36days}$ | $\text{MOLHO}_{1.5km,36days}$ | $\text{MOLHO}_{2km,36days}$ | $\text{MOLHO}_{3km,36days}$ | $\text{SSA}_{1km,36days}$ | $\text{SSA}_{1.5km,36days}$ | $\text{SSA}_{2km,36days}$ | $\text{SSA}_{3km,36days}$ | $\text{SSA}_{1km,365days}$ | $\text{SSA}_{1.5km,365days}$ | $\text{SSA}_{2km,365days}$ | $\text{SSA}_{3km,365days}$ |
|---|---|---|---|---|---|---|---|---|---|---|---|---|---|
| $\text{MOLHO}^*_{1km,9days}$ | 1.0000 | 1.0000 | 0.9989 | 0.9988 | 0.9919 | 0.9998 | 0.9986 | 0.9990 | 0.9923 | 0.9998 | 0.9987 | 0.9990 | 0.9923 |
| $\text{MOLHO}_{1km,36days}$ | 1.0000 | 1.0000 | 0.9989 | 0.9988 | 0.9919 | 0.9998 | 0.9986 | 0.9990 | 0.9923 | 0.9998 | 0.9987 | 0.9990 | 0.9923 |
| $\text{MOLHO}_{1.5km,36days}$ | 0.9989 | 0.9989 | 1.0000 | 0.9990 | 0.9913 | 0.9988 | 0.9998 | 0.9993 | 0.9918 | 0.9987 | 0.9998 | 0.9993 | 0.9918 |
| $\text{MOLHO}_{2km,36days}$ | 0.9988 | 0.9988 | 0.9990 | 1.0000 | 0.9927 | 0.9982 | 0.9983 | 0.9999 | 0.9925 | 0.9981 | 0.9983 | 0.9999 | 0.9925 |
| $\text{MOLHO}_{3km,36days}$ | 0.9919 | 0.9919 | 0.9913 | 0.9927 | 1.0000 | 0.9916 | 0.9910 | 0.9929 | 0.9997 | 0.9916 | 0.9911 | 0.9930 | 0.9997 |
| $\text{SSA}_{1km,36days}$ | 0.9998 | 0.9998 | 0.9988 | 0.9982 | 0.9916 | 1.0000 | 0.9989 | 0.9987 | 0.9924 | 1.0000 | 0.9989 | 0.9987 | 0.9923 |
| $\text{SSA}_{1.5km,36days}$ | 0.9986 | 0.9986 | 0.9998 | 0.9983 | 0.9910 | 0.9989 | 1.0000 | 0.9990 | 0.9919 | 0.9989 | 1.0000 | 0.9990 | 0.9919 |
| $\text{SSA}_{2km,36days}$ | 0.9990 | 0.9990 | 0.9993 | 0.9999 | 0.9929 | 0.9987 | 0.9990 | 1.0000 | 0.9931 | 0.9986 | 0.9990 | 1.0000 | 0.9931 |
| $\text{SSA}_{3km,36days}$ | 0.9923 | 0.9923 | 0.9918 | 0.9925 | 0.9997 | 0.9924 | 0.9919 | 0.9931 | 1.0000 | 0.9924 | 0.9920 | 0.9931 | 1.0000 |
| $\text{SSA}_{1km,365days}$ | 0.9998 | 0.9998 | 0.9987 | 0.9981 | 0.9916 | 1.0000 | 0.9989 | 0.9986 | 0.9924 | 1.0000 | 0.9989 | 0.9987 | 0.9924 |
| $\text{SSA}_{1.5km,365days}$ | 0.9987 | 0.9987 | 0.9998 | 0.9983 | 0.9911 | 0.9989 | 1.0000 | 0.9990 | 0.9920 | 0.9989 | 1.0000 | 0.9990 | 0.9920 |
| $\text{SSA}_{2km,365days}$ | 0.9990 | 0.9990 | 0.9993 | 0.9999 | 0.9930 | 0.9987 | 0.9990 | 1.0000 | 0.9931 | 0.9987 | 0.9990 | 1.0000 | 0.9931 |
| $\text{SSA}_{3km,365days}$ | 0.9923 | 0.9923 | 0.9918 | 0.9925 | 0.9997 | 0.9923 | 0.9919 | 0.9931 | 1.0000 | 0.9924 | 0.9920 | 0.9931 | 1.0000 |

**Figure B3.** The correlations, $\mathbb{C}\text{orr}_\pi[\boldsymbol{f},\boldsymbol{f}]$ with $\boldsymbol{f}=[f_0,\ldots,f_M]^\top$, between the 13 ice-sheet models considered by this study using 20 pilot samples of the KLE and Eq. (B1).

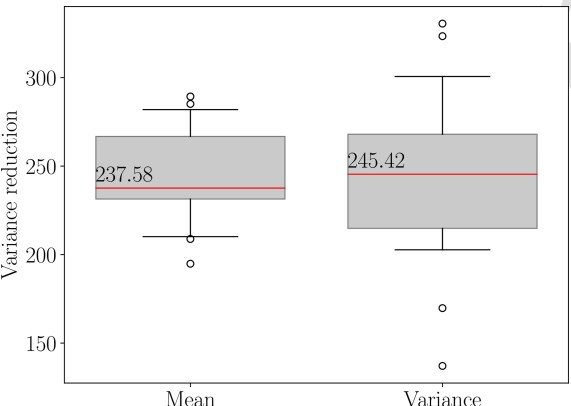

**Figure B4.** The predicted variance reductions $\mathcal{R}_\Theta[Q^\mu_{\text{ACV}}]$ (mean) and $\mathcal{R}_\Theta[Q^{\sigma^2}_{\text{ACV}}]$ (variance) (see Eq. 29), obtained using bootstrapping of the 20 pilot samples of the KLE (Eq. B1). The red lines represent the median estimator variance reductions. The lower and upper whiskers represent the 10 % and 90 % quantiles.

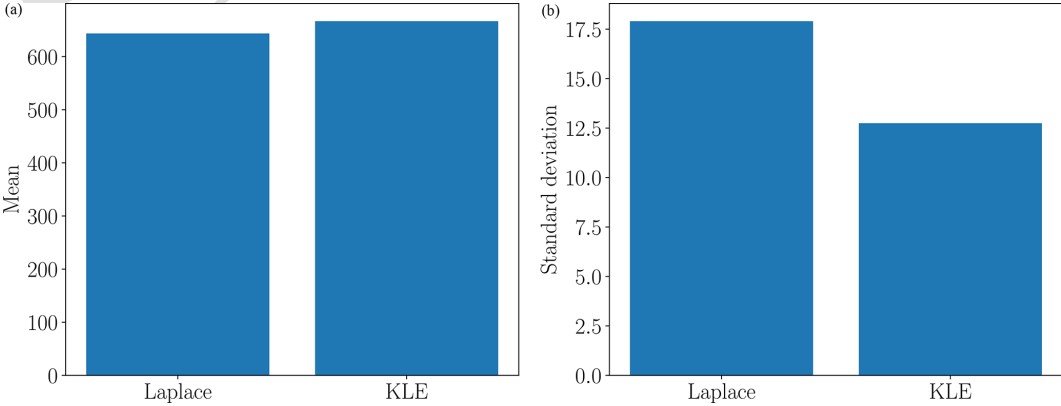

**Figure B5. (a)** The mean, $Q^\mu_0(\Theta_{\text{pilot}})$, and **(b)** standard deviation, $\sqrt{Q^{\sigma^2}_0(\Theta_{\text{pilot}})}$, computed using 20 pilot samples from the Laplace approximation of the posterior and the KLE.

**Code availability.** The code used to construct ACV estimators has been released in the open-source Python package `PyApprox` (https://github.com/sandialabs/pyapprox, Jakeman, 2023).

**Data availability.** The National Snow and Ice Data Center QGreenland package (https://doi.org/10.5281/zenodo.12823307, Moon et al., 2023 TS3) was used to produce the image of Greenland in Fig. 1. TS4

**Author contributions.** JDJ was responsible for formulation of the overarching research goals and aims (conceptualization), data curation, application of the statistical techniques used to analyze the data (formal analysis), conducting the computer experiments (investigation), developing the methodology, implementing and maintaining the software used (software), oversight of and leadership for the research planning and execution (supervision), and writing the original draft. MP was responsible for investigation, methodology, software and writing the original draft. DTS was responsible for data curation, investigation, software, and writing the original draft. TAH was responsible for software and writing the original draft. TRH was responsible for developing the data curation, methodology, and writing the original draft. MJH was responsible for developing the conceptualization, methodology, supervision, funding acquisition, and writing the original draft. SFP was responsible for developing the conceptualization, funding acquisition, and writing the original draft.

**Competing interests.** The contact author has declared that none of the authors has any competing interests.

**Special issue statement.** This article is part of the special issue "Theoretical and computational aspects of ensemble design, implementation, and interpretation in climate science (ESD/GMD/NPG inter-journal SI)". It is not associated with a conference.

**Acknowledgements.** The authors would like to thank Michael Eldred for his helpful discussions on how to practically assess the impact of a small number of pilot samples on the predicted variance of ACV estimators. The authors would also like to thank Thomas Dixon and Alex Gorodetsky for their useful discussions on constructing ACV estimators for multiple statistics. Finally, the authors would like to thank Kenneth Chad Sockwell for developing the `FEniCS` ice-sheet code.

**Financial support.** This research has been supported by the Advanced Scientific Computing Research and the Biological and Environmental Research programs and the US Department of Energy's Office of Science Advanced Scientific Computing Research, Biological and Environmental Research, and Scientific Discovery through Advanced Computing (SciDAC) programs. TS5

**Review statement.** This paper was edited by Francisco de Melo Viríssimo and reviewed by Douglas Brinkerhoff, Vincent Verjans, and Dan Goldberg.

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

## Remarks from the language copy-editor

CE1 Thank you for the definition. Because "s.t." is used in a few places, I added the definition here rather than write it out in every instance.

## Remarks from the typesetter

TS1 Please give an explanation of why the equation needs to be changed. We have to ask the handling editor for approval. Thanks.

TS2 Please confirm the equation.

TS3 Please confirm DOI and citation.

TS4 Should this be moved back to the Acknowledgements, and "No data sets were used in this article." be included in the Data availability section (note: the Data availability section is mandatory)?

TS5 Please confirm the Disclaimer, Acknowledgements and Financial support sections.

TS6 Is this a paper or software?

TS7 Please confirm reference list entry with the information given in the DOI. Is it correct?

TS8 Please confirm reference list entry.

TS9 Please confirm addition and provide date of last access.