# Peer review of "An evaluation of multi-fidelity methods for quantifying uncertainty in projections of ice-sheet mass-change"

_EGUsphere, 2024_

## Referee Comment (RC1)

**Review of 'An evaluation of multi-fidelity methods for quantifying uncertainty in projections of ice-sheet mass-change' by Jakeman et al.**

Doug Brinkerhoff

August 20, 2024

In this manuscript, Jakeman et al. present the application of multi-fidelity uncertainty quantification to accelerate – and hopefully provide more accurate – estimates of first- and second-order ensemble statistics. They begin by describing two approximations to the Stokes' equations, which trade solution expressivity for computational expense and serve as the basis for their multifidelity methods. They next describe a mechanism by which to characterize an approximate posterior distribution (based on a low-rank Laplace approximation) over basal traction conditioned on surface velocity observation, which serves as the source of samples for Monte Carlo sampling of ice volume evolution, the primary quantity of interest in this work. The primary methodological advance is the introduction of an adaptive control variate (ACV) estimator for the mean and variance of mass change after approximately a century of ice evolution. This estimator leverages correlations between so-called low- and high-fidelity models (which have different computational expense) to effectively reduce the error in Monte Carlo estimates of ice volume change relative to predictions made using a limited number of high-fidelity model predictions on its own. They present this method for a single high and low-fidelity model, and then extend the analysis to the case where there exists one high-fidelity model and a hierarchy of multiple low-fidelity models. Such methods require the establishment of statistical relationships between the low and high-fidelity models, and also careful selection of the number of samples evaluated for each consituent model: the authors carefully present strategies for these tasks within the framework of a fixed computational budget, and explore the implications of these strategies when they are better informed by empirical analysis than theory. The manuscript applies these methods to the Humboldt Glacier basin of Northwest Greenland and show that the present methods can be used to perform effective uncertainty quantification, at least over the limited subset of uncertain parameters that the authors' consider.

This work is an important and timely contribution to the growing effort towards robust uncertainty quantification in ice sheet modeling. I have no major objections to the scientific content of this work, which I find to be well-motivated and defensible. I do think that the work relies on language and a presentation style that will be challenging for many readers, particularly those without a specialized statistical background. As a general comment, I would encourage the authors to try to provide more intuition and plain-language summaries, particularly in Section 4. I provide more specific examples, alongside other detailed comments below.

**L26** Here and elsewhere, 'effects' should be 'affects'

**L97** Should $H(x, y, z)$ be $H(x, y, t)$?

**L98** The MOLHO (or the Blatter-Pattyn approximation) doesn't neglect vertical velocity, it is just eliminated from the system of equations via mass conservation and the assumption of hydrostatic pressure. It can always be determined a posteriori from the horizontal velocity components.

**L127** I don't think Dukowicz is the best reference here. Pattyn (2003) is more commonly cited, or if the preference is for something that clearly describes the hierarchy of approximations, Schoof and Hindmarsh (2009).

**L137** I think that citet should be used rather than citep here.

**L152** The discretization of the continuity equation is non-trivial and should be described here. How was it stabilized? How was positivity (even in the absence of negative forcing) ensured?

**L172** Who specifically considers friction to be such a large source of uncertainty? Plenty of recent work has shown that forcing terms are the most important uncertainty sources, particularly at long time scales. I don't have a problem with focusing on traction here, but I think it is important to contextualize this choice a little bit more fully.

**L180** I think that the community is using the term 'Gaussian process' with some frequency now, so it would be good to at least mention that as a name for what is going on here (and a reference to, say Rasmussen and Williams (2006)).

**L212** For what it's worth, it's a stretch to call BedMachine 'data' – it is the result of a PDE-constrained optimization scheme that relies on assumptions of climate, smoothness, and a variety of other things. Again, nothing different needs to be done, but it is important to state that this inferred geometry is assumed error-free.

**L232** This scaling term is less mysterious when reported with units 'number of observations per area'.

**L263–267** Here and elsewhere, please be sure to use a consistent tense. This switches from present to past inside a sentence.

**L298** Should these sums have $N^{-1}$ in front of them?

**L298** For the inner sum, please use a different index variable than $n$.

**L298** Is the optimization of $\eta$ mandatory or does this work with arbitrary $\eta$? What is the objective that is optimized?

**L298** The reader would benefit from a description of what this equation means and some intuition of why this works. It appears to be that the low-fidelity terms yield a correction to the high fidelity statistic, but it is somewhat surprising that this doesn't need to include any explicitly quantified relationship between the two models. It would also be helpful to emphasize that the $\Theta_0$ and $\Theta_1$ can have different set sizes.

**L323** The line about some samples being shared is vague. Please elaborate on what this means.

**L328** 'statistics' $\rightarrow$ 'statistic'.

**Eq. 14** As before, are there alternatives to using this value for $\eta$?

**Eq. 16** Split this into two equations, and add matrix sizes for each.

**L352** I'm not sure I understand this sentence.

**L356** I think it would be better to include more detail about how these expressions are used to compute Eq. 15 than just referencing Dixon (2023). Otherwise, it sort of feels like a lot of space gets used describing Eqs. 16 and 17, but they never really go anywhere.

**Eq. 20** Why is it the case that minimizing the determinant of the covariance determines an optimal sampling strategy?

**Fig. 4 and accompanying text** I don't think that the text does a sufficient job of describing the principles behind these different sampling strategies. Figure 4 tells me that $\Theta_0$ and $\Theta_1^*$ share their samples, and different schemes use entirely different or appended different samples for $\Theta_1$, but I cannot grasp from the text why this is significant. This needs to be motivated fully or de-emphasized and more carefully referenced.

**L394** 'model' appears twice.

**L433–439** Is there an argument that can be made here to reassure a reader that the observed changes are due to real climate/ice dynamic effects and not so-called 'transients' resulting from inconsistency between initial conditions, physics, and input fields? Fig. 7 (left) has some rather surprising high-frequency noise in the surface elevation change – it would be helpful to know where this comes from.

**L453** Just to clarify, was $\Theta_{pilot}$ shared across models, or were the samples different for each model?

**L458** 'significant' is an unfortunately subjective term here - to my eye, the differences in the heights of the referenced bars seems rather insignificant. Is it possible to elaborate on the meaning of 'significant' here, and why it should be viewed as such?

**L467** This sentence is a bit challenging, with 5(!) nested prepositions and two uses of 'variance' each describing different things. I recognize that it is challenging to compactly describe the statistics of statistics, but is it possible to relax this sentence a bit?

**L476** This assertion is surprising to me, and a citation describing the assumption of pilot statistic exactness would be useful.

**L494** Is there an interpretation of why some models appear to be more informative than others, or is this just random chance? I can't identify a mechanism for why some low-fidelity models were chosen more frequently, but understanding that (or being able to predict it a priori) would be exceptionally useful.

**L496** I don't understand what is meant by a 'hierarchical relationship' here.

**L525** I get where these numbers come from after some digging back through the other sections, but it would be helpful to remind the reader where each of the terms in the cost expression represent.

**L536** Is this extreme asymmetry between the number of high- and low-fidelity model evaluations typical? I think that this is a significant and interesting result if the high-fidelity model is only really needed to, e.g. characterize the spatial variability of the mean solution, but the low-fidelity models are sufficient to characterize all of the uncertainty.

**Table 1** I don't think this needs to be a table.

**L589** The variance doesn't have the same units as the mean, so I'm not sure what numbers I'm looking at. Is 17.68 the standard deviation?

---

## Referee Comment (RC2)

Review of "An evaluation of multi-fidelity methods for quantifying uncertainty in projections of ice-sheet mass-change" by Jakeman et al.
Reviewer: Vincent Verjans

This study proposes a multi-model method for evaluating uncertainties in ice sheet model projections. This method uses models of different degrees of fidelity to simulate glacier mass change projections, therefore referred to as multi-fidelity uncertainty quantification (MFUQ). It exploits correlation between the different model realizations to approximate the statistics that would be obtained by the highest-fidelity model available, but at reduced computational cost. Here, the study focuses on uncertainty arising from the uncertain basal friction input field, and shows an application at Humboldt glacier, Greenland. Random samples of the basal friction field are drawn from a Laplace approximation of the posterior probability distribution, which is calibrated to match output from an ice flow model to the present-day Humboldt glacier configuration. The study then compares the MFUQ method with Monte Carlo sampling using the highest-fidelity model only, which is referred to as single-fidelity Monte Carlo (SFMC). Results show that, applied to this problem, MFUQ can serve to infer the mean and variance statistics with large computational savings compared to SFMC. The MFUQ procedure splits the computational burden by using only few high-fidelity model runs and a large number of lower-fidelity model runs, and then exploiting the correlation between both sets of runs.

This study is a valuable contribution to the field of uncertainty quantification in ice sheet modeling. It demonstrates that combining multiple levels of model fidelity can serve to improve uncertainty estimates in useful quantities, which is an approach scarcely used in this field. The science presented in this study uses elaborate statistical techniques, which is a good thing. And I evaluate the scientific aspects of this study positively. However, I believe that major efforts should be made on two presentation aspects. First, more clarity is needed in the presentation of the MFUQ method. I needed to re-read and go back-and-forth between different sections multiple times to really understand the procedure. Second, the authors should try to guide the reader in understanding the procedure, and to provide some intuitive explanations of the different steps in addition to the mathematical details. This latter aspect would better align with the readership of Earth System Dynamics, which is not primarily focused on methodological developments per se. I separate this review in one Major comment, focused on the most important clarifications required, and line-by-line comments focused on less important aspects that need elaboration, as well as on scientific aspects that could be slightly adjusted or more thoroughly explored. Line numbers (L) refer to lines in the preprint. Although my review insists a lot on presentation aspects, I find it also important that the science-related comments are addressed. I encourage the authors to revise their manuscript following comments from other reviewers and me. Given the strong scientific basis of this study, I am certain that a revised version of this manuscript will be a valuable contribution to the literature.

**Major comment: mathematical presentation**
There is no single specific aspect that makes the mathematical presentation unclear. Instead, it is the accumulation of various elements that renders understanding the methods challenging. I try to identify some of these elements here.

I) Equations should be better explained and without errors.
In all equations with matrices, please provide explicitly the dimensions of the matrices involved. This would help to understand, for example, Eqs. 13, 16, 17. It would also be helpful to explicitly mention if a quantity is a scalar, vector, or matrix when it is used for the first time.

In Eq. 12, the covariance term has twice the same argument, and can therefore not be a covariance. Furthermore, I am not convinced of the validity of the $\mathrm{Var}(Q^{\sigma^2})$ formula, so please provide a reference and/or a detailed derivation in the response.

In L298, to be valid, this equation requires some normalizing terms $(1/N_0, 1/N_0, 1/N_1)$.

In the first part of Eq. 16, one term should be $\mathrm{Cov}[\Delta_\alpha^{\sigma^2}, \Delta_\beta^\mu]$.

In Eq. 11, both $Q_\alpha$ and $Q$ are referred to as MC estimators (in Eq. (10) and on L271, respectively). It would be nice define the MC estimator precisely, as well as the quantity that it is estimating.

Please be consistent in the notation. For example, $Q_{\mathrm{ACV}}$ is bold in Eq. 15, but not in Eq. 20.

Please use a symbol to represent only one single variable or parameter. For example, the letter $\eta$ and the letter $n$ are both used to represent different things in the manuscript.

Please use equation numbers for all equations.

Throughout the text, try as much as possible to refer to the relevant equations and/or mathematical variables. That would be incredibly helpful for the reader to understand the methods more easily. For example, refer to Eq. 15 every time the "ACV estimator covariance" is mentioned, refer to L197 when mentioning sampling from the prior, refer to L227 when mentioning sampling from the posterior. And there are many more instances, which I will not enumerate here. But I encourage the authors to look for every instance where the reader would benefit from knowing clearly which quantity or equation a certain statement relates to.

II) Adding some intuitive explanations

Here and there, it would help to add a simple sentence to give a better intuition about some concepts. I provide a few examples here below. Again, this is not an exhaustive list. So, I encourage the authors to actively look for similar statements, equations, or paragraphs that could benefit from some intuitive explanations.

- Towards the beginning of the manuscript, please provide one short paragraph to explain what the statistics of interest are, and why they are uncertain. I believe that all readers might not intuitively understand the concept of variance of a variance.

- L228: Please add one or two sentences to explain that $g(\theta)$ can be computed without time stepping model solves, and why this is the case.

- L247: Please explain that $\Sigma_{\mathrm{post}}$ characterizes the balance between the prior uncertainty in the friction field estimate, and the model-observation mismatch weighted by the observational noise.

- L298: Why is this valid regardless of how truthfully $f_1$ approximates $f_0$?

- L322: What do the control variates represent?

- Figure 4: How do results between these different sample allocation strategies differ? For example, does one approach prioritize minimizing the diagonal entries of the ACV covariance, versus another better constraining the correlation between different models?

- Etc.

**Line-by-line comments**

○ General (1): The text would benefit from the use of many more commas. I encourage the authors to, at least, double the number of commas in the manuscript. The general writing level is good, so I have no doubt that the authors can find sentences that need (or would benefit) from commas.

○ General (2): The quality of the figures is low. Color scales should be more informative, units should be provided, span of y axes should be appropriate for the range of values shown, a scale in km should be added when showing Humboldt, labels should be added to colorbars, etc.

L4
Here and elsewhere, the term "accuracy" is used very loosely, and encompasses a wide range of concepts. When used to describe a model degree of fidelity, please always use "fidelity" since this is the technical term used for the name of the method (MFUQ). When describing the amount of variance, please rather use precision, which is mathematically the inverse of the variance.
L4
Replace ice sheet by glacier.
L5
The problem size is not "representative" of continental scale studies. Please use more careful wording.
L11
prediction should be plural.
L15
Add report after IPCC.
L15
Ice sheets are all land-based.
L26
Throughout the manuscript, affect should be used as a verb instead of effect.
L28
Replace inadequacy by uncertainty.
L31
Throughout the manuscript, there is confusion in the wording of "parameters" and "inputs". For example, both terms are used interchangeably to characterize the basal friction field. Please (i) always use the same term for a same meaning, and (ii) clearly define the difference between parameters and inputs in the Introduction.
L39
There are also methods that have been developed to reduce the problem dimensionality. Please cite Brinkerhoff (2022).
L48
When using the notion of MSE, it is important to clearly define with respect to which quantity the error is considered. In this study, I believe that the error is considered with respect to the expectation of the mass change from the high-fidelity model with respect to the posterior distribution of the basal friction field. I realize that this is not straightforward to include. But I recommend adding a couple of sentences to give the definition, and possibly explain its meaning.
L50
I think that the authors might not be aware of the study of Bulthuis et al. (2020). Please consider referring to it.
L61
I find the changes between past and present tense somewhat confusing. I recommend consistently using a single tense.
L85
Model simulations do not only capture the "melting", since they represent the dynamic response of the glacier as well. This should induce changes in the amount of ice flowing out of the simulated

domain.

L102-104

In this sentence, the summary of the Stokes and MOLHO models sound identical to me.

L113

Replace exorbitant by impractical.

L121

Provide a reference for $q = 1/3$.

L125

Define the $\|$ notation here.

L139

The $\psi$ term is already multiplied by $\mathbf{n}$ above, so this multiplication should not be included in the definition of $\psi$. Also, why is there an extra term $\rho g(s - z)$ in the boundary condition on $\Gamma_m$ here compared to the Stokes model?

L145

$\partial \Sigma$ is not defined.

Figure 2

Show the meshes side-by-side (+ all comments from General (2)).

L172

The statement "one of the largest sources of prediction uncertainty" should be quantified and referenced with a citation.

L180

"we set $\mu = 0$". I believe that this is only for the prior. It seems strange to me that the posterior is forced to have zero mean. Please specify.

L183

There is a switch from $C$ to $\Sigma$ without mentioning it. Specify that $\Sigma_{\text{prior}}$ is a covariance.

L185

Why is the source term only integrated over $\Gamma_g$ and not over $\Gamma_f$? I would assume that snow accumulation and surface melting should also be computed over the floating parts of the domain.

L199

Please specify "this Gaussian prior".

L199

Replace "on" by "constrained with".

L207

The authors sort of sweep under the rug the possible influence of ocean melt on their methods. Melt at the boundary can induce strong dynamical responses by a marine-terminating glacier. It can be expected that differences between models of different levels of fidelity would be exacerbated, potentially diminishing the advantages of the MFUQ. Please discuss this more thoroughly in the Discussion.

L213

Please clarify why this assumption is required in the procedure. I believe that it is needed to compute the $g(\theta)$ function represented by the Blatter-Pattyn flow model. And that without this assumption, the PDE-constrained optimization cannot be solved.

L217

"However, such approaches ignore the uncertainty in the model parameters due to using a finite amount of noisy observational data". This statement is incorrect. Observational uncertainty can be incorporated in cost functions. See for example Eq. (1) from Goldberg (2015).

L222

"the likelihood distribution": the likelihood is a function, not a distribution.

L232

Please add a justification for this choice of $\alpha$.

L233

Please specify "samples from the posterior of $\log(\beta)$".

L249

Again, this statement is likely not obvious to most readers. At first sight, the computation that is referred to here is a simple addition of two matrices ($H_{\mathrm{MAP}} + \Sigma_{\mathrm{prior}}^{-1}$). Thus, a brief sentence to explain why this is intractable would be beneficial.

L254 and 255

Replace ice sheet by glacier.

L263

What do the authors mean by "robust"?

L264

"three-step"

L265

The $m$ superscript should be $n$ (which would preferably be another letter than $n$, see Major comment).

L266

Specify "basal friction field".

L278

"The bias term of the MSE (11) is caused by using a numerical model, with inadequacy and discretization errors, to compute the mass change." Here also, I ask for clarification: bias with respect to what? If it is with respect to observations, then observational uncertainty should also be discussed. If it is with respect to the highest-fidelity model, then the latter is also a "numerical model", and the sentence is inappropriate. If it is with respect to the unknown true dynamical behavior of Humboldt glacier, then there is a philosophical question of how to compute a mean squared error with respect to a quantity that cannot be known.

L287

Typo estimated.

L296

Two-model

L316

QoI is not defined.

L324

Concerning $\Theta_\alpha^* \cup \Theta_\beta \neq \emptyset$, (i) I believe that $\cup$ should be $\cap$, (ii) I believe that "for $\alpha \neq \beta$" should be specified.

Eq. (16)

Is $\mathrm{Cov}[\mathbf{Q}_0, \mathbf{\Delta}_0](\Theta_{\mathrm{ACV}})$ a covariance matrix? If so, it should be symmetric. However, the (0,1) and (1,0) entries of the right-hand-side seem different to me. Please explain.

L351

"following standard practice": provide citation.

L352

Please add an additional explanatory statement, for example: This involves computing the high-fidelity and all the low-fidelity models for the same set of samples $\Theta_{\mathrm{pilot}}$.

L360

Specify "introduce sampling errors".

L367

I believe that the same should be specified for $\alpha \cup \beta^*$ and $\alpha \cap \beta^*$.

L382

There is no verb in this sentence.

L391

Please add an additional explanatory statement, for example: This can happen if a subset of the low-fidelity models correlate much better with the high-fidelity model than the rest of the low-fidelity models, for example.

L394

I think this should be estimator types.

L394

"model models subsets" is either a typo, or very confusing language.

L402

was should be were.

L413

ice-sheet should be glacier.

L415

Typo an an.

L424

Specify MALI ice-sheet code with the Blatter-Pattyn flow model.

Figure 9

I provide here a concrete example of how to help the reader navigate through the technical details of the study. The caption should specify: "... MAP point ($\theta_{\mathrm{MAP}}$ in Eq. (9)) ... prior variance ($\Sigma_{\mathrm{prior}}$ in Eq. (xxx)) ... posterior variance ($\Sigma_{\mathrm{post}}$ in Eq. (xxx))". Using more such links between text and mathematics would really help reading the study.

L437

"speeds up as it thins": I think that this statement is incorrect, although I see what the authors mean. A glacier does not speed up because of thinning. It speeds up because of increasing surface slope, caused by enhanced thinning at the front. Also, the inverted relation holds: as a glacier speeds up, it discharges more ice into the ocean, leading to thinning.

Remark 5.2

I believe that this is an important scientific aspect, which is also somewhat swept under the rug. In their results, the authors demonstrate that the simulated mass change is sensitive to high-frequency variability in the basal friction field. As such, the interpolation method from fine to coarse meshes is potentially very influential. Which interpolation method has been used here? If it is simple linear interpolation, then all the high-frequency variability will be smoothed out. This would affect the behavior of low-fidelity models with coarser meshes. I recommend that the authors try interpolation methods that better preserve high-frequency variability (nearest neighbor, or maybe polynomial interpolation) and evaluate the impacts on their results.

L458

"significant differences": the word significant is misused here, because no statistical test has been performed. If a statistical test has been performed, please specify which one, and provide p-values. Furthermore, by eye, the differences do not seem very large in Figure 8 compared to the standard deviations. However, this is difficult to say because of the terrible choice of y-axes span in Figure 8, which should be changed.

L460

The meaning of accuracy is not clear here (see comment on L4).

L476

Provide a citation to support this statement.

L477

Please quantify "the error introduced".

L477

"not insignificant": this wording is misused here, because no statistical test has been performed. If a statistical test has been performed, please specify which one, and provide p-values.

L479

Please specify the number of realizations per bootstrap. From the rest of the text, I believe that it is 20 realizations per bootstrap samples, but this should be clarified explicitly.

Figure 10 (1)

I am puzzled by the very high upper bound on the variance reduction of the variance. In the ratio, the SFMC variance estimator is the denominator, which should therefore be the same for all the bootstrap samples. As such, the very high upper bound is caused by an unrealistically low estimated ACV variance via Eq. (15). This leads me to the question: is the approximation on pilot samples via Eqs. (15,20) unstable when using bootstrap with replacement? In any case, please provide an explanation about the very high value of the 95% quantile.

Figure 10 (2)

It is not immediately clear why a same model combination would give different estimates of the variance reduction, since the ice sheet models are deterministic. If I understand correctly, some of this variability comes from the random bootstrapping within the pilot samples, and some of the variability comes from the ACV estimator selected (MLMC, MFMC, ACVMF). Is it possible to quantify how these two sources of variability compare? And in turn, is it possible to quantify how much of the boxplot spread in Fig. 10a is due to these two factors versus the fact that different subsets of low-fidelity models have been selected?

L489

Specify subsets of model combinations.

L491

the original 20 pilot samples are used.

L491

Specify were determined useful to include for reducing. (Probably that individually, all the models would be useful. But they are not relative to including other better-correlated or computationally-cheaper models.)

L496-499

I could not understand the end of this paragraph. It would be helpful if the authors defined the notion of hierarchical relationship.

Figure 11

Please specify the number of samples for each case (2, 3, and 4 models).

L520

Again, the meaning of "accurate" is not well-defined.

L520

"even the smallest variance reduction was greater than 20". This is not what is shown in Fig. 11. Certainly not for the cases of 2 and 3 models. And for the case of 4 models, it seems to me that even the 5% quantile is below 20, suggesting that the smallest value is definitely smaller than 20.

L522

Replace that by which (with a comma, see General comment (1)).

L523

The three models listed do not include $MOLHO^*_{1km,9days}$. As such, I believe that it corresponds to the case "4 models" in Fig. 11. I find the discrepancy between the number of low-fidelity versus the total number of models confusing. Please use a consistent manner to quantify the number of models used.

L525

Please remind the readers where these numbers come from.

L527

I do not see any right or left panel.

L535

Please specify another estimator (i.e., MLMC or ACVMF).

In Discussion

This question relates to my curiosity concerning the complementarity between this method and stochastic ice sheet modeling. Here, the MFUQ samples uncertainty from a single time-constant uncertain input. In contrast, stochastic modeling (e.g., Verjans et al., 2022) samples uncertainties between multiple correlated uncertain inputs, and at different time steps (for example SMB variability in time is prescribed as stochastic). However, since the statistical properties of the time-varying stochastic inputs (i.e., the auto-correlation, the covariance structure and the mean of each stochastic input) can be specified a priori, I suppose that, in theory, the MFUQ method could be applied. But I wonder if this is practically feasible. I think that the Discussion would benefit from a short paragraph about this point.

L567

Appendix B.

Figure 13 (1)

Changing the color scale here is absolutely necessary.

Figure 13 (2)

If I understand correctly, the basal friction field should be model-independent. The differences only stem from the interpolation method. This should be specified in the caption. Furthermore, this Figure seems to confirm my comment about Remark 5.2.

L589

"variance" should be standard deviation here, since Gigaton units are specified.

L590

"significance": the word significant is misused here, because no statistical test has been performed. If a statistical test has been performed, please specify which one, and provide p-values. Furthermore, even the meaning of "the significance of these numbers" is not clear to me.

L593

In this study, the basal friction field distribution was derived assuming that all other variables were perfectly known. In reality, different sources of uncertainty can mix. Please cite Gudmundsson and Raymond (2008) and add one or two sentences about this to the Discussion.

L614

Please mention here that this study explores the use of MFUQ for low-order moments only. One can wonder if this method can be used for statistics such as skewness or quantiles in the tails of the distribution. This can be particularly important for evaluating the response to an input that could introduce instabilities and feedback mechanisms in the system, such as ocean or SMB forcing.

L616

Here, and in many other instances, the authors insist about the fact that MFUQ can be used at continental scale to estimate uncertainty on ice-sheet mass change statistics. However, such a statement is not well-supported by their results. Just looking at the results, one can argue that the MFUQ framework presented here requires 36 CPU days for a single glacier. Scaling this linearly to the Greenland ice sheet results in $\mathcal{O}(1-10)$ years of computation. Thus, there should be a slightly more in-depth explanation of why MFUQ is applicable for studies at the ice-sheet-scale.

L618

"substantially": please quantify and provide a citation.

L638

I do not understand the underlying meaning of this sentence. Please expand or remove it.

L641

Antarctica and Greenland.

L661

Again, the meaning of accurate is unclear here. It would be more correct to explain that the approximation level depends on the variance retained in the truncation.

L668

Please define $K$ here as well. Otherwise, the reader needs to go back to the main text.

L674

I do not see why the representation is "bi-Laplacian". I wonder if this term is not inadvertently misused here. Could this please be clarified? I believe that applying the Laplace approximation has no link with the bi-Laplacian operator, but sorry if I am misunderstanding here.

L685

Typo: in this study

L686

Typo: modes

L700

I believe that MF estimator should be ACV estimator

L701

I believe that MFUQ estimator should be ACV estimator

L702

This should be: The mean and variance bootstrapped (...).

L706

This should be: the uncertainty in the mean mass change (...)

L706-708

Please refer to Figure B5.

**References**

Douglas J Brinkerhoff. Variational inference at glacier scale. *Journal of Computational Physics*, 459:111095, 2022.

Kevin Bulthuis, Frank Pattyn, and Maarten Arnst. A multifidelity quantile-based approach for confidence sets of random excursion sets with application to ice-sheet dynamics. *SIAM/ASA Journal on Uncertainty Quantification*, 8(3):860–890, 2020.

DN Goldberg. Committed retreat of smith, pope, and kohler glaciers over the next 30 years inferred by transient model calibration. *The Cryosphere*, 9(6):2429–2446, 2015.

G Hilmar Gudmundsson and Melanie Raymond. On the limit to resolution and information on basal properties obtainable from surface data on ice streams. *The Cryosphere*, 2(2):167–178, 2008.

Vincent Verjans, Alexander A Robel, Helene Seroussi, Lizz Ultee, and Andrew F Thompson. The stochastic ice-sheet and sea-level system model v1. 0 (stissm v1. 0). *Geoscientific Model Development*, 15(22):8269–8293, 2022.

---

## Referee Comment (RC3)

The manuscript, An evaluation of multi-fidelity methods for quantifying uncertainty in projections of ice-sheet mass change by Jakeman et al, uses a new computational approach to determining the posterior uncertainty of ice mass change in a glacier forecast conditioned on observational data and uncertainty. The main contribution is a Multi Fidelity Uncertainty Quantification (MFUQ) scheme which samples from a probability distribution (see below) and provides an inexpensive means of Monte Carlo variance reduction in the calculated statistics that requires far less simulation time. This is achieved through generating ensembles from models that are of lower fidelity (coarser resolution / longer time steps) whose dependencies on the input parameters are similar. The probability distribution which is sampled -- that of the sliding parameter conditioned on observations and model physics -- would be too expensive to find via Monte Carlo methods. Rather, a method introduced by others in the literature -- which approximates this posterior as Gaussian and finds a low rank approximation to the inverse covariance matrix to make the problem tractable -- is used.

The methodology introduced in the paper – the MFUQ scheme – is fairly well described and seems quite useful, and its results deserve to be shown.

However, there are a number of major issues I have with the manuscript. Aside from a number of writing issues, such as inconsistent statements and introducing of terms and symbols without definition or explanation (see specific comments), I feel that the messaging of the paper in the introduction is not in line with what the authors have actually done. Furthermore they have downplayed or overlooked recent works in the literature – works which, in some cases, bring the methodology of this study into question. I will highlight these in general comments below.

Finally I should point out first though that it monte carlo methods are not my area of expertise. I have some specific comments about certain things that looked as thought they might be typos or need more explanation. Largely however I do not have much to say about the actual MFUQ methodology and its presentation, and I hope that other referees can assess it better.

**General Comments.**

1. The paper sets out to deal with parametric uncertainty, which is the case. But the introduction is written in a way that makes it seem that MFUQ is used to solve the "full" problem – that is, quantifying the probability density of mass change conditioned on the model and observations, which can be termed $p\,(Q\,|\,m,\,U)$ where $Q$ is the mass change, $m$ is the model and $U$ is the observations. But in truth a different method (Hessian-based) was used to find the posterior density of the frictional field $\theta$, and then this was sampled from to find the posterior of Q i.e. $p\,(Q\,|\,m,\,\theta)\,p\,(\theta\,|\,m,\,U)$ -- and $p\,(Q\,|\,\theta)$ is the only component being determined by MFUQ.

I think this could be potentially very misleading and give the impression that MFUQ is capable of the "full" problem when from the results of the paper it definitely is not. This is very important: given the newness of the fields of ice-sheet modelling and ice-sheet uncertainty quantification there is extensive misunderstanding about which problems can be tackled by sampling methods and which require alternative methods. Although this is somewhat covered in lines 61-71 of the manuscript, the passage requires familiarity with the field and with both MC and Hessian-based UQ. It needs to be much more clear – with mathematical formality – which distribution is being quantified using MFUQ.

2. The manuscript is also misleading about contributions in this paper versus in the literature. Specific examples are given below, but the manuscript does not acknowledge previous authors' attempts to quantify the uncertainty of high-dimensional parametric uncertainty. In particular, a recent paper in The Cryosphere (Recinos et al, 2023, hereby shortened as BR23) has been overlooked. The authors can certainly be forgiven for this of course as the paper came out only last year, but it is extremely relevant to many of the assumptions and calculations within the manuscript (and is mentioned extensively in the specific comments below). Additionally, based on this paper there are several assumptions and/or approximations that give me serious reservations about this paper's results – these are easily identifiable in the specific comments where BR23 is mentioned.

3. The underlying premise of the paper is that, given a Hessian-based approximation of the posterior parameter density has *already* been carried out, "traditional" means of sampling from this posterior density is too expensive. But another such approach – using the *linearization* of the mass change model $f(\theta)$ (using either Automatic Differentiation or some other form of differentiation) to project the posterior uncertainty of $\theta$ onto the quantity of interest – exists, and is not at all mentioned. Playing devil's advocate, such an approach assumes near-linearity of $f(\theta)$, but linearity has already been assumed in the posterior calculation of $\theta$. Moreover at least two prior papers – Isaac et al (2015) and BR23 – have used this method (see eq. 24 of Isaac et al 2015, or eq. 15 of BR23), and the latter comprehensively tested the linearity assumption. Given this, I would expect acknowledgement of this very relevant and related approach, its drawbacks and benefits, and fit (or lack thereof) to the current problem.

**Specific Comments.**

L23-25. This is a good outlay of the different sources of uncertainty. What is missing is a definitive statement that the only type of uncertainty being quantified in this paper is parametric uncertainty.

L26-27. "but the impact of discretization errors has not been explicitly considered with other sources of uncertainty". And it has not in this study either, right? As I understand it the MFUQ scheme is solely to estimate parameter uncertainty of the 1km, 9-day MOLHO model – it did not quantify disc. uncertainty despite using different discretizations.

L37-41. This is a good place to cite works such as Isaac et al 2015 (and various papers by Noemi Petra e.g. Petra et al 2013), and BR23.

L60-61. As noted above, quantifying the impact of a high-dimensional parameterization of basal friction on long-term projections is not novel (cf. BR23 – unless you are distinctly saying that 40 years is not long-term and 80 years is!)

L62-64. As noted above, Isaac et al, whose methodology you cite and use, arguably did this.

L66. Im not sure why you include Isaac 2015 in a list of papers using low-dimensional parameterisations – they used $O(10^6)$ parameters in their basal sliding parameterization.

Fig 2, 3, 5, 6, 7, and 13: you need to show the coordinate axes in all visualisations of the model domain – and there should be one figure showing the placement of Humboldt in Greenland.

L181: "covariance" – prior or posterior?

L190-193. I have deep concerns about your parameter choices. Firstly, what is the pointwise variance? Secondly, how did you arrive at this correlation length as suitable – on what basis? I do not see any physical reasoning leading to it. You are saying that the data essentially does not need to constrain variability on a scale smaller than this, which I don't think is an accurate statement. BR23 chose far smaller autocorrelations (~3km) using some degree of physical inference, and moreover showed that it was necessary to give reasonable values of posterior uncertainty (see comment on TABLE 1 regarding this assessment), and it is possible that in choosing such large numbers you are making the posterior uncertainty artificially small by choosing an overly-informative prior. This may be why you only needed < 1000 eigenvalues to represent the posterior as shown in the appendix. (see BR23 for details.)

L205. How did you generate mass balance? Did you run a regional climate model that incorporates firn and snow processes? If so, say so. Did you use a parameterization? If so, state it and the source.

L231. On what basis do you assume they are uncorrelated? The fact that the products are not posted with spatial correlations of error is not a reason – this is simply too difficult for them to calculate. Please highlight this, and state what the consequences of such an assumption could be for estimating posterior uncertainty.

Section 4: in general I think this section should be read over very carefully to look for typos and variables introduced without definition. Ill mention several below but these sections (the ones that I read closely) seem to have been written hastily.

L263, mean $Q^\mu$: mean of what?? And what is Q? and are these "true" statistics or estimators since they have no subscript?

L265. Try to be consistent with tense throughout, and definitely within a sentence: "The second step simulate**s** the model at each realization ... and comput**ed** the mass change.."

L271: "Any MC estimator Q" – do you mean $Q_\alpha^\mu$ or $Q_\alpha^{\sigma^{\wedge}2}$, or both or neither?

Eq 11 – can you show how this is derived? At first glance it looked similar to the identity $E[(X-E[X])^2] = E[X^2]-(E[X])^2$ but I could not derive it using similar reasoning.

L279 did you mean MSE (II), rather than MSE (11)?

L279: I don't believe that all of these sources of uncertainty go into the bias term. My interpretation is that, for the purpose of your MFUQ, you are given a density of $\theta$ arising from the Isaac methodology. You then have a deterministic function $f_\alpha(X)$ which is given by your high fidelity model and its discretization, and is therefore deterministic. You are seeking properties of the probability distribution *induced by* $f_\alpha$ and the only actual uncertainty is how fast the MC converges. Model uncertainty and discretization uncertainty, while very real, are not accounted for in such a calculation.

L280 what does MSE (10) mean?

L280 ensures, for any set of model input samples,

First eq in 4.2.1 (not numbered) – is the 2nd term in brackets not divided by N1?

L316 – QoI not defined previously.

L324 – for the union of these sets to be null, both need to be null. Should it be an intersection symbol?

Eq 18. You seem to be estimating these statistics using straightforward (Naïve) MC. Why is this OK given the whole thrust of your study is that MC is too expensive to apply to the statistics of the ice model?

L418-422. State # of elements In models

L424 in the 1st para of 2.4 you state you use FEniCS. MALI is a C++ model with Fortran libraries and not, to my knowledge, written with fenics. **Which model(s) did you use???**

Fig 10 – I might be misunderstanding the methods but shouldn't there be units??

Table 1. This value is presented without validation. It is possible to do a "sanity check". BR23 use 2 essentially independent measurements of velocity (ITS_LIVE and MEaSUREs) to invert for parameters and simulate mass loss. If the difference seen is of almost negligible probability under the calculated posterior for mass loss – then there must be an issue with the calculated posterior. You are capable of doing this as well…

L550. "the SSA model was not.." can you provide an example or evidence of this?

L565. Im confused – I thought that the MFUQ was needed as you are sampling from a distribution of ~600 dimensions (the number of Eigenvals retained in the Hessian based UQ). If you have only 10 dimensions can you not use standard (naïve) MC?

L567: Appendix B

References

Recinos, B., Goldberg, D., Maddison, J. R., and Todd, J.: A framework for time-dependent ice sheet uncertainty quantification, applied to three West Antarctic ice streams, The Cryosphere, 17, 4241–4266, https://doi.org/10.5194/tc-17-4241-2023, 2023.

Koziol, C. P., Todd, J. A., Goldberg, D. N., and Maddison, J. R.: fenics_ice 1.0: a framework for quantifying initialization uncertainty for time-dependent ice sheet models, Geosci. Model Dev., 14, 5843–5861, https://doi.org/10.5194/gmd-14-5843-2021, 2021.

Petra, Noemi, et al. "A computational framework for infinite-dimensional Bayesian inverse problems, Part II: Stochastic Newton MCMC with application to ice sheet flow inverse problems." SIAM Journal on Scientific Computing 36.4 (2014): A1525-A1555.

---

## Author Comment (AC1)

**1 Summary of comments and our proposed changes**

In this document, we respond to every comment from the three reviewers. The reviews were extremely thorough and valuable, and addressing the comments given has substantially improved our manuscript. Below we address all 187 comments. However, first we summarize here what we inferred were the main concerns raised by the reviewers.

Please note that while this stage of the review process did not require that we prepare a new revision, we choose to do so anyway to ensure that sufficient thought was given to each comment. Consequently, our responses below use the past tense. Also note that, we changed the acronym used in this paper from MFUQ to MFSE, which refers to multi-fidelity statistical estimation. We believe this term better captures the algorithms used in this paper than term multi-fidelity uncertainty quantification (MFUQ), which is broader and could be perceived to include methods such as multi-fidelity surrogate modeling. Ideally, we would use the term multi-fidelity Monte Carlo (MFMC), instead of either of the two aforementioned terms. However, MFMC is used in the literature to to a particular type of MFSE algorithm.

First, all three reviewers stated that the Section 4, which introduced multi-fidelity methods, needed to be rewritten more clearly. We were happy to do this, as we want to make the paper as easily accessible as possible. We also thank the reviewers for their constructive advice on how to improve our exposition.

Second, the reviewers asked to more clearly and precisely establish the novel contributions of our work, especially compared to a recent paper that we were unaware of. Additionally, we were asked to add further discussion of the limitations of our study to the paper. We have rewritten the introduction and discussion sections to address this concern.

Third, but not least, the third reviewer (listed in this document) asked how MFSE compares to a recent method introduced in a paper referred to as BR23. The method in BR23 linearized the parameter-to-QoI map of an ice-sheet model to computationally efficiently estimate the distribution of the QoI conditioned on observations. Moreover, the reviewer also remarked that our procedure for selecting the prior distribution of our uncertain parameters could be improved by pointing to the tuning method proposed in BR23. Specifically, the reviewer correctly pointed out that not using a rigorous method for tuning the hyper-parameters of the prior distribution, such as that proposed in BR23, can lead to uncertainty being underestimated. However, the method requires automatic differentiation to linearize the parameter-to-QoI map, which our codes (and many others) do not support, so we were unable to make a comparison or adopt the tuning procedure proposed in the cited paper. Secondly, linearizing the parameter-to-QoI map introduces an error when the map is nonlinear. As the cited paper points out this, error depends on the strength of the nonlinearity in the map. As we could not construct the linearization used in BR23, we could not compute the size of this error. However, we have included plots in the paper and response that demonstrate the parameter-to-QoI map is nonlinear. Moreover, we now remark in the paper that if linearization is possible, then it will likely be a computationally efficient and accurate low-fidelity model that could potentially improve the performance of MSE further.

In the following we list the reviewers comments with *"black italic"* font. We list text from the original document with *"blue italic"*. Additionally, we include new text added to the revised document in red. Lastly, we made additional edits to the paper to improve readability, but we only highlighted such changes if they were in response to a reviewer comment or change the narrative substantially.

**1.1 Miscellaneous**

Please note that there was an issue with section numbering in Section 4 which we have now corrected. In the revised document we created a new subsection 4.4 called *"Computational considerations for multi-fidelity uncertainty quantification"* which discusses the important computational aspects of the exploration and exploitation phases of multi-fidelity UQ. Section 4.4 now includes subsections 4.2.3, 4.2.4, 4.25, 4.2.6 in the original submission. Moreover, Remark 4.1 in the original submission has now been moved to form the basis of the introduction in Section 4.4.

**2 Reviewer 1 (Douglas Brinkerhoff)**

**2.1 General comments**

*"In this manuscript, Jakeman et al. present the application of multi-fidelity uncertainty quantification to accelerate – and hopefully provide more accurate – estimates of first- and second-order ensemble statistics. They begin by describing two approximations to the Stokes' equations, which trade solution expressivity for computational expense and serve as the basis for their multifidelity methods. They next describe a mechanism by which to characterize an approximate posterior distribution (based on a low-rank Laplace approximation) over basal traction conditioned on surface velocity observation, which serves as the source of samples for Monte Carlo sampling of ice volume evolution, the primary quantity of interest in this work. The primary methodological advance is the introduction of an adaptive control variate (ACV) estimator for the mean and variance of mass change after approximately a century of ice evolution. This estimator leverages correlations between so-called low- and high-fidelity models (which have different computational expense) to effectively reduce the error in Monte Carlo estimates of ice volume change relative to predictions made using a limited number of high-fidelity model predictions on its own. They present this method for a single high and low-fidelity model, and then extend the analysis to the case where there exists one high-fidelity model and a hierarchy of multiple low-fidelity models. Such methods require the establishment of statistical relationships between the low and high-fidelity models, and also careful selection of the number of samples evaluated for each constituent model: the authors carefully present strategies for these tasks within the framework of a fixed computational budget, and explore the implications of these strategies when they are better informed by empirical analysis than theory. The manuscript applies these methods to the Humboldt Glacier basin of Northwest Greenland and show that the present methods can be used to perform effective uncertainty quantification, at least over the limited subset of uncertain parameters that the authors' consider. This work is an important and timely contribution to the growing effort towards robust uncertainty quantification in ice sheet modeling. I have no major objections to the scientific content of this work, which I find to be well-motivated and defensible. I do think that the work relies on language and a presentation style that will be challenging for many readers, particularly those without a specialized statistical background. As a general comment, I would encourage the authors to try to provide more intuition and plain-language summaries, particularly in Section 4. I provide more specific examples, alongside other detailed comments below."*

**2.2 Specific comments**

1. *"L26 Here and elsewhere, 'effects' should be 'affects'".*

   Fixed.

2. *"L97 Should $H(x, y, z)$ be $H(x, y, t)$?"*

Yes. We fixed this typo.

3. *"L98 The MOLHO (or the Blatter-Pattyn approximation) doesn't neglect vertical velocity, it is just eliminated from the system of equations via mass conservation and the assumption of hydrostatic pressure. It can always be determined a posteriori from the horizontal velocity components."*

We changed the sentence *" In contrast, using the observation that ice-sheets are typically shallow, i.e. their horizontal extent is much greater than their thickness, the MOLHO model neglects the vertical velocity $w$ and only simulates the horizontal velocities $u(x, y, z, t), v(x, y, z, t)$ but still as functions of the three spatial coordinates"* to In contrast, the MOLHO model makes simplifications based on the observation that ice-sheets are typically shallow, i.e. their horizontal extent is much greater than their thickness. These simplifications lead to a model that does not explicitly estimate the vertical velocity $w$ and only simulates the horizontal velocities $u(x, y, z, t)$, $v(x, y, z, t)$ as functions of the three spatial coordinates.

4. *"I don't think Dukowicz is the best reference here. Pattyn (2003) is more commonly cited, or if the preference is for something that clearly describes the hierarchy of approximations, Schoof and Hindmarsh (2009)."*

We have added a citation to Schoof and Hindmarsh (2009).

5. *"L137 I think that citet should be used rather than citep here."*

We now use citet.

6. *"L152 The discretization of the continuity equation is non-trivial and should be described here. How was it stabilized? How was positivity (even in the absence of negative forcing) ensured?"*

We added the following details:

Specifically, the continuity equation was discretized with nodal finite elements, using streamline upwind stabilization. Additionally, the advection term was integrated by part and the thickness was treated implicitly. Using this time evolution process, we did not observe any numerical instabilities when using the time-step sizes adopted in this study.

7. *"L172 Who specifically considers friction to be such a large source of uncertainty? Plenty of recent work has shown that forcing terms are the most important uncertainty sources, particularly at long time scales. I don't have a problem with focusing on traction here, but I think it is important to contextualize this choice a little bit more fully."*

We agree that forcing terms are large sources of uncertainty. Carr et al. (2024) in particular have recently made this argument. However, basal sliding is widely acknowledged to also be a large source of uncertainty (see e.g., Nias et al. (2018); Joughin et al. (2019); Brondex et al. (2019); Åkesson et al. (2021); Hillebrand et al. (2022)), especially when using model configurations with active calving, which Carr et al. (2024) ignored. While we also do not account for active calving except to prevent advance, our methodology could be extended to situations that include physically based calving laws.

We changed the statement *"While all sources of uncertainty may significantly impact predictions of mass change from ice sheets, this study focused on quantifying uncertainty due to the unknown basal friction, which is considered one of the largest sources of prediction uncertainty. This singular focus was made to improve our ability to assess whether MSE is useful for ice-sheet modeling for a very high-dimensional source of uncertainty, which cannot be*

*tractably tackled using most existing UQ methods. This ensures that the conclusions drawn by our study can be plausibly extended to studies considering additional sources of uncertainty."* to While all these sources of parametric uncertainty may significantly impact predictions of mass change from ice sheets, this study focused on quantifying uncertainty due to unknown basal friction, which is considered one of the largest sources of prediction uncertainty after future environmental forcing (Nias et al., 2018; Joughin et al., 2019; Brondex et al., 2019; Åkesson et al., 2021; Hillebrand et al., 2022; Nias et al.,190 2023). This singular focus was made to improve our ability to assess whether MFSE is useful for quantifying uncertain in ice-sheet modeling with high-dimensional parameter uncertainty, which most existing UQ methods cannot tractably address. By doing so, we ensured that the conclusions drawn by our study can be plausibly extended to studies considering additional sources of uncertainty.

8. *"L180 I think that the community is using the term 'Gaussian process' with some frequency now, so it would be good to at least mention that as a name for what is going on here (and a reference to, say Rasmussen and Williams (2006))."* We did not add this or a similar citation. We want to avoid using the term Gaussian process because the method we used to generate the Gaussian random field, which the reviewer is referring to, and condition it on data differs from the approach used by Gaussian process used in machine learning, e.g. (Rasmussen and Williams, 2006).

9. *"L212 For what it's worth, it's a stretch to call BedMachine 'data' – it is the result of a PDE-constrained optimization scheme that relies on assumptions of climate, smoothness, and a variety of other things. Again, nothing different needs to be done, but it is important to state that this inferred geometry is assumed error-free."*

   We clarified that we assumed the geometry was error-free.

10. *"L232 This scaling term is less mysterious when reported with units 'number of observations per area'."*

    We think that the "scaling term" the reviewer is referring to is the inverse of our coefficient $\alpha$ in the original submission. In our case, the observations are given as a spatial field, not as a finite number of data points, therefore $\alpha$ has the units of area, which is consistent with the reported units of $km^2$. In an attempt to make the scaling coefficient more intuitive, we changed the definition of $\alpha$ and considered its inverse, which is more inline with the typical definition of scaling terms in objective functionals. We also expanded Remark 5.1 to discuss how the scaling parameter $\alpha$ has been selected.

11. *"L263–267 Here and elsewhere, please be sure to use a consistent tense. This switches from present to past inside a sentence."*

    We have corrected tense here and throughout the document.

12. *"L298 Should these sums have $N-1$ in front of them?"* Yes. We corrected this mistake.

13. *"L298 For the inner sum, please use a different index variable than n."*

    We now use the symbol $j$.

14. *"L298 Is the optimization of $\eta$ mandatory or does this work with arbitrary $\eta$? What is the objective that is optimized?"*

    The optimization of $\eta$ is not mandatory. Any value can be used, indeed MLMC uses $\eta = -1$, however using a non-optimized value of $\eta$ can substantially degrade the accuracy of the

estimator. We now mention that $\eta$ can be set a priori. To see the exact changes refer to the added text in red in response to comment 15.

Also note that, in the original manuscript we stated that *"the MSE of the ACV estimator can be minimized by optimizing the determinant of the estimator covariance matrix"*. For a fixed sample size, the optimization of $\eta$ is analytical as reported in Equation (14) of the original manuscript.

15. *"L298 The reader would benefit from a description of what this equation means and some intuition of why this works. It appears to be that the low-fidelity terms yield a correction to the high fidelity statistic, but it is somewhat surprising that this doesn't need to include any explicitly quantified relationship between the two models. It would also be helpful to emphasize that the $\Theta_0$ and $\Theta_1$ can have different set sizes."*

We added the following text to the revised document to provide the intuition requested and discuss the sizes of the sample sets $\Theta_0$ and $\Theta_1$.

[revised manuscript text omitted]

17. *"L328 'statistics' → 'statistic'."*

Fixed.

18. *"Eq. 14 As before, are there alternatives to using this value for $\eta$?"*

See responses 14 and 15.

19. *"Eq. 16 Split this into two equations, and add matrix sizes for each."*

Fixed.

20. *"L352 I'm not sure I understand this sentence."*

We changed this sentence when responding to the next comment.

21. *"L356 I think it would be better to include more detail about how these expressions are used to compute Eq. 15 than just referencing Dixon (2023). Otherwise, it sort of feels like a lot of space gets used describing Eqs. 16 and 17, but they never really go anywhere."*

We removed equation 16 and 17, as we agree that they did not really build intuition. Moreover, including the expressions in Dixon et al (2023) are extremely complicated and would likely turn off all but the most mathematically inclined reader.

22. *"Eq. 20 Why is it the case that minimizing the determinant of the covariance determines an optimial sampling strategy?"*

Because we are computing a vector valued statistic, comprised of the mean and variance of the mass loss at the final time, the MSE error of the statistic is a matrix. The determinant was first proposed in as a scalar metric that quantifies the error of a vector-valued estimator. It is likely possible to use alternative measures, such as the trace to quantify error, however to date only the determinant has been used. Indeed, most MSE literature only focuses on estimating a single statistic of a scalar function, in which case the trace and the determinant are equal.

We now state in Section 4.4.2: Unfortunately, a tractable algorithm for solving Eq. (29) has not yet been developed, largely due to the extremely high number of possible sample allocations in the set $\mathbb{A}$. Consequently, various ACV estimators have been derived in the literature that simplify the optimization problem, by specifying what we call the sample structure $\mathcal{T}$, which restricts how samples are shared between the sets $\Theta_\alpha, \Theta_\alpha^*$. For example, optimizing the estimator variance, Eq. (23), of a two model MFMC (Peherstorfer et al., 2016) mean estimator, Eq. (21), requires solving

$$\min_{N_0, N_1} N_0^{-1} \mathbb{V}[f_0] \left( 1 - \frac{N_1 - N_0}{N_1} \mathbb{C}\text{orr}[f_0, f_1]^2 \right)$$

$$\text{s.t.} \quad N_0 w_0 + N_1 w_1 \leq W_{\max},$$

$$\mathcal{T} = \{ N_{0 \cap 1*} = N_0, N_{0 \cup 1*} = N_0, N_{0 \cap 1} = N_0, N_{0 \cup 1} = N_1, N_{1* \cap 1} = N_0, N_{1* \cup 1} = N_1 \}.$$

Alternatively, minimizing the estimator variance of the two model MLMC (Giles, 2015) mean estimator requires solving

$$\min_{N_0, N_1} N_0^{-1} \mathbb{V}[f_1 - f_0] + (N_1 - N_0)^{-1} \mathbb{V}[f_1]$$

$$\text{s.t.} \quad N_0 w_0 + N_1 w_1 \leq W_{\max},$$

$$\mathcal{T} = \{ N_{0 \cap 1*} = N_0, N_{0 \cup 1*} = N_0, N_{0 \cap 1} = 0, N_{0 \cup 1} = N_1, N_{1* \cap 1} = 0, N_{1* \cup 1} = N_1 \}.$$

MLMC and MFMC employ sample structures $\mathcal{T}$ that simplify the general expression for the estimator covariance
$\mathbb{C}\text{ov}_\Theta[\boldsymbol{Q}_{\text{ACV}}, \boldsymbol{Q}_{\text{ACV}}]$ in Eq. (27). These simplifications were used to derive analytically solutions of the sample allocation optimization problem in Eq. (29) when estimating the mean, $\mathbb{E}_\Theta[f_0]$ in Eq. (16), for a scalar-valued model. However, the optimal sample allocation of MLMC and MFMC must be computed numerically when estimating other statistics, such as variance $\mathbb{V}_\Theta[f_0]$ in Eq. (17). Similarly, numerical optimization must be used to optimize the estimator covariance, $\mathbb{C}\text{ov}_\Theta[\boldsymbol{Q}_{\text{ACV}}, \boldsymbol{Q}_{\text{ACV}}]$ in Eq. (27), of most other ACV estimators, including the ACVMF and ACVIS (Gorodetsky et al., 2020), as well as their tunable generalizations (Bomarito et al., 2022).

23. *"Fig. 4 and accompanying text I don't think that the text does a sufficient job of describing the principles behind these different sampling strategies. Figure 4 tells me that $\Theta_0$ and $\Theta_1^*$ share their samples, and different schemes use entirely different or appended different samples for $\Theta_1$, but I cannot grasp from the text why this is significant. This needs to be motivated fully or de-emphasized and more carefully referenced."*

We removed the figure and instead now try to provide intuitive descriptions on why different sample structures target different relationships between models.

See our response to comment 16.

24. *"L394 'model' appears twice."*

    Fixed.

25. *"L433–439 Is there an argument that can be made here to reassure a reader that the observed changes are due to real climate/ice dynamic effects and not so-called 'transients' resulting from inconsistency between initial conditions, physics, and input fields? Fig. 7 (left) has some rather surprising high-frequency noise in the surface elevation change – it would be helpful to know where this comes from."*

    The high-frequency content is largely due to the oscillations in the posterior sample of the basal friction (Figure 6). We have added a similar statement to the document when discussing Figure 7.

26. *"L453 Just to clarify, was $\Theta_{pilot}$ shared across models, or were the samples different for each model?"*

    We now state:

    First, we evaluated each of our 13 models at the same 20 random pilot samples of the model inputs $\Theta_{\text{pilot}}$

27. *"L458 'significant' is an unfortunately subjective term here - to my eye, the differences in the heights of the referenced bars seems rather insignificant. Is it possible to elaborate on the meaning of 'significant' here, and why it should be viewed as such?"*

    We updated the figure to more clearly show the differences between the means and standard deviations of each model. We also now state:

    The middle and lower panels of Figure 8 show that the means and standard deviations of each model differ.

28. *"L467 This sentence is a bit challenging, with 5(!) nested prepositions and two uses of 'variance' each describing different things. I recognize that it is challenging to compactly describe the statistics of statistics, but is it possible to relax this sentence a bit?"*

    The previous text was *"Given estimates of the pilot statistics, (18) and (19), we used (20) to predict the determinant of the variance of the ACV estimator of the mean and variance of the mass change. "*

    we change this to We used Eq. (20), with estimates of the pilot statistics obtained using Eq. (18) and Eq. (19), to predict the determinant of the the ACV estimator covariance.

29. *"L476 This assertion is surprising to me, and a citation describing the assumption of pilot statistic exactness would be useful."*

    We have added a citation to (Peherstorfer et al., 2016) which was cited elsewhere in the paper for another reason. In the last paragraph from Page A3181 the authors state "We use the sample variances and the sample correlation coefficients to determine the number of model evaluations m and the coefficients $\alpha$. Table 2 compares sample variances and sample correlation coefficients computed from 10, 100, and 1000 samples. The different number of samples leads to different estimates."

    The authors also state that "the variations in the sample variances have only a minor effect on the coefficients $\alpha$" where $\alpha$ refers to the control variate weights denoted $\eta$ in our paper. However our results show that the impact of the size of the pilot is problem dependent.

30. *"L494 Is there an interpretation of why some models appear to be more informative than others, or is this just random chance? I can't identify a mechanism for why some low-fidelity models were chosen more frequently, but understanding that (or being able to predict it a priori) would be exceptionally useful."*

We included the following statement at the end of section 4:

Whether a model is useful for reducing the MSE error of a multi-fidelity estimator depends on the correlations between that model, the high-fidelity, and the other low-fidelity models. For toy parameterized PDE problems, such as the diffusion equation with an uncertain diffusion coefficient, theoretical convergence rates and theoretical estimates of computational costs can be used to rank models. However, for the models we used in this study, and likely many other ice-sheet studies, ordering models hierarchically, that is, by bias or correlation relative to the highest-fidelity model, before evaluating them is challenging. Indeed, the best model ensemble for multi-fidelity UQ may not be hierarchical (see Gorodetsky, 2020). Yet, estimators such as MLMC and MFMC only work well on model hierarchies. Consequently, having a practical approach for learning the best model ensemble is needed. Yet, to date this issue has received little attention in the multi-fidelity literature. Section 5 provides a sorely needed discussion of the impact of the pilot study on model selection and the error a multi-fidelity estimator.

31. *"L496 I don't understand what is meant by a 'hierarchical relationship' here."*

See response 30.

32. *"L525 I get where these numbers come from after some digging back through the other sections, but it would be helpful to remind the reader where each of the terms in the cost expression represent."*

We added the following to the text:

The cost of constructing our final estimator was equal to the sum of the pilot cost (197.13 hours) and the exploitation cost ($160 \times 4.18$) hours, which was approximately 36 days. The pilot cost was the sum of evaluating all 13 models on the initial 20 pilot samples and 8 models on an additional 10 pilot samples (see Section 5.2). The exploitation cost was fixed at the beginning of the study to the computational cost equivalent to evaluating the high-fidelity model 160 times, which takes a median time of 4.18 hour to simulate.

33. *"L536 Is this extreme asymmetry between the number of high- and low-fidelity model evaluations typical? I think that this is a significant and interesting result if the high-fidelity model is only really needed to, e.g. characterize the spatial variability of the mean solution, but the low-fidelity models are sufficient to characterize all of the uncertainty."*

We added the following statement to Section 5.5:

The allocation of the small number of samples to the highest-fidelity model is due to the extremely high-correlation between that model and the model $MOLHO_{1km,36days}$. This high-correlation suggests that the temporal discretization error of the highest-fidelity model is smaller than the spatial discretization error.

34. *"Table 1 I don't think this needs to be a table."*

We removed the table and now state the following in the main text:

The mean and standard deviation computed using the best ACV estimator were $-639.06 \pm 0.23$ and $17.68 \pm 6.67$, respectively.

35. *"L589 The variance doesn't have the same units as the mean, so I'm not sure what numbers I'm looking at. Is 17.68 the standard deviation?"*

You are right. We now state the values in question represent standard deviation.

**3 Reviewer 2 (Vincent Verjans)**

**3.1 General comments**

*"This study proposes a multi-model method for evaluating uncertainties in ice sheet model projections. This method uses models of different degrees of fidelity to simulate glacier mass change projections, therefore referred to as multi-fidelity uncertainty quantification (MFUQ). It exploits correlation between the different model realizations to approximate the statistics that would be obtained by the highest-fidelity model available, but at reduced computational cost. Here, the study focuses on uncertainty arising from the uncertain basal friction input field, and shows an application at Humboldt glacier, Greenland. Random samples of the basal friction field are drawn from a Laplace approximation of the posterior probability distribution, which is calibrated to match output from an ice flow model to the present-day Humboldt glacier configuration. The study then compares the MFUQ method with Monte Carlo sampling using the highest-fidelity model only, which is referred to as single-fidelity Monte Carlo (SFMC). Results show that, applied to this problem, MFUQ can serve to infer the mean and variance statistics with large computational savings compared to SFMC. The MFUQ procedure splits the computational burden by using only few high-fidelity model runs and a large number of lower-fidelity model runs, and then exploiting the correlation between both sets of runs. This study is a valuable contribution to the field of uncertainty quantification in ice sheet modeling. It demonstrates that combining multiple levels of model fidelity can serve to improve uncertainty estimates in useful quantities, which is an approach scarcely used in this field. The science presented in this study uses elaborate statistical techniques, which is a good thing. And I evaluate the scientific aspects of this study positively. However, I believe that major efforts should be made on two presentation aspects. First, more clarity is needed in the presentation of the MFUQ method. I needed to re-read and go back-and-forth between different sections multiple times to really un- derstand the procedure. Second, the authors should try to guide the reader in understanding the procedure, and to provide some intuitive explanations of the different steps in addition to the mathematical details. This latter aspect would better align with the readership of Earth System Dynamics, which is not primarily focused on methodological developments per se. I separate this re- view in one Major comment, focused on the most important clarifications required, and line-by-line comments focused on less important aspects that need elaboration, as well as on scientific aspects that could be slightly adjusted or more thoroughly explored. Line numbers (L) refer to lines in the preprint. Although my review insists a lot on presentation aspects, I find it also important that the science-related comments are addressed. I encourage the authors to revise their manuscript following comments from other reviewers and me. Given the strong scientific basis of this study, I am certain that a revised version of this manuscript will be a valuable contribution to the literature.*

*Major comment: mathematical presentation. There is no single specific aspect that makes the mathematical presentation unclear. Instead, it is the accumulation of various elements that renders understanding the methods challenging. I try to identify some of these elements here."*

We want to make the paper as easily accessible as possible and have rewritten the paper. Specific focus was given to improving clarity and providing intuition wherever possible.

**3.1.1 Equations should be better explained and without errors.**

36. *"In all equations with matrices, please provide explicitly the dimensions of the matrices involved. This would help to understand, for example, Eqs. 13, 16, 17. It would also be helpful to explicitly mention if a quantity is a scalar, vector, or matrix when it is used for the first time."*

    We added matrices sizes to all relevant equations in the revised document. To further improve clarity, we our revised paper uses bold italics to represent vectors and bold regular font for matrices. This is inline with the instructions on how to denote matrices and vectors n the journal's guide for authors. We apologize for missing these instructions in our previous submission.

37. *"In Eq. 12, the covariance term has twice the same argument, and can therefore not be a covariance. Furthermore, I am not convinced of the validity of the $Var(Q^{\sigma^2})$ formula, so please provide a reference and/or a detailed derivation in the response."*

    Eq 12 is a valid covariance. $\mathbb{C}\text{ov}\,[X,Y] = \mathbb{E}\,[(X - \mathbb{E}\,[X])(Y - \mathbb{E}\,[Y])]$ and $\mathbb{C}\text{ov}\,[X,X] = \mathbb{E}\,[(X - \mathbb{E}\,[X])(X - \mathbb{E}\,[X])] = \mathbb{V}\,[X]$ is just a special case that occurs when $X = Y$. We want to use the notation $\mathbb{C}\text{ov}\,[X,X]$ so to emphasize that the covariance is a matrix when the random variable $X$ is a vector. However since $X = (f_\alpha - \mathbb{E}\,[f_\alpha])^2$ is only a scalar we replaced $\mathbb{C}\text{ov}\,\left[(f_\alpha - \mathbb{E}\,[f_\alpha])^2, (f_\alpha - \mathbb{E}\,[f_\alpha])^2\right]$. We also added the following note after equation 15 that states:

    Note that, in (14) and (15), and the remainder of this paper we use $\mathbb{C}\text{ov}\,[X,X]$ as long hand for $\mathbb{V}\,[X]$ to emphasize that the covariance is a matrix when the random variable $X$ is a vector.

    A detailed derivation of the expression can be found in (Dixon et al., 2023). See equation 2.3 and the proof of proposition 3.4 on page 10 of their ARXIV manuscript. We added the following to the paper:

    A detailed derivation of the expression for $\mathbb{V}\,\left[Q_\alpha^{\sigma^2}(\Theta)\right]$ can be found in (Dixon et al., 2023).

38. *"In L298, to be valid, this equation requires some normalizing terms (1/N0,1/N0,1/N1)."*

    We corrected these equations.

39. *"In the first part of Eq. 16, one term should be $\mathbb{C}ov\left[\Delta_\alpha^{\sigma^2}, \Delta_\beta^\mu\right]$."*

    Fixed.

40. *"In Eq. 11, both $Q_\alpha$ and $Q$ are referred to as MC estimators (in Eq. (10) and on L271, respectively)."*

    We now state the following before equation 11.

    Consequently, any MC estimator $Q_\alpha(\Theta)$ of an exact statistic $Q$, such as $Q_\alpha^\mu(\Theta)$ and $Q_\alpha^{\sigma^2}(\Theta)$, is a random variable and the mean-squared error (MSE)

41. *"It would be nice define the MC estimator precisely, as well as the quantity that it is estimating."* Eq. (16) and Eq. (17) define the MC estimators of the mean and variance of mass change, respectively. We now reference these equations throughout the paper to make clear which quantities we are talking about as suggested comment 44.

42. *"Please be consistent in the notation. For example, QACV is bold in Eq. 15, but not in Eq. 20."*

    Fixed.

43. *"Please use equation numbers for all equations."*

    We would to only number equations if they are referenced in the text. I cannot find guidance online about the journals requirement to number all equations. We will do so if the editor requests we do so. However, we have increased the number of numbered equations to allow the reader to easily refer back to relevant equations, when discussing them in the text to address comment 44.

44. *"Throughout the text, try as much as possible to refer to the relevant equations and/or mathematical variables. That would be incredibly helpful for the reader to understand the methods more easily. For example, refer to Eq. 15 every time the "ACV estimator covariance" is mentioned, refer to L197 when mentioning sampling from the prior, refer to L227 when mentioning sampling from the posterior. And there are many more instances, which I will not enumerate here. But I encourage the authors to look for every instance where the reader would benefit from knowing clearly which quantity or equation a certain statement relates to."*

    We have added references throughout the revised paper.

**3.1.2 Adding some intuitive explanations**

*"Here and there, it would help to add a simple sentence to give a better intuition about some concepts. I provide a few examples here below. Again, this is not an exhaustive list. So, I encourage the authors to actively look for similar statements, equations, or paragraphs that could benefit from some intuitive explanations."*

We added intuitive explanations whenever suggested by the reviewer. We also took the opportunity to provide intuitive explanations at other places in the text. Any such changes are marked in red in the revised document.

45. *"Towards the beginning of the manuscript, please provide one short paragraph to explain what the statistics of interest are, and why they are uncertain. I believe that all readers might not intuitively understand the concept of variance of a variance."*

    We added the following statement to the introduction: *"However, the substantial computational cost of evaluating ice-sheet models limits the number of model simulations that can be run, and thus the accuracy of uncertainty estimates."* For example, when estimating the mean of a model with Monte Carlo, the mean squared error (MSE) in the estimated value only decreases linearly as the number of model simulations increases.

    We also added the following statement in Section 4.1:

    MC estimators converge to the true mean and variance of $f_\alpha$ as the number of samples tends to infinity, but using a finite number of samples $N$ introduces an error into the MC estimator that depends on the sample realizations used to compute the estimators. That is, two different realizations of $N$ parameter samples $\Theta$, and the associated QoI values, will produce two different mean and variance estimates (see Figure 4). Consequently, any MC estimator $Q_\alpha(\Theta)$ of an exact statistic $Q$, such as $Q_\alpha^\mu(\Theta)$ and $Q_\alpha^{\sigma^2}(\Theta)$, is a random variable.

46. *"L228: Please add one or two sentences to explain that $g(\theta)$ can be computed without time stepping model solves, and why this is the case."*

We added the following text. After line 228. We were able to calibrate the model using only a steady model without time-stepping because we assumed that the velocity data were collected when the ice sheet which was in equilibrium.

47. *"L247: Please explain that $\Sigma_{post}$ characterizes the balance between the prior uncertainty in the friction field estimate, and the model-observation mismatch weighted by the observational noise."*

We added the following statement at the end of section 3:

The posterior characterizes the balance between the prior uncertainty in the friction field and the model-observation mismatch, weighted by the observational noise. In the limit of infinite observational data, the posterior distribution will collapse to a single value. However, in practice when using a finite amount of data, the posterior will only change substantially from the prior in directions of the parameter space informed by the available data, which were captured by our low-rank approximation.

48. *"L298: Why is this valid regardless of how truthfully $f_1$ approximates $f_0$?"*

Please see response 15 above.

49. *"L322: What do the control variates represent?"*

We now state:

The estimator of the low-fidelity mean $Q_1^\mu(\Theta_0)$ is referred to as a control variate because it is a random variable, which is correlated with the random estimator $Q_0^\mu(\Theta_0)$, and can be used to control the variance of that high-fidelity estimator. The term $Q_1^\mu(\Theta_1) \approx Q_1^\mu$ is an approximation of the true low-fidelity statistic $Q_1$ that is used to ensure that the ACV estimator is unbiased, i.e. $\mathbb{E}_\Theta \left[ Q_{\text{ACV}}^\mu(\Theta_0, \Theta_1) \right] = \mathbb{E}_\Theta \left[ Q_0^\mu(\Theta) \right] + \eta \left( \mathbb{E}_\Theta \left[ Q_1^\mu(\Theta_0) \right] - \mathbb{E}_\Theta \left[ Q_1^\mu(\Theta_1) \right] \right) = Q_0^\mu + \eta \left( Q_1^\mu - Q_1^\mu \right) = Q_0^\mu$.

50. *"Figure 4: How do results between these different sample allocation strategies differ? For example, does one approach prioritize minimizing the diagonal entries of the ACV covariance, versus another better constraining the correlation between different models?"*

See our response to comment 16.

**3.2 Line-by-line comments**

51. *"General (1): The text would benefit from the use of many more commas. I encourage the authors to, at least, double the number of commas in the manuscript. The general writing level is good, so I have no doubt that the authors can find sentences that need (or would benefit) from commas."*

While revising the document, we broke compound sentences into multiple sentences where appropriate and used additional commas for complex sentences where needed.

52. *"General (2): The quality of the figures is low. Color scales should be more informative, units should be provided, span of y axes should be appropriate for the range of values shown, a scale in km should be added when showing Humboldt, labels should be added to colorbars, etc."*

We regenerated all figures in the document to improve readability.

53. *"L4 Here and elsewhere, the term "accuracy" is used very loosely, and encompasses a wide range of concepts. When used to describe a model degree of fidelity, please always use "fidelity" since this is the technical term used for the name of the method (MFUQ). When describing the amount of variance, please rather use precision, which is mathematically the inverse of the variance."*

   We now use precision when referring to the accuracy of a statistic. Instances of its use are highlighted in red in the revised document. We also use fidelity when approriate.

54. *"L4 Replace ice sheet by glacier."*

   Fixed.

55. *"L5 The problem size is not "representative" of continental scale studies. Please use more careful wording."*

   We now state:

   The problem size and complexity were chosen to reflect the challenges posed by future continental scale studies while still facilitating a computationally feasible investigation of MSE methods.

56. *"L11 prediction should be plural."*

   Fixed.

57. *"L15 Add report after IPCC."*

   Fixed.

58. *"L15 Ice sheets are all land-based."*

   Thanks. We removed "land-based".

59. *"L26 Throughout the manuscript, affect should be used as a verb instead of effect."*

   Fixed.

60. *"L28 Replace inadequacy by uncertainty."*

   We now state

   In addition, while the comparison of model outputs has been used to estimate uncertainty arising from model inadequacy

61. *"Throughout the manuscript, there is confusion in the wording of "parameters" and "inputs". For example, both terms are used interchangeably to characterize the basal friction field. Please (i) always use the same term for a same meaning, and (ii) clearly define the difference between parameters and inputs in the Introduction."*

   We now always use the term parameters.

62. *"L39 There are also methods that have been developed to reduce the problem dimensionality. Please cite Brinkerhoff (2022)."*

   We added a reference to Brinkerhoff (2022) to Section 3, which discusses Bayesian calibration. The introduction now focuses on quantifying uncertainty in predictions, whereas the focus of the cited paper is on Bayesian inference.

63. *"L48 When using the notion of MSE, it is important to clearly define with respect to which quantity the error is considered. In this study, I believe that the error is considered with respect to the expec- tation of the mass change from the high-fidelity model with respect to the posterior distribution of the basal friction field. I realize that this is not straightforward to include. But I recommend adding a couple of sentences to give the definition, and possibly explain its meaning."*

We added the following to Section 4.1:

The MSE of an MC estimator, Eq. (18), consists of two terms referred to as the estimator variance (I) and the estimator bias (II). The bias term of the MSE is caused by using a numerical model, with inadequacy and discretization errors, to compute the mass change. More specifically, letting $Q_\infty$ denote the exact value of the statistic of a numerical model with zero discretization error but non-zero model inadequacy error, and $Q_0$ denote the highest-fidelity computationally tractable model approximation of $Q_\infty$, then the bias can be decomposed into three terms

$$(\mathbb{E}[Q_\alpha(\Theta)] - Q) = (\mathbb{E}_\Theta[Q_\alpha(\Theta)] - Q_\infty + Q_\infty - Q_0 + Q_0 - Q) \tag{3}$$
$$= (\mathbb{E}_\Theta[Q_\alpha(\Theta)] - Q_0) + (Q_0 - Q_\infty) + (Q_\infty - Q) \tag{4}$$

The first term is caused by using a model $f_\alpha$ with numerical discretization that is inferior to that employed by the highest fidelity model $f_0$. The second term represents the error in the statistic introduced by the numerical discretization of the highest-fidelity model. The third term quantifies the model inadequacy error caused by the numerical model being an approximation of reality.

Later in that section we now also state:

Constructing a SFMC estimator of statistics, such as the mean Eq. (16) or variance Eq. (17), with a small MSE ensures that the value of the estimator will be likely close to the true value, for any set of model parameters samples. However, when using numerical models approximating a physical system, constructing an unbiased estimator of $Q$ is not possible. All models are approximations of reality and thus the model inadequacy contribution $Q_\infty - Q$ to the bias decomposition in Eq. (3) can never be driven to zero. Additionally, it is impractical to quantify the discretization error $Q_\infty - Q_0$. Consequently, SFUQ methods focus on producing unbiased estimators of $Q_0$, such that $\mathbb{E}_\Theta[Q_\alpha(\Theta)] = Q_0$.

64. *"L50 I think that the authors might not be aware of the study of Bulthuis et al. (2020). Please consider referring to it."*

The study cited uses a surrogate based multi-fidelity method which has no direct relationship to the statistical estimation methods discussed in this paper. Moreover, the paper's reliance on a surrogate means that it can not be applied to ice-sheet models with large numbers of parameters as is the focus of this paper. Consequently, we did not cite this paper in the revised manuscript.

However, we did find the a paper using MFSE on an ice-sheet model Gruber et al. (2022). This paper, uses one type of ACVMF estimator, i.e. MFMC, to estimate uncertainty in a steady-state Blatter-Pattyn model of an ice sheet defined on rectangular a rectangular parallelepiped domain. Specifically, MFMC was used to estimate uncertainty in the $L^2$ norm of the ice-velocity caused by uncertainty in two parameters, specifically a scalar representing basal friction and a variable parameterizing the simple bed topography.

We added a citation to this work in the revised document. Specifically, we now state: Note, Gruber et al. (2022) previously applied MFSE to a ice-sheet model; however, their study was highly simplified, as it only quantified uncertainty arising from two uncertain parameters of an ice-sheet model define on a simple geometric domain.

65. *"L61 I find the changes between past and present tense somewhat confusing. I recommend consistently using a single tense."*

    We tried to improve the consistency of tense wherever possible. In revising our paper we used the following guidelines.

66. *"L85 Model simulations do not only capture the "melting", since they represent the dynamic response of the glacier as well. This should induce changes in the amount of ice flowing out of the simulated domain."*

    We revised the sentence.

67. *"L102-104 In this sentence, the summary of the Stokes and MOLHO models sound identical to me."*

    We now state: In summary, the simpler 2D SSA model is formulated to simulate grounded ice with significant sliding at the bed or ice shelves, while the 3D MOLHO model is designed to capture the evolution of ice sheets over frozen and thawed beds, as well as ice shelves.

68. *"L113 Replace exorbitant by impractical."*

    Fixed.

69. *"L121 Provide a reference for q = 1/3."*

    Done.

70. *"L125. Define the ∥ notation here."*

    Done.

71. *"L139 The $\psi$ term is already multiplied by $n$ above, so this multiplication should not be included in the definition of $\psi$. Also, why is there an extra term $\rho g(s - z)$ in the boundary condition on $\Gamma_m$ here compared to the Stokes model?"*

    Fixed.

72. *"L145 $\partial \Sigma$ is not defined."*

    Fixed.

73. *"Figure 2 Show the meshes side-by-side (+ all comments from General (2))."*

    The paper now includes plots of all four meshes side-by-side.

74. *"L172 The statement "one of the largest sources of prediction uncertainty" should be quantified and referenced with a citation."*

    Please see response 7.

75. *"L180 "we set $\mu = 0$". I believe that this is only for the prior. It seems strange to me that the posterior is forced to have zero mean. Please specify."*

    We now state: In this study, we adopted a fully distributed approach that treated the friction as a log-Gaussian random field that is $\theta = \log(\beta)$ with a Gaussian prior distribution $p(\theta) \sim \mathcal{N}(\mu, \mathcal{C})$ with $\mu = 0$.

76. *"L183 There is a switch from C to $\Sigma$ without mentioning it. Specify that $\Sigma_{prior}$ is a covariance."*

    We now state:

    a finite-dimensional discretization of the operator $\Sigma_{\text{prior}} \approx \mathcal{C}$ with

77. *"L185 Why is the source term only integrated over $\Gamma_g$ and not over $\Gamma_f$ ? I would assume that snow accu- mulation and surface melting should also be computed over the floating parts of the domain."*

    That was a typo, the integrals are over the lower surface of the ice sheet $\Gamma_l = \Gamma_g \cup \Gamma_f$. Fixed.

78. *"L199 Please specify "this Gaussian prior"."*

    Fixed.

79. *"L199 Replace "on" by "constrained with"."*

    Fixed.

80. *"L207 The authors sort of sweep under the rug the possible influence of ocean melt on their methods. Melt at the boundary can induce strong dynamical responses by a marine-terminating glacier. It can be expected that differences between models of different levels of fidelity would be exacerbated, potentially diminishing the advantages of the MFUQ. Please discuss this more thoroughly in the Discussion."*

    We now state in the discussion: Moreover, introducing more complicated physics in the highest-fidelity model, such as calving, could degrade the performance of MFSE. For example, ice melt at the boundary can induce strong dynamical responses in a marine-terminating glacier, which could potentially reduce the correlation between models that do not capture this phenomenon.

81. *"L213 Please clarify why this assumption is required in the procedure. I believe that it is needed to compute the $g(\theta)$ function represented by the Blatter-Pattyn flow model. And that without this assumption, the PDE-constrained optimization cannot be solved."*

    We now state: Before continuing, we wish to emphasize two important aspects of the calibration used in this study that mean our results must be viewed with some caution. First, we assumed the observational data to be uncorrelated, as assumed in most ice-sheet inference studies, including (Recinos et al., 2023; Isaac et al., 2015). Moreover, we also assumed our Gaussian error model to be exact. However, neither of these assumptions are likely to be perfect in reality. For example, Koziol et al. (2021) showed that, for an idealized problem, ignoring spatial correlation in the observational noise can lead to uncertainty being underestimated. Second, our optimization of the MAP point was constrained by the coupled velocity flow equations and steady-state enthalpy equation, which is equivalent to implicitly assume that the ice is at thermal equilibrium. Theoretically, this assumption could be avoided if the temperature tendencies were known, but they are not. Alternatively, transient optimization over long time periods, comparable to the temperature time scales, could be used. However, this approach would be computationally expensive and would require including time-varying temperature data (e.g., inferred by ice cores) which are very sparse.

82. *"L217 "However, such approaches ignore the uncertainty in the model parameters due to using a finite amount of noisy observational data". This statement is incorrect. Observational*

*uncertainty can be incorporated in cost functions. See for example Eq. (1) from Goldberg (2015).*"

That's correct. We also account for the uncertainty in our deterministic inversion to compute the MAP point. We have rephrased our statement: However, such approaches only use a single optimized parameter value to represent the uncertainty in the model parameters that arises from using a finite amount of noisy observational data.

83. "*L222 "the likelihood distribution": the likelihood is a function, not a distribution.*"

   We disagree. The likelihood is the probability of the data given the parameters $\theta$ $p(y|\theta)$. However, we have dropped the word distribution from the sentence.

84. "*L232 Please add a justification for this choice of $\alpha$.*"

   We expanded Remark 5.1 to discuss how we heuristically chose parameter $\alpha$.

85. "*L233 Please specify "samples from the posterior of $\log(\beta)$".*"

   Fixed.

86. "*L249 Again, this statement is likely not obvious to most readers. At first sight, the computation that is referred to here is a simple addition of two matrices $(H_{MAP} + \Sigma^{-1})$ prior explain why this is intractable would be beneficial.*"

   We rephrased the sentence and explained why it is intractable to compute $H_{\mathrm{MAP}}$ and invert high-dimensional dense matrices.

87. "*L254 and 255 Replace ice sheet by glacier.*"

   Fixed.

88. "*L263 What do the authors mean by "robust"?*"

   We now state:

   SFMC quadrature is a highly versatile procedure that can be used to estimate a wide range of statistics for nearly any function regardless of the number of parameters involved.

89. "*L264 "three-step""*"

   Fixed.

90. "*L265 The m superscript should be n (which would preferably be another letter than n, see Major comment).*"

   We changed $m$ to $n$. However, we like to use capital $N$ to denote the number of samples and lower case n to denote the index $n = 1, \ldots, N$.

91. "*L266 Specify "basal friction field".*"

   Fixed.

92. "*L278 "The bias term of the MSE (11) is caused by using a numerical model, with inadequacy and discretization errors, to compute the mass change." Here also, I ask for clarification: bias with respect to what? If it is with respect to observations, then observational uncertainty should also be discussed. If it is with respect to the highest-fidelity model, then the latter is also a "numerical model", and the sentence is inappropriate. If it is with respect to the unknown*

*true dynamical behavior of Humboldt glacier, then there is a philosophical question of how to compute a mean squared error with respect to a quantity that cannot be known."*

See our response to comment 63.

93. *"L287 Typo estimated."*

    Fixed.

94. *"L296 Two-model"*

    Fixed.

95. *"L316 QoI is not defined."*

    Fixed.

96. *"L324 Concerning $\Theta^*_\alpha \cup \Theta_\beta = \emptyset$. (i) I believe that $\cup$ should be $\cap$, (ii) I believe that "for $\alpha \neq \beta$" should be specified."*

    We have added additional clarifying text.

97. *"Eq. (16) Is $\mathbb{C}ov[Q_0, \Delta_0](\Theta_{ACV})$ a covariance matrix? If so, it should be symmetric. However, the $(0,1)$ and $(1,0)$ entries of the right-hand-side seem different to me. Please explain."*

    The quantity is a covariance matrix. It is symmetric in the arguments $\mu$ and $\sigma^2$.

98. *"L351 "following standard practice": provide citation."*

    We added a citation

99. *"L352 Please add an additional explanatory statement, for example: This involves computing the high- fidelity and all the low-fidelity models for the same set of samples $\Theta$ pilot."*

    We added the following statement This involves computing the high-fidelity and all the low-fidelity models at the same set of samples $\theta_{\text{pilot}}$.

100. *"L360 Specify "introduce sampling errors"."*

    Fixed.

101. *"L367 I believe that the same should be specified for $\alpha \cup \beta^*$ and $\alpha \cap \beta^*$"*

    Fixed.

102. *"L382 There is no verb in this sentence."*

    Fixed.

103. *"L391 Please add an additional explanatory statement, for example: This can happen if a subset of the low-fidelity models correlate much better with the high-fidelity model than the rest of the low-fidelity models, for example."*

    We added the statement:

    This occurs when a subset of the low-fidelity models correlate much better with the high-fidelity model than the rest of the low-fidelity models. For instance, some low-fidelity models may fail to capture physical behaviours that are important to estimating the QoI.

104. *"L394 I think this should be estimator types."*

    Fixed.

105. *"L394 "model models subsets" is either a typo, or very confusing language."*

We fixed the typo.

106. *"L402 was should be were."*

Fixed.

107. *"L413 ice-sheet should be glacier."*

Fixed.

108. *"L415 Typo an an."*

Fixed.

109. *"L424 Specify MALI ice-sheet code with the Blatter-Pattyn flow model."*

See response 182.

110. *"Figure 9 I provide here a concrete example of how to help the reader navigate through the technical details of the study. The caption should specify: "... MAP point ($\theta_{MAP}$ in Eq. (9)) ... prior variance ($\Sigma_{prior}$ in Eq. (xxx)) ... posterior variance ($\Sigma_{post}$ in Eq. (xxx))". Using more such links between text and mathematics would really help reading the study."*

We have added symbols and equation references throughout the paper.

111. *"L437 "speeds up as it thins": I think that this statement is incorrect, although I see what the authors mean. A glacier does not speed up because of thinning. It speeds up because of increasing surface slope, caused by enhanced thinning at the front. Also, the inverted relation holds: as a glacier speeds up, it discharges more ice into the ocean, leading to thinning."*

We now state In general, the glacier speeds up as negative surface mass balance causes the surface to steepen near the terminus. The largest speedup occurs in the region of fast flow in the north where basal friction is small.

112. *"Remark 5.2 I believe that this is an important scientific aspect, which is also somewhat swept under the rug. In their results, the authors demonstrate that the simulated mass change is sensitive to high-frequency variability in the basal friction field. As such, the interpolation method from fine to coarse meshes is potentially very influential. Which interpolation method has been used here? If it is simple linear interpolation, then all the high-frequency variability will be smoothed out. This would affect the behavior of low-fidelity models with coarser meshes. I recommend that the authors try interpolation methods that better preserve high-frequency variability (nearest neighbor, or maybe polynomial interpolation) and evaluate the impacts on their results."*

We used the finite element mesh of the high-fidelity model to interpolate the basal friction from that mesh onto the coarser meshes. We added a note to this effect to the text.

However, we do not believe that a different interpolation strategy is needed. Our results show that multi-fidelity models are able to use coarser meshes despite those meshes not being able to accurately representing the basal friction. See Figure 13 in the original submission. We make this point on paragraph starting on line 551 in the original submission.

113. *"L458 "significant differences": the word significant is misused here, because no statistical test has been performed. If a statistical test has been performed, please specify which one, and provide p-values. Furthermore, by eye, the differences do not seem very large in Figure*

*8 compared to the standard deviations. However, this is difficult to say because of the terrible choice of y-axes span in Figure 8, which should be changed."*

See response 27.

114. *"L460 The meaning of accuracy is not clear here (see comment on L4)."*

See response 53.

115. *"L476 Provide a citation to support this statement."*

See response 29.

116. *"477. Please quantify "the error introduced". "not insignificant": this wording is misused here, because no statistical test has been performed. If a statistical test has been performed, please specify which one, and provide p-values."*

We now state:

Moreover, we found that the error introduced by using a small number of pilot samples can be substantial, yet it is typically ignored in existing literature.

117. *"L479 Please specify the number of realizations per bootstrap. From the rest of the text, I believe that it is 20 realizations per bootstrap samples, but this should be clarified explicitly."*

Fixed.

118. *"Figure 10 (1) I am puzzled by the very high upper bound on the variance reduction of the variance. In the ratio, the SFMC variance estimator is the denominator, which should therefore be the same for all the bootstrap samples. As such, the very high upper bound is caused by an unrealistically low estimated ACV variance via Eq. (15). This leads me to the question: is the approximation on pilot samples via Eqs. (15,20) unstable when using bootstrap with replacement? In any case, please provide an explanation about the very high value of the 95% quantile."*

We now report the 10% and 90% quantiles to more clearly show how variance reduction distributes. We also refer the reviewer to the statement on page 26 *"However, the box plots in Figure 10a highlight that using only 20 samples introduces a large degree of uncertainty into the estimated variance reduction."* We also refer the reviewer to the statement on page 28 *"While, increasing the number of pilot samples decreased variability, we believed that the benefit of further increasing the number of pilot samples would be outweighed by the resulting drop in the variance reduction."*

119. *"Figure 10 (2) It is not immediately clear why a same model combination would give different estimates of the variance reduction, since the ice sheet models are deterministic. If I understand correctly, some of this variability comes from the random bootstrapping within the pilot samples, and some of the variability comes from the ACV estimator selected (MLMC, MFMC, ACVMF). Is it possible to quantify how these two sources of variability compare? And in turn, is it possible to quantify how much of the boxplot spread in Fig. 10a is due to these two factors versus the fact that different subsets of low-fidelity models have been selected?"*

The variability in the plots is induced entirely by the bootstrapping. This plot was included to demonstrate that using a small number of pilot samples introduces a non-trivial error. Each estimator does have a different estimator variance. However, the box plots just report the smallest estimator variance, across all estimators, for each bootstrapped sample.

We now state in the paper: Please note that, while we enumerate over numerous estimators, each with a different variance reduction, the variability in the plots is induced entirely by the bootstrapping procedure we employed. The box plots report the largest variance reduction, across all estimators, for each bootstrapped sample.

120. *"L489 Specify subsets of model combinations."*

Fixed.

121. *"L491 the original 20 pilot samples are used."*

Fixed.

122. *"L491 Specify were determined useful to include for reducing. (Probably that individually, all the models would be useful. But they are not relative to including other better-correlated or computationally- cheaper models.)"*

We now state:

Moreover, bootstrapping the estimators also revealed that using all models simultaneously to reducing the variance of the ACV estimator was not as effective as using a smaller subset of models.

123. *"L496-499 I could not understand the end of this paragraph. It would be helpful if the authors defined the notion of hierarchical relationship."*

See response 30.

124. *"Figure 11. Please specify the number of samples for each case (2, 3, and 4 models)."*

The number of samples allocated to each model depends on the bootstrapped realization of the pilot data to compute the pilot statistics. Consequently, there is no one number that we can provide.

125. *"L520 Again, the meaning of "accurate" is not well-defined."*

We now state: MSE of the final ACV estimator we would construct would be much smaller than the MSE of a SFMC estimator of the same cost because even the smallest variance reduction was greater than 14.

126. *"L520 "even the smallest variance reduction was greater than 20". This is not what is shown in Fig. 11. Certainly not for the cases of 2 and 3 models. And for the case of 4 models, it seems to me that even the 5% quantile is below 20, suggesting that the smallest value is definitely smaller than 20."*

See our response to comment 125.

127. *"L522 Replace that by which (with a comma, see General comment (1))."*

Fixed.

128. *"L523 The three models listed do not include $MOLHO^*_{1km,9days}$. As such, I believe that it corresponds to the case "4 models" in Fig. 11. I find the discrepancy between the number of low-fidelity versus the total number of models confusing. Please use a consistent manner to quantify the number of models used."*

Throughout the paper, We now always include the highest-fidelity model when counting the number of models used by an estimator. The particular statement highlighted by the reviewer

had a typo, which we corrected. Three models were chosen despite the estimator being allowed to use four.

We now state in the paper: Note, an estimator allowed to choose four models may still choose less than four models, which will happen when some of those models are not highly-informative.

129. *"L525 Please remind the readers where these numbers come from."*

See response 32.

130. *"L527 I do not see any right or left panel."*

Fixed.

131. *"535 Please specify another estimator (i.e., MLMC or ACVMF)."*

Fixed.

132. *"In Discussion. This question relates to my curiosity concerning the complementarity between this method and stochastic ice sheet modeling. Here, the MFUQ samples uncertainty from a single time-constant uncertain input. In contrast, stochastic modeling (e.g., Verjans et al., 2022) samples uncertainties between multiple correlated uncertain inputs, and at different time steps (for example SMB variability in time is prescribed as stochastic). However, since the statistical properties of the time-varying stochastic inputs (i.e., the auto-correlation, the covariance structure and the mean of each stochastic input) can be specified a priori, I suppose that, in theory, the MFUQ method could be applied. But I wonder if this is practically feasible. I think that the Discussion would benefit from a short paragraph about this point."*

We added the follwing to the discussion:

This study focused on investigating the efficacy of using MFSE to accelerate the quantification of parametric uncertainty using deterministic ice-sheet models. We did not quantify the uncertainty arising from model inadequacy. Recently Verjans et al. (2022), attempted to quantify model uncertainty by developing stochastic ice-sheet models designed to simulate the impact of glaciological processes that exhibit variability that cannot be captured by the spatiotemporal resolution typically employed by ice-sheet models, such as calving and subglacial hydrology. The MFSE algorithms presented in this paper can be applied to such stochastic models, by sampling the model parameters and treating the stochasticity of model as noise. However, the noise typically reduces the correlation between models and thus the efficiency of MFSE (Reuter et al., 2024). Moreover, this study only focused on estimating the mean and variance of mass change. Consequently, the efficacy of MFSE may change when estimating statistics – such as probability of failure, entropic risk, and average value at risk (Rockafellar and Uryasev, 2013; Jakeman et al., 2022) – to quantify the impact of rare instabilities and feedback mechanisms in the system. We anticipate that larger number of pilot samples than the amount used in this study will be needed to estimate such tail statistics, potentially reducing the efficiency of MFSE.

133. *"L567 Appendix B."*

Fixed.

134. *"Figure 13 (1) Changing the color scale here is absolutely necessary."*

It was difficult to find a color scheme that more clearly highlighted the difference. However, we believe the other two plots in Figure 13 supported our argument that the models did not produce exactly the same predictions. Consequently, we removed the left panel of Figure 13.

135. *"Figure 13 (2) If I understand correctly, the basal friction field should be model-independent. The differences only stem from the interpolation method. This should be specified in the caption. Furthermore, this Figure seems to confirm my comment about Remark 5.2."*

   See response 112

136. *"L589 "variance" should be standard deviation here, since Gigaton units are specified."*

   Fixed.

137. *"L590 "significance": the word significant is misused here, because no statistical test has been performed. If a statistical test has been performed, please specify which one, and provide p-values. Furthermore, even the meaning of "the significance of these numbers" is not clear to me."*

   We now state:

   However, the exact values of these statistics were impacted by our modeling choices.

138. *"L593 In this study, the basal friction field distribution was derived assuming that all other variables were perfectly known. In reality, different sources of uncertainty can mix. Please cite Gudmundsson and Raymond (2008) and add one or two sentences about this to the Discussion."*

   We added further discussion of the limitations of our approach. For more details, see our response to comment 80.

139. *"L614 Please mention here that this study explores the use of MFUQ for low-order moments only. One can wonder if this method can be used for statistics such as skewness or quantiles in the tails of the distribution. This can be particularly important for evaluating the response to an input that could introduce instabilities and feedback mechanisms in the system, such as ocean or SMB forcing."*

   We have added the following to the discussion:

   Moreover, this study only focused on estimating the mean and variance of mass change. Consequently, the efficacy of MFSE may change when estimating statistics such as probability of failure, entropic risk, average value at risk, etc. to quantify the impact of rare instabilities and feedback mechanisms in the system. We anticipate that larger number of pilot samples, than used here, will be needed to estimate such tail statistics, thus potentially reducing the efficiency of MFSE.

140. *"L616 Here, and in many other instances, the authors insist about the fact that MFUQ can be used at continental scale to estimate uncertainty on ice-sheet mass change statistics. However, such a statement is not well-supported by their results. Just looking at the results, one can argue that the MFUQ framework presented here requires 36 CPU days for a single glacier. Scaling this linearly to the Greenland ice sheet results in $O(1-10)$ years of computation. Thus, there should be a slightly more in-depth explanation of why MFUQ is applicable for studies at the ice-sheet-scale."*

   The following statement from the original paper, is one example that raised the reviewers concerns:

*"Thus, MFSE reduced the cost of estimating uncertainty from over two and a half years of CPU time to just over a month, assuming the models are evaluated in serial."*

To address the reviewers concern, we had added the following statement to the discussion.

Note that while applying MFSE to the Humboldt Glacier took over a month of serial computations, the clock time needed for MFSE can be substantially reduced because MFSE is embarrassingly parallel. Each simulation run in the pilot stage can be run in parallel without communication between. Similarly, for the exploitation phase. Moreover, each simulation can be computed in parallel. Consequently, while using MFSE for continental scale UQ studies may require years of serial CPU time, distributed computing could substantially reduce this cost, potentially one to two orders of magnitude. The exact reduction would depend on the number of CPUs used.

141. *"L618 "substantially": please quantify and provide a citation."*

We now state: Mass loss from ice sheets is anticipated to contribute 10s of cm to sea-level rise in the next century under all but the lowest emission scenarios (Edwards et al., 2021).

142. *"L638 I do not understand the underlying meaning of this sentence. Please expand or remove it."*

We now state:

Moreover, while the utility of the lower-fidelity models ultimately chosen for MFSE were not clear at the onset of the study, we were still able to estimate uncertainty at a fraction of the cost of single fidelity MC. This was achieved despite the need to conduct a pilot study that evaluated all models a small number of times.

143. *"L641 Antarctica and Greenland."*

Fixed.

144. *"L661 Again, the meaning of accurate is unclear here. It would be more correct to explain that the approximation level depends on the variance retained in the truncation."*

We now state: In order to compute a low-rank approximation of the matrix $T$ we truncated the spectral decomposition of $\mathbf{W} = \mathbf{U} \left( (\mathbf{\Lambda} + \mathbf{I})^{-\frac{1}{2}} - \mathbf{I} \right) \mathbf{U}^\top$ by discarding the eigenvalues $\lambda_i$ such that $\left| 1 - \frac{1}{\sqrt{\lambda_i + 1}} \right| \ll 1$. This ensured that the low-rank approximation of $T$ well approximated $T$ in the spectral norm sense.

145. *"L668 Please define K here as well. Otherwise, the reader needs to go back to the main text."*

Fixed.

146. *"L674 I do not see why the representation is "bi-Laplacian". I wonder if this term is not inadvertently misused here. Could this please be clarified? I believe that applying the Laplace approximation has no link with the bi-Laplacian operator, but sorry if I am misunderstanding here."*

The bi-laplacian is used to define the prior-distribution of the log normal basal friction field. We dropped the mention of bi-laplcian to avoid confusion.

147. *"L685 Typo: in this study"*

Fixed.

148. *"L686 Typo: modes"*

Fixed.

149. *"L700 I believe that MF estimator should be ACV estimator"*

Fixed.

150. *"L701 I believe that MFUQ estimator should be ACV estimator"*

Fixed.

151. *"L702 This should be: The mean and variance bootstrapped (...)."*

Fixed.

152. *"L706 This should be: the uncertainty in the mean mass change (...)"*

Fixed.

153. *"L706-708 Please refer to Figure B5."*

Fixed.

**4 Reviewer 3 (Dan Goldberg)**

**4.1 General comments**

*"The manuscript, An evaluation of multi-fidelity methods for quantifying uncertainty in projections of ice-sheet mass change by Jakeman et al, uses a new computational approach to determining the posterior uncertainty of ice mass change in a glacier forecast conditioned on observational data and uncertainty. The main contribution is a Multi Fidelity Uncertainty Quantification (MFUQ) scheme which samples from a probability distribution (see below) and provides an inexpensive means of Monte Carlo variance reduction in the calculated statistics that requires far less simulation time. This is achieved through generating ensembles from models that are of lower fidelity (coarser resolution / longer time steps) whose dependencies on the input parameters are similar. The probability distribution which is sampled – that of the sliding parameter conditioned on observations and model physics – would be too expensive to find via Monte Carlo methods. Rather, a method introduced by others in the literature – which approximates this posterior as Gaussian and finds a low rank approximation to the inverse covariance matrix to make the problem tractable – is used.*

*The methodology introduced in the paper – the MFUQ scheme – is fairly well described and seems quite useful, and its results deserve to be shown.*

*However, there are a number of major issues I have with the manuscript. Aside from a number of writing issues, such as inconsistent statements and introducing of terms and symbols without definition or explanation (see specific comments), I feel that the messaging of the paper in the introduction is not in line with what the authors have actually done. Furthermore they have downplayed or overlooked recent works in the literature – works which, in some cases, bring the methodology of this study into question. I will highlight these in general comments below.*

*Finally I should point out first though that it monte carlo methods are not my area of expertise. I have some specific comments about certain things that looked as thought they might be typos or need more explanation. Largely however I do not have much to say about the actual MFUQ methodology and its presentation, and I hope that other referees can assess it better. "*

**4.2 High-level comments**

154. *"The paper sets out to deal with parametric uncertainty, which is the case. But the introduction is written in a way that makes it seem that MFUQ is used to solve the "full" problem – that is, quantifying the probability density of mass change conditioned on the model and observations, which can be termed $p(Q|m, U)$ where $Q$ is the mass change, $m$ is the model and $U$ is the observations. But in truth a different method (Hessian-based) was used to find the posterior density of the frictional field $q$, and then this was sampled from to find the posterior of $Q$ i.e. $p(Q|m, q)p(q|m, U)$ – and $p(Q|q)$ is the only component being determined by MFSE. I think this could be potentially very misleading and give the impression that MFSE is capable of the "full" problem when from the results of the paper it definitely is not. This is very important: given the newness of the fields of ice-sheet modelling and ice-sheet uncertainty quantification there is extensive misunderstanding about which problems can be tackled by sampling methods and which require alternative methods. Although this is somewhat covered in lines 61-71 of the manuscript, the passage requires familiarity with the field and with both MC and Hessian-based UQ. It needs to be much more clear – with mathematical formality – which distribution is being quantified using MFSE."*

We now state the following in the introduction:

*"This study investigated the efficacy of using MFUQ methods to reduce the computational cost needed to accurately estimate statistics summarizing the uncertainty in predictions of sea-level rise obtained using ice-sheet models parameterized by large numbers of inputs."* To facilitate a computationally feasible investigation, we focused on reducing the computational cost of estimating the mean and variance of mass change in the Humboldt Glacier in northern Greenland. This mass change was driven by uncertainty in the spatially varying basal friction between the ice sheet and land mass, under a single climate change scenario between 2007 and 2100. Specifically, letting $f$ denote the mass change at 2100 computed by a mono-layer higher-order (MOLHO) (Dias dos Santos et al., 2022) model $\mathcal{M}$, $\boldsymbol{\theta}$ denote the parameters of the model characterizing the Basal friction field, and $\boldsymbol{y}$ denote the observational data, we estimated the mean and variance of the distribution $p(f \mid \mathcal{M}, \boldsymbol{y}) = p(f \mid \boldsymbol{\theta})p(\boldsymbol{\theta} \mid \mathcal{M}, \boldsymbol{y})$ in two steps. First, using a piecewise linear discretization of a log-normal basal friction field, we used Bayesian inference to calibrate the resulting 11,536 dimensional uncertain variable to match available observations of glacier surface velocity. Specifically, we constructed a low-rank Gaussian approximation (Isaac et al., 2015; Recinos et al., 2023; Barnes et al., 2021; Johnson et al., 2023; Perego et al., 2014) of the Bayesian posterior distribution of the model parameters $p(\boldsymbol{\theta} \mid \mathcal{M}, \boldsymbol{y})$ using a Blatter-Pattyn model (Hoffman et al., 2018). Second, we estimated the mean and mass of glacier mass change using 13 different model fidelities (including the highest-fidelity model), based on different numerical discretizations of the MOLHO and shallow-shelf (SSA) physics approximations (Morland and Johnson, 1980; Weis et al., 1999).

155. *"The manuscript is also misleading about contributions in this paper versus in the literature. Specific examples are given below, but the manuscript does not acknowledge previous authors' attempts to quantify the uncertainty of high-dimensional parametric uncertainty. In particular, a recent paper in The Cryosphere (Recinos et al, 2023, hereby shortened as BR23) has been overlooked. The authors can certainly be forgiven for this of course as the paper came out only last year, but it is extremely relevant to many of the assumptions and calculations within the manuscript (and is mentioned extensively in the specific comments below). Additionally, based on this paper there are several assumptions and/or approximations that give me serious reservations about this paper's results – these are easily identifiable in the specific comments*

*where BR23 is mentioned."*

We apologize for missing the work in BR23. We agree that the paper is highly relevant to this manuscript. We have edited the paper to address the reviewers reservations about the results we present. First, in our response to comment 156 we show that despite linearizing when computing the posterior distribution of the parameters, the parameter-to-QoI map is nonlinear. Second, we acknowledge that BR23 presents a more rigorous approach to setting the hyper-parameters of the prior imposed on the Basal friction field which is then used during Bayesian inference and we now state the positive benefits of this approach in multiple places in our paper. However, aside from our approach for choosing the prior hyper-parameters, the method we used for Bayesian inference, which is also used in BR23, is state-of-the-art in the ice-sheet community. Moreover, the goal of our study was to demonstrate the efficacy of MFSE on a ice sheet problem that has challenges representative of papers used to predict ice-sheet evolution and not to produce scientifically meaningful values for sea-level rise. We detail the limitations of our study in the discussion. Finally, we do not believe our approach and BR23 are mutually exclusive. Indeed, we believe that when adjoints are available in a ice sheet code, then the linearization approach in BR23 could be used to provide a highly-computationally efficient and accurate low-fidelity model that could be used to further increase the accuracy and computational gains of MFSE reported in this manuscript.

We now state in the introduction:

Most recent studies have focused on estimating uncertainty in the predictions of ice-sheet model with small numbers of parameters, e.g. (Nias et al., 2023; Ritz et al., 2015; Schlegel et al., 2018; Jantre et al., 2024), despite large numbers of parameters being necessary to calibrate ice sheet model to observations (Barnes et al., 2021; Johnson et al., 2023; Perego et al., 2014). However, recently Recinos et al. (2023) used the adjoint sensitivity method to construct a linear approximation of the map from a high-dimensional parameterization of uncertain basal friction coefficient, and ice stiffness, to quantities of interest (QoI) – specifically the loss of ice volume above flotation predicted by a shallow-shelf approximation model at various future times. The linearized map and the Gaussian characterization of the distribution of the parameter uncertainty was then exploited to estimate statistics of the QoI. While this method is very computationally efficient, linearizing the parameter-to-QoI map will introduce errors (bias) into estimates of uncertainty, which will depend on how accurately the linearized parameter-to-QoI map approximates the true map (Koziol et al., 2021). Moreover, the approach requires using adjoints or automatic differentiation to estimate gradients, which many ice-sheet models do not support. Consequently, in this study we focused on Multi-fidelity statistical estimation (MFSE) methods that do not require gradients.

We removed our claim that we were the first to compute uncertainty in QoI when using Bayesian inference to calibrate a large number of model parameters. We now state in the introduction:

Our study makes two novel contributions to previously published glaciology literature. First, it represents the first application of MFSE methods to quantify the impact of high-dimensional parameter uncertainty on transient projections of ice-sheet models defined a realistic physical domains. Our results demonstrate that MFSE can reduce the serial computational time required for a precise UQ study of ice-sheet contribution to sea-level rise from years to a month. Note, Gruber et al. (2022) previously applied MFSE to a ice-sheet model; however, their study was highly simplified, as it only quantified uncertainty arising from two uncertain parameters of an ice-sheet model define on a simple geometric domain. Second, our paper

provides a comprehensive discussion of the practical issues that arise when using MFSE, which are often overlooked in all MFSE literature.

This statement also includes reference to an earlier attempt at using MFSE with ice-sheet models. Please refer to our response to comment 64 for more details on the limitations of that study.

156. *"The underlying premise of the paper is that, given a Hessian-based approximation of the posterior parameter density has already been carried out, "traditional" means of sampling from this posterior density is too expensive. But another such approach – using the linearization of the mass change model f(q) (using either Automatic Differentiation or some other form of differentiation) to project the posterior uncertainty of q onto the quantity of interest – exists, and is not at all mentioned. Playing devil's advocate, such an approach assumes near-linearity of f(q), but linearity has already been assumed in the posterior calculation of q. Moreover at least two prior papers – Isaac et al (2015) and BR23 – have used this method (see eq. 24 of Isaac et al 2015, or eq. 15 of BR23), and the latter comprehensively tested the linearity assumption. Given this, I would expect acknowledgement of this very relevant and related approach, its drawbacks and benefits, and fit (or lack thereof) to the current problem"*

Again, we apologize for missing the work in BR23. We agree that the method we used to compute the Laplace approximation of the posterior (also used in Isaac et al (2015) and BR23 is only exact if the model used to predict the observations is linear. This approach was only computationally feasiable for us (and in the other papers) because we could solve adjoint equations to compute the action of the Hessian of the misfit (between the model and the observations) on a vector using our steady-state model implemented in MALI. However, our codes do not have the ability to solve adjoint equations to compute gradients of the transient model used to predict mass change at year 2100. Consequently, we could not linearize the parameter-to-QoI map as done in BR23. A forward finite difference approximation of the gradient would require 11,537 model evaluations. However, if the parameter-to-QoI could be linearized computationally efficiently, using it to compute the mean and variance of mass change would introduce an error because the map is nonlinear see Figure 1). Specifically, Figure 1) plots one-dimensional sweeps through the 11,536 dimensional parameter space used to represent Basal friction computed using the lowest fidelity model in our 13 model ensemble, that is $SSA_{3km,365days}$. Each sweep is along a random direction through the parameter space that pass through the origin (which corresponds to using the mean friction field). The extremes of the sweep correspond to $\pm\sigma$, where $\sigma$ is the standard deviation of the posterior along the sweeps. It is clear that the parameter-to-QoI map is nonlinear but because we cannot linearize the map we cannot compute the error introduced when using it to estimate the mean and variance of mass change. However, if a linearized map was available we could use it as an additional computationally efficient low-fidelity model when using MFSE to compute statistics. The exact benefit of doing so would depend on the correlation between the new model and the other models we used in the paper.

We added the following remark to Section 5.1 of the revised manuscript:

The models we used here are all different numerical discretizations of the two different physics models. However, in the future we could also use alternative classes of low-fidelity models if they become available. For example, we could use linearizations of the parameter-to-QoI map (as done in (Recinos et al., 2023), if our MOLHO and/or SSA codes become capable of efficiently computing the gradient of the map by solving adjoint equations or by using automatic differentiation. Such an approach would require only one non-linear forward transient

solve of the governing equations followed by lone linear solve of the corresponding backward adjoint. Once constructed, the linearized map could then be evaluated very cheaply and used to reduce the MSE of the MFSE estimators, provided the error introduced by the linearization was not substantial. Other types of surrogates could also be used in principle, however, the large number of parameters used pose significant challenges to traditional methods such as the Gaussian processes used in Jantre et al. (2024). Recently developed machine-learning surrogates (Jouvet et al., 2021; Brinkerhoff et al., 2021; He et al., 2023) could be competitive alternatives to the low fidelity models considered in this work.

[Figure]

Figure 1: One-dimensional sweeps through the 11,536 dimensional parameter space used to represent Basal friction. Each sweep is along a random direction through the parameter space that pass through the origin (which corresponds to using the mean friction field). The extremes of the sweep correspond to $\pm\sigma$, where $\sigma$ is the standard deviation of the posterior along the sweeps.

We did not include Figure 1 in the revised manuscript because we believe it distracts from the main message of the manuscript, that is that MFSE can substantially reduce the computational cost of computing statistics of prediction uncertainty. However, we did add the following statements and plots.

The left panel of Figure 13 (Figure 2 in this response document) plots the time evolution of mass loss predicted by the three models selected by our final ACV estimator. The right panel plots the distribution of mass loss at the final year, 2100, computed using the $\mathrm{SSA}_{1.5km,365days}$ model. The bias of the $\mathrm{SSA}_{1.5km,365days}$ is clear in both plots, for example, in the right panel the mean of the blue distribution is not close to the mean computed by the ACV estimator. However, we must emphasize that, by construction, the ACV estimate of the mean mass loss, and its variance, is unbiased with respect to the highest-fidelity model $\mathrm{MOLHO}_{1km,9days}$. We also point out that while our Laplace approximation of the posterior is a Gaussian, the push-forward of this distribution through the $\mathrm{SSA}_{1.5km,365days}$ model model is nonlinear. Specifically, the push-forward of a Gaussian through a linear model remains Gaussian; however, in this case, the right tail of the push-forward density is longer than the left tail, indicating that it is not Gaussian. This suggests that the mapping from the basal friction parameters to the quantity of interest is nonlinear. We were unable to compute reasonable push-forward densities with the simulations obtained from the other two models used to construct the ACV estimator due to an insufficient number of simulations. However, we believe it is reasonable to assume that the parameter-to-QoI map if these models is also non-linear.

[Figure]

Figure 2: (*Left*) The evolution of mass loss predicted by the three models we used in our final ACV estimator, corresponding to each of the simulations used to construct the estimator. (*Right*) The probability of mass loss computed using the $SSA_{1.5km,365days}$ model. The black vertical line represents the ACV estimate of the mean, while the gray shaded region represents plus and minus 2 standard deviations, again computed by the ACV estimator.

**4.3   Specific comments**

157. *"L23-25. This is a good outlay of the different sources of uncertainty. What is missing is a definitive statement that the only type of uncertainty being quantified in this paper is parametric uncertainty."*

    We now state explicitly that we are quantified parametric uncertainty in the introduction. We also now point out that model form error and model discretization error impact the bias of our MFSE estimates of mean and variance in section 4. See our response to comment 63 for details.

158. *"L26-27. "but the impact of discretization errors has not been explicitly considered with other sources of uncertainty". And it has not in this study either, right? As I understand it the MFUQ scheme is solely to estimate parameter uncertainty of the 1km, 9-day MOLHO model – it did not quantify disc. uncertainty despite using different discretizations."*

    You are correct. We apologize for our sloppy statement that may have led reviewers to believe we were quantifying discretization uncertainty. We have edited the introduction accordingly. See our responses to comments 155 and 63.

159. *"L37-41. This is a good place to cite works such as Isaac et al 2015 (and various papers by Noemi Petra e.g. Petra et al 2013), and BR23."*

    See our response to comment 155 that explains our changes to the introduction intended to clarify which studies are low-dimensional and which are high-dimensional.

160. *"L60-61. As noted above, quantifying the impact of a high-dimensional parameterization of basal friction on long-term projections is not novel (cf. BR23 – unless you are distinctly saying that 40 years is not long-term and 80 years is!)"*

    We were not aware of BR23 and now we have read agree that quantifying the impact of a high-dimensional parameterization of basal friction on long-term projections is not novel.

Consequently, we removed that claim and now state in the introduction: We discussed the changes made to the introduction in our response to comment 155.

161. *"L62-64. As noted above, Isaac et al, whose methodology you cite and use, arguably did this."* We do not think that Isaac et al *"quantified the impact of a high-dimensional parameterizations of basal friction on long-term ice-sheet projections"* as stated in the first sentence on line 62. However, we agree we are using the method from Isaac et al for inference. We hope our revised introduction clarifies these facts.

162. *"L66. I'm not sure why you include Isaac 2015 in a list of papers using low-dimensional parameterisations – they used $O(10^6)$ parameters in their basal sliding parameterization."*

    The original manuscript stated *"In contrast, previous UQ studies (Nias et al., 2023; Ritz et al., 2015; Schlegel et al., 65 2018; Jantre et al., 2024) only employed low-dimensional parameterizations despite high-dimensional parameterizations being necessary to calibrate ice-sheet models to observational data (Barnes et al., 2021; Isaac et al., 2015; Johnson et al., 2023; Perego et al., 2014)"*. We cited Isaac et al to point out that high-dimensional parameterizations are needed to calibrate ice-sheet models well not saying they used a low-dimensional parameterization. Again we hope the new introduction clarifies this point.

163. *"Fig 2, 3, 5, 6, 7, and 13: you need to show the coordinate axes in all visualisations of the model domain – and there should be one figure showing the placement of Humboldt in Greenland."*

    We regenerated all figures in the revised document to improve readability.

164. *"L181: "covariance" – prior or posterior?"*

    We changed it to prior covariance.

165. *"L190-193. I have deep concerns about your parameter choices. Firstly, what is the pointwise variance? Secondly, how did you arrive at this correlation length as suitable – on what basis? I do not see any physical reasoning leading to it. You are saying that the data essentially does not need to constrain variability on a scale smaller than this, which I don't think is an accurate statement. BR23 chose far smaller autocorrelations ( 3km) using some degree of physical inference, and moreover showed that it was necessary to give reasonable values of posterior uncertainty (see comment on TABLE 1 regarding this assessment), and it is possible that in choosing such large numbers you are making the posterior uncertainty artificially small by choosing an overly-informative prior. This may be why you only needed ¡ 1000 eigenvalues to represent the posterior as shown in the appendix. (see BR23 for details.)"*

    Thank you for pointing us to the reference BR23. We believe it provides a computationally tractable method for tuning the hyper-parameters of the prior when automatic differentiation (AD) is available with an ice-sheet code. Unfortunately, our codes do not have AD capabilities for transient simulations. (We do have them for steady state simulations. Indeed we use AD to compute the action of the hessian when constructing the Laplace approximation of the posterior.) Because we did not have AD, we had to tune the hyper-parameters heuristically. We informally varied the hyper-parameters of the prior and used our judgment to pick a correlation length and variance that resulted in a posterior MAP point that was able to match the observations well. We have expanded Remark 5.1 to better discuss how we heuristically choose the parameters: "In this study we used our domain experience to determine the best values of the prior hyper-parameters $\gamma, \delta, \eta$ reported in Section 2.3 and the likelihood hyper-parameter $\alpha$ reported in Section 3. However, varying these hyper-parameters, would likely

change the estimates of uncertainty in ice-sheet predictions produced by this study. Similar to previous studies (Isaac et al., 2015), we did not investigate these sensitivities extensively. We heuristically chose the prior hyper-parameters so that the prior samples would have a variance and spatial variability that we deemed inline with our experience. Further, we found that reducing $\alpha$ substantially from the value we ultimately used while keeping the prior hyper-parameter fixed prevented the MAP point from capturing the high-frequency content of the basal friction field needed to accurately match the observed surface velocities. Future studies should investigate the sensitivity of mass change to the values of the hyper-parameters more rigorously using an approach such as the one developed by Recinos et al. (2023)."

Despite not using the rigorous tuning procedure in BR23, we believe that our calibration is still close to state-of-the-art and is sufficient to demonstrate the ability of MFSE to reduce the computational cost of computing uncertainty. Moreover, the results in BR23 have further motivated us to consider adding AD capabilities to our models for future studies. However, doing so will require substantial human hours and so will not be feasible for this paper.

166. *"L205. How did you generate mass balance? Did you run a regional climate model that incorporates firn and snow processes? If so, say so. Did you use a parameterization? If so, state it and the source."*

     We added the following to Section 2.4: *"Second, the MIROC5 climate forcing from the CMIP5 for the Representative Concentration Pathway (RCP) 2.6 scenario was used to generate the surface mass balance (difference between ice accumulation and ablation) $f_H$ and drive the ice-sheet evolution from 2007 to 2100."* This surface mass balance was provided by the Ice Sheet Model Intercomparison Project for CMIP6 (ISMIP6), which down-scaled output from Earth system models using the state-of-the-art regional climate model MAR (Nowicki et al., 2020).

167. *"L231. On what basis do you assume they are uncorrelated? The fact that the products are not posted with spatial correlations of error is not a reason – this is simply too difficult for them to calculate. Please highlight this, and state what the consequences of such an assumption could be for estimating posterior uncertainty."*

     We have added the following statement.

     In this study we assumed that the observational data are independent, as also assumed in (Recinos et al., 2023) Moreover, we also assumed our Gaussian error model to be exact. However, neither of these assumptions are likely to perfect in reality. Consequently, our results must be viewed with some caution. For example, Koziol et al. (2021) showed that, for an idealized problem, ignoring spatial correlation in the observational noise can lead to uncertainty being underestimated.

168. *"Section 4: in general I think this section should be read over very carefully to look for typos and variables introduced without definition. Ill mention several below but these sections (the ones that I read closely) seem to have been written hastily."*

     We have corrected mistakes pointed out by you and the other reviewers as well as some additional ones. We will also spend considerable effort to improve section 4 in the revised manuscript.

169. *"L263, mean $Q^\mu$: mean of what?? And what is Q? and are these "true" statistics or estimators since they have no subscript?"*

     We have made extensive edits to this section, including clarifying what $Q^\mu$ represents.

170. *"L265. Try to be consistent with tense throughout, and definitely within a sentence: "The second step simulates the model at each realization ... and computed the mass change..""*

    We have corrected tense here and throughout the document. See comment 66.

171. *"L271: "Any MC estimator Q"* – do you mean $Q_\alpha^\mu$ or $Q_\alpha^{\sigma^2}$, or both or neither?"*

    See response 40.

172. *"Eq 11 – can you show how this is derived? At first glance it looked similar to the identity $E[(X - E[X])^2] = E[X^2] - (E[X])^2$ but I could not derive it using similar reasoning."* The right most expression is correct. However the middle expression had a typo which has been corrected. You can find a derivation of the final expression for bias in one of our software tutorials. `https://sandialabs.github.io/pyapprox/auto_tutorials/multi_fidelity/plot_monte_carlo.html`. We did not include this proof in the paper.

173. *"L279 did you mean MSE (II), rather than MSE (11)?"*

    We mean Eq. 11. All in text equation references in the paper have been changed from (EQNO) to Eq. (EQNO).

174. *"L279: I don't believe that all of these sources of uncertainty go into the bias term. My interpretation is that, for the purpose of your MFUQ, you are given a density of q arising from the Isaac methodology. You then have a deterministic function fa(X) which is given by your high fidelity model and its discretization, and is therefore deterministic. You are seeking properties of the probability distribution induced by fa and the only actual uncertainty is how fast the MC converges. Model uncertainty and discretization uncertainty, while very real, are not accounted for in such a calculation."*

    Line 279 stated *"The bias term of the MSE (11) is caused by using a numerical model, with inadequacy and discretization errors, to compute the mass change."*

    We hope that our response to comment 63 answers this question.

175. *"L280 what does MSE (10) mean?"*

    We now say Constructing a SFMC estimator with a small MSE, Eq. (10), ...

176. *"L280 ensures, for any set of model input samples,"*

    Fixed.

177. *"First eq in 4.2.1 (not numbered) – is the 2nd term in brackets not divided by N1?"* You are correct. See response 15.

178. *"L316 – QoI not defined previously."*

    We now define QoI as quantities of interest the first time it is introduced.

179. *"L324 – for the union of these sets to be null, both need to be null. Should it be an intersection symbol?"*

    We fixed this mistake.

180. *"Eq 18. You seem to be estimating these statistics using straighforward (Naïve) MC. Why is this OK given the whole thrust of your study is that MC is too expensive to apply to the statistics of the ice model?"*

You raised an important aspect of MF UQ. All current theory assumes that quantities such as (18) in the original manuscript are known exactly. However, in practice they must be estimated using estimates such as (18), typically with a small number of so called pilot samples. An important contribution of this paper is to show that estimating these quantities introduces an error that can be non-trivial. We also provide a strategy for estimating the impact of this pilot error. See figures 10a, 11, a, b and c in the initial submission. In the initial submission we stated the following in the second paragraph after equation (18).

*"Unfortunately, using a finite P introduces errors into (16) and (17), which in turn induces error in the ACV estimator covariance. This error can be decreased by using a large P but this would require additional evaluations of expensive numerical models, which we were trying to avoid. Consequently, in this study we investigated the sensitivity of the number of pilot samples on the accuracy of ACV MC estimators."*

181. *"L418-422. State # of elements In models"*

We now state:

The number of elements associated with the four meshes with characteristic element sizes 1km, 1.5km, 2km and 3km, were 2611, 9238, 13744, 22334, respectively. The number of nodes for the same four meshes were 1422, 4846, 7154, 11536.

182. *"L424 in the 1st para of 2.4 you state you use FEniCS. MALI is a C++ model with Fortran libraries and not, to my knowledge, written with fenics. Which model(s) did you use???"*

We have added the following to the paper to Section 5.1:

Lastly, note that we used a different model, to the 13 described above, for the Bayesian calibration of the basal friction parameters. Specifically, we used the C++ code `MALI` (Hoffman et al., 2018), which can solve the Blatter-Pattyn equations (Pattyn, 2003; Dukowicz et al., 2010) and compute the action of the Hessian on a vector. `MALI` efficiently computed these Hessian-vector products, needed to compute our Laplace approximation of the posterior in Eq. (14), by solving the adjoint equations for the steady state Blatter-Pattyn equations. However, SSA equations (Section 2.1.3) are not currently implemented in `MALI` and the MOLHO (Section 2.1.2) equations have only recently been implemented (after the simulations for this work were perfomed). Consequently, we used `FEniCS` (Alnæs et al., 2015) to implement both MOLHO and SSA to ensure that the relative computational timings of these models would be consistent. Solving the Blatter-Pattyn model using the C++-based `MALI` code and solving MOLHO and SSA using the python based `FEniCS`, would have corrupted the MFSE results. Moreover, implementing SSA in `MALI` would be time consuming because it is currently only used to solve 3D models and not 2D models, such as SSA. Indeed, a partial motivation for this study was to to determine the utility of implementing the SSA equations in `MALI`.

183. *"Fig 10 – I might be misunderstanding the methods but shouldn't there be units??"*

Figure 10a plots a dimensionless quantity it is the ratio (variance of the MC estimators) of two quantities with the same units. We added Eq (30) to clarify the quantities plotted.

184. *"Table 1. This value is presented without validation. It is possible to do a "sanity check". BR23 use 2 essentially independent measurements of velocity (ITS_LIVE and MEaSUREs) to invert for parameters and simulate mass loss. If the difference seen is of almost negligible probability under the calculated posterior for mass loss - then there must be an issue with the calculated posterior. You are capable of doing this as well ..."*

We do not have access to two different observational data sets for the Humboldt Glacier for the year 2007. We used the best available data set (MEaSUREs) for our Bayesian calibration starting in 2007. Yet, while ITS_LIVE velocity data exists for this year, its coverage at Humboldt is limited, so we could not perform an inversion using ITS_LIVE data alone.

Additionally, while such a sanity check is indeed valuable, computing exactly the correct posterior is not the focus of this paper. Rather the goal of this paper is to demonstrate the utility of using MFSE to quantify uncertainty in the predictions of ice sheet models using a problem setup (data, calibration etc.) that is close to what is used in practice by the best papers in the literature, e.g. in BR23. Changing the prior would likely change the exact values of mass change we reported, but it would likely not substantially change the variance reduction we observed. The variance reduction of a statistic is the ratio of the single-fidelity MC estimator variance divided by the MFSE estimator variance. Thus, when estimating the mean mass change, the exact value of the statistic cancels. We have expanded the discussion of the limitations of our study in the summary and conclusions sections.

185. *"L550. "the SSA model was not.." can you provide an example or evidence of this?"* The original manuscript stated: *"While the highest-fidelity model MOLHO was capable of capturing ice-sheet dynamics that the SSA model was not, that is vertical changes in the horizontal velocities,"* Moreover, Figure 13 in the original manuscript shows that these two model produce different estimates of thickness at the final time. A reference to figure 13 has been placed in the discussion.

186. *"L565. Im confused – I thought that the MFUQ was needed as you are sampling from a distribution of 600 dimensions (the number of Eigenvals retained in the Hessian based UQ). If you have only 10 dimensions can you not use standard (naïve) MC?"*

The original submission stated. *"Our study used a high-dimensional representation of the basal friction field that is capable of capturing high-frequency modes, however it has been common in previous studies to use lower-dimensional parameterizations. Consequently, we investigated the impact of using a low-frequency/lower-dimensional representation of the friction field on the efficiency of ACV estimators using ice-sheet models. Specifically, we estimated the mean and variance of the mass change using a 10 dimensional Karhunen Loeve expansion (KLE) to represent the posterior uncertainty of the basal friction field (complete details are presented in B). "*

Note, while we retained 1125 modes from the prior-preconditioned hessian, computing uncertainty in the QoI still required sampling the 11,536 variables used to parameterize the friction field. Only the KLE study required sampling 10 variables. Moreover, the dimensionality of the parameter space does not explicitly effect the MSE of a MC estimator, see equations (11) and (12) in the original manuscript. We explored the use of MFMC because the number of variables parameterizing the friction field was 11,536 (reported in the introduction of the original submission) prevented us from using surrogate methods. We also do not have the capability to compute gradients of mass loss using adjoint methods such as done in BR23. Our investigation of the impact of using a 10 term KLE was to show that one must avoid the temptation to use a lower-dimensional parameterization of the friction field, to enable the use of surrogates or increase the performance of MFSE, as doing so severely underestimates uncertainty. BR23 shows this very well and we have added a citation to that paper. Specifically, we now state in the discussion

[revised manuscript text omitted]
_{\text{ACV}}) = -\mathbb{Cov}_\Theta[\boldsymbol{Q}_0, \boldsymbol{\Delta}]\mathbb{Cov}_\Theta[\boldsymbol{\Delta}, \boldsymbol{\Delta}]^{-1}, \qquad \mathbb{Cov}_\Theta[\boldsymbol{Q}_0, \boldsymbol{\Delta}] \in \mathbb{R}^{2 \times 2M}, \ \mathbb{Cov}_\Theta[\boldsymbol{\Delta}, \boldsymbol{\Delta}] \in \mathbb{R}^{2M \times 2M}, \tag{26}$$

which produces an ACV estimator with covariance

$$\mathbb{Cov}_\Theta[\boldsymbol{Q}_{\text{ACV}}, \boldsymbol{Q}_{\text{ACV}}](\Theta_{\text{ACV}}) = \mathbb{Cov}_\Theta[\boldsymbol{Q}_0, \boldsymbol{Q}_0] - \mathbb{Cov}_\Theta[\boldsymbol{Q}_0, \boldsymbol{\Delta}]\mathbb{Cov}_\Theta[\boldsymbol{\Delta}, \boldsymbol{\Delta}]^{-1}\mathbb{Cov}_\
[revised manuscript text omitted]

---

## Author Response (AR2)

Thank you again to the reviewers for their additional comments. Below we respond to each comment. Changes made while responding to the reviewers remarks are made in red in the revised document.

**1 Reviewer 1 (Douglas Brinkerhoff)**

No further comments were provided.

**2 Reviewer 2 (Vincent Verjans)**

**2.1 General Comment**

*"This is my second review of this manuscript. Firstly, I want to commend the authors for their thorough work on the revisions. I find this second version of the manuscript much improved compared to the first submission. The methodology is much better explained, and the presentation quality is also greatly improved. At this stage, I have no major reservation concerning the manuscript. However, I still have plenty of minor technical comments, mostly focused on presentation details. Once these comments are addressed, as well as possible comments from other reviewers, I believe that this study will be a valuable contribution to the field of uncertainty quantification in ice sheet modeling."*

**2.2 Specific comments**

- *"L72 "Basal friction field" should not be capitalized."*

  Fixed.

- *"L78 I believe that "mean and mass" should be mean and variance."*

  Fixed.

- *"L83 Typo: "defined a realistic physical domains"."*

  Fixed.

- *"L85 Typo: "a ice sheet model"."*

  Fixed.

- *"L86 I suggest specifying: two uncertain scalar parameters. Otherwise, one could think that the current study investigates only a single parameter (the basal friction field), while Gruber et al. (2023) quantifies uncertainty from two parameters."*

  Fixed.

- *"L120 For this summary sentence to be useful, it should include information about the Stokes model as well."*

  We removed the sentence completely, as it has caused confusion in multiple reviews and is captured in the previous discussion.

- *"L158 Why is there an extra term $\rho g(s - z)$ in the boundary condition on $\Gamma_m$ here compared to the Stokes model?"*

  The Stokes boundary condition on $\Gamma_m$ is

  $$\sigma \mathbf{n} = \rho_w \, g \, \min(z, 0)\mathbf{n},$$

or, equivalently,

$$2\mu D\mathbf{n} = p\mathbf{n} + \rho_w\, g\, \min(z, 0)\mathbf{n}.$$

The Blatter-Pattyn and MOLHO model use the approximated condition

$$2\mu\hat{\mathbf{D}}\,\mathbf{n} = \rho g(s - z)\mathbf{n} + \rho_w\, g\, \min(z, 0)\mathbf{n},$$

where $\hat{\mathbf{D}}$ is an approximation of $\mathbf{D}$, and $\rho g(s - z)$ is the hydrostatic approximation of $p$. Therefore the term $\rho g(s - z)$ is not an "extra" term, but just the approximation of the ice pressure $p$, which is included in the Stokes boundary condition.

- *"L172 "integrated by part": part should be plural."*

  Fixed.

- *"L193 (...) quantifying uncertainty in modeled ice mass change subject to high-dimensional parameter uncertainty (...)."*

  Fixed

- *"L206 Typo: "a infinite-dimensional"."*

  Fixed.

- *"L240 Please use the wording in thermal equilibrium."*

  Fixed.

- *"L251 Please specify two-dimensional surface velocities, so that the dimensionality $2N_{obs}$ is immediately clear to the readers."*

  Fixed.

- *"L257 is approximately in steady-state."*

  Fixed.

- *"Equation 12 I apologize for this request, but I consider it bad practice to use a same symbol for different notions. In this manuscript, $\alpha$ is used to denote the likelihood hyper-parameter as well as for model indexing. Please consider using another symbol here."*

  We now use the symbol $\xi$ for the scaling term in Eq. 12.

- *"L263 I believe that it is the errors in the observational data that are assumed to be uncorrelated, not the data themselves."*

  Fixed.

- *"L268 Please use the wording in thermal equilibrium."* Fixed.

- *"L270 Please cite the work of Adalgeirsdottir et al. (2014) here."*

  The citation was added.

- *"L321 Typo: "In out study"."*

  Fixed.

- *"L331-332 Please italicize I and II to be consistent with Eq. (18)."*

  Fixed.

- *"L349 Although equivalent, this should be $Q_0 - Q_\infty$."*

  Fixed.

- *"L368 "MSE error" should simply be MSE."*

  Fixed.

- *"L388 Typo: "can be simulate"."*

  Fixed.

- *"L388 Typo: end sentence before "However"."*

  Fixed.

- *"L388-390 This statement is not clear to me: (i) Even though it is assumed that the discretization can be refined indefinitely, in practice, how can $Q_\infty$ be evaluated? (ii) What is the advantage of having a numerical discretization error equal to $V_\Theta[Q_{ACV}]$? Finally, note that the reference to Eq. (18) should be Eq. (19) here."*

  We added the statement: Balancing these two errors ensures that computational effort is not wasted on resolving one source of error more than the other. However, in practice, the geometric complexity of many spatial domains makes generating large numbers of meshes impractical, and estimating the discretization error using techniques like posterior error estimation can be challenging. The correct equation is now referenced.

- *"L394 Typo: 0 and 1 should be subscripts."*

  Fixed.

- *"L413 Please add a sentence to explain intuitively the advantage of having $\eta_{1,2}, \eta_{1,4}, \ldots, \eta_{1,2M}$ and $\eta_{2,1}, \eta_{2,3}, \ldots, \eta_{2,2M-1}$ potentially different from 0. That is, what is the advantage of including the terms capturing differences in $\sigma^2$ affecting $Q_{ACV}^\mu$, and the terms capturing differences in $\mu$ affecting $Q_{ACV}^{\sigma^2}$?"*

  We added the statement Formulating the control variate weights as a matrix enables the ACV estimator to exploit the correlation between the statistics $Q^\mu$ and $Q^{\sigma^2}$, producing estimates of these individual statistics with lower mean squared error (MSE) than would be possible if the two statistics were estimated independently.

- *"L445 These equations were included in the previous iteration of the manuscript. Instead of referring the reader to Dixon et al. (2024), I strongly recommend to include the equations for these quantities into an Appendix C."*

  We included only some of the equations needed in our initial submission. At the suggestion of another reviewer, we removed these equations from our first revision because including all equations would require many pages and additional notation. We still believe that adding these equations would overly complicate the paper when they can be easily found in the cited paper and in tutorials provided by PyApprox which was used to generate the results.

- *"L452 This should be: to the number of pilot samples of the error in ACV MC estimators."*

  Fixed.

- *"L470 two-model."*

  Fixed.

- *"L474 two-model."*

  Fixed.

- *"L490 I believe that "next highest-fidelity" should be next higher-fidelity."*

  Fixed.

- *"L513 "MSE error" should simply be MSE."*

  Fixed.

- *"L520 Please remove "sorely needed"."*

  Fixed.

- *"L520 Typo: "and the error a multi-fidelity estimator"."*

  Fixed.

- *"L540-542 There is an error here. The mesh with lowest characteristic element size should have the most elements and nodes."*

  Fixed.

- *"L560 "Solving the Blatter-Pattyn model using the C++-based MALI code and solving MOLHO and SSA using the python based FEniCS, would have corrupted the MFSE results": from this paragraph, I understand that this is exactly what the authors did. Sorry if I misunderstand this, but I think that more clarity would be beneficial."*

  Sorry for the typo. We meant to say MOLHO instead of Blatter-Pattyn. We now state Solving the MOLHO model using the C++-based `MALI` code and solving and SSA using the python based `FEniCS`, would have corrupted the MFSE results.

- *"Figures 5 and 6 Please mention explicitly in the captions that the color scales of different sub-panels span different ranges."*

  We added the following statement to both plots: Note, the color scales of each plot span different ranges so that the variability in the quantities plotted is visible.

- *"L588 "hyper-parameter" should be plural."*

  Fixed.

- *"Remark 5.2 In my previous review, I made a comment about the potential sensitivity of the low-resolution models to the interpolation method used for the basal friction field. Essentially, I argue that the results from the low-fidelity models are likely sensitive to the interpolation method used, because the authors demonstrate (Figure 7) that the simulated thickness change is sensitive to high-frequency variability in the basal friction field. I believe that my comment has not been addressed appropriately. In particular, I believe that the authors should at least (i) state clearly which interpolation method is used (linear, nearest neighbor, polynomial, other), (ii) point out that the basal friction field is too smooth due to the interpolation, as suggested by Figure 14, and (iii) explicitly state that using interpolation methods preserving more of the high-frequency variability in basal friction would likely improve the ability of the low-resolution models to reproduce $f_0$. Points (ii) and (iii) are mentioned in the first paragraph of Appendix B, and should be incorporated in the main text."*

We added the following statement to the remark Specifically, we used the linear finite element basis of the fine mesh to interpolate onto the coarser meshes. This procedure ensured that varying the basal friction field (the random parameters to the model) would affect each model similarly, regardless of the mesh discretization employed. However, the linear interpolation we used may have overly smooth the friction on coarse meshes relative to alternative higher-order interpoolation methods. Consequently, using alternative interpolation methods may increase the correlation between the mass loss predicted by the coarse meshes and that predicted using the finest mesh. However, we did not explore the use of alternative iterpolation schemes because our results demonstrate that linear interpolation still produces models that can be used produce a computationally efficient MFSE.

- *"L603 Typo: "of the evaluating all 13 models"."*

  Fixed.

- *"Figure 9 Because of the colormap chosen and the white text font, some correlation entries are impossible to read."*

  Fixed.

- *"Figure 9 caption Typo: two commas."*

  Fixed.

- *"L623 If the authors are referring to Peherstorfer et al. (2016), then they should cite it here."*

  Most papers, if not all papers, make this assumption.

- *"L642 I believe that the language here could be slightly confusing. The authors never use "all models simultaneously" because they impose maximum 4 models per sample. I know this is not what is meant here, but more precise language would be beneficial."*

  We now state: Moreover, bootstrapping the estimators also revealed that not all models are equally useful when reducing the variance of the ACV estimator. In some cases, using 3 models was more effective than using four models.

- *"L651 I believe that "next highest-fidelity" should be next higher-fidelity."*

  Fixed.

- *"Figure 11 In my previous review, I asked the authors to specify the number of samples for each case (2, 3, and 4 models). The authors responded that no one number can be provided. I do not understand this. If the authors can make a boxplot, it means that they have a list of values (in this case, values of variance reduction). As such, they can provide the number of values from which each single boxplot was made."*

  We apologize for the confusion, However, we are still not sure what number of samples you are referring to. As stated on L663 of the first revision " 21 different bootstraps of the final 30 pilot samples were used to quantify the error of the variance reductions caused by only using a small number of pilot samples. " Thus, 21 different samples of the covariance matrix were used to generate the box plots for the 2, 3, and 4 model cases. Each bootstrap randomly selected 30 samples, with replacement, from the evaluations of each model at the 30 unique pilot samples. In our last response, the number of samples we referred to when stating "no one number can be provided" was the optimal number of samples allocated to each model for each bootstrap.

- *"Figure 11 Please increase the font size of the median values."*

  Fixed.

- *"L666 Typo: "this increasing"."*

  Fixed

- *"L666 Here, "which" refers to the increased computational cost of the pilot study. However, the computational cost does not affect the variance reduction value. Instead, as explained in the following paragraph, the lower median variance reduction is caused by having more pilot samples of the highest-fidelity model for the SFMC estimator. Please avoid this confusing wording."*

  The increased computational cost of the pilot study does affect the variance reduction value. We have added the statement: This fact can be explained by recalling the SFMC estimator variance, e.g. $\mathbb{V}_\Theta \left[Q_0^\mu\right]$, was obtained using a computational budget equivalent to 160 high-fidelity evaluations plus the computational cost of collecting the pilot model evaluations. In contrast, the ACV estimator variance, e.g, $\mathbb{V}_\Theta \left[Q_{\mathrm{ACV}}^\mu\right]$, does not depend on the number of pilot samples. Therefore, while increasing the number of pilot samples decreases the SFMC estimator variance, it does not decrease the the ACV estimator variance. Consequently, increasing the pilot cost reduces the variance reduction achieved by the ACV estimator.

- *"L687 Please rephrase to make totally clear that it is each one of the 160 model evaluations that takes 4.18 hour, and not "evaluating the high-fidelity model 160 times" that takes 4.18 hour."*

  We now state: The exploitation cost was fixed at the beginning of the study to the computational cost equivalent to evaluating the high-fidelity model 160 times, with each simulation taking a median time of 4.18 hours to complete for a single realization of basal friction.

- *"L698-699 Please specify that this statement about error from spatial versus temporal discretizations is valid within the ranges of discretizations tested here."*

  Fixed.

- *"L700-703 Please break this sentence in two."* Fixed.

- *"L704 Add units."*

  Fixed.

- *"L716 Typo: "model model"."*

  Fixed.

- *"L720 Typo: "if" should be of."*

  Fixed.

- *"L732-733 Needs rephrasing: "Similarly, for the exploitation phase. Moreover, each simulation ca can be computed in parallel."."*

  We now state Each simulation run in the pilot stage can be executed in parallel without communication between them. Similarly, in the exploitation phase, each simulation can also be computed in parallel.

- *"L756 Typo: citation format."*

  Fixed.

- *"L760 Typo: "estmators"."*

  Fixed.

- *"L784 Change: correlation with models."*

  Fixed.

- *"L805 Rephrase: "facilitate computationally efficiently quantify"."*

  Fixed.

- *"L832 Typo: "In out study"."*

  Fixed.

- *"L850 Add comma after "limitations"."*

  Fixed.

- *"L919 Change "Please refer to Figure B5" to simply putting (see Fig. B5). Sorry if my comment in my previous review was unclear."*

  Fixed.

- *"Caption of Figure B3 Add space between Eq. and (B1)."*

  Fixed.

- *"Caption of Figure B4 Add space between Eq. and (B1)."*

  Fixed.

**3  Reviewer 3 (Dan Goldberg)**

**3.1  General comments**

*"I have read in detail the responses of the authors to my comments, and appreciate they have worked extensively both to improve the manuscript and address my concerns. I think if the other reviewers are satisfied then [this was cutoff in the user portal]*

*However, my one remaining concern is about the strength of the prior, in particular the autocorrelation scale of 30 km, which i still am confused about as it seems large, and wonder if the authors can give some physical or observational justification. is it based on velocity or bed semivariograms? I fully accept this is not a variable the authors can tractably explore, but i feel that it should be defensible, given the accepted knowledge that bed conditions and water pressure (which determines bed conditions) vary on quite short length scales. And if not, then the authors should caveat – \*\*very clearly\*\* – that these scales are very difficult to know a priori, and that overly informative prior densities could significantly impact posterior mass loss estimates. (they make clear in their response that the QoI uncertainty is not the main thrust of the paper so doing so should not affect their main results)*

*I may seem I am being pedantic, but this is a paper on ice-sheet uncertainty quantification, of which there are very few, which is exactly why such things should be brought to light. "*

We now state the following in the final paragraph of the conclusion section of the paper.

Finally, this study demonstrated that MFSE can be used to reduce the computational cost of quantifying parametric uncertainty in projections of a single glacier, which suggests that MFSE could plausibly be used for continental-scale studies of ice-sheet evolution in Greenland and Antarctica. However, the predicted mean mass loss from Humboldt Glacier that we reported, should be viewed with caution, as the length scales of the prior distribution we employed for the uncertain basal friction field were not finely tuned, e.g. as done by Recinos et al. (2023), due to the computational expense of such a procedure. Since the length scales of basal friction are very difficult to determine a priori, the potentially over-informative length scales used in this paper could substantially impact the posterior mass loss estimates. Future research should address this issue and increase the complexity of this study in two further directions.